# Varying Coefficient Tensor Regression

Jiaxin He

Department of Statistics and Data Science

National University of Singapore

Xuelin Zhu

Department of Statistics and Data Science

National University of Singapore

Binyan Jiang

Department of Data Science and Artificial Intelligence

The Hong Kong Polytechnic University

Jialiang Li *

Department of Statistics and Data Science

National University of Singapore

**Abstract**

We propose a new varying coefficient model for tensor data regression analysis. To manage the complexity of multi-dimensional tensors, we first employ a tensor partitioning strategy to reduce dimensionality, followed by a tensor decomposition technique for the tensor covariates. By extracting key features from the tensor covariates, we feed these low-dimensional representations into a varying coefficient model, alongside other one-way covariates. Additionally, we apply a non-concave penalty estimation to simultaneously identify the model structure and select significant predictors. A subsequent refined smoothing step enhances the model's accuracy. We study the asymptotic properties of estimated functions and coefficients. Extensive simulations are conducted to evaluate the performance of our approach. Our study is motivated by a real fundus image dataset, which is analyzed using our model to improve glaucoma management.

**Keywords:** High dimensionality, Region of interest, Tensor decomposition, Varying-coefficient model

**Mathematics Subject Classification (2020):** 62XXX

## 1 Introduction

Tensors, as multidimensional generalizations of matrices, have become increasingly prominent due to their ability to represent and analyze complex, high-dimensional data structures. Extending from low-dimensional vectors and matrices, tensor objects can encapsulate data across

---

*Corresponding author: stalj@nus.edu.sg

multiple dimensions, offering a more comprehensive representation of multiway data. This flexibility makes tensors particularly valuable in many fields, such as neuroimaging (Zhou et al., 2013; Li et al., 2018), genomics (Hore et al., 2016), social network analysis (Dunlavy et al., 2011), hyperspectral image (Liu et al., 2019), and recommendation system (Zhang et al., 2021). The inherent high-dimension in tensors (Hillar and Lim, 2013) pose difficulties and challenges in real data analysis. Recent studies in statistical tensors mainly focused on tensor decomposition (Kolda and Bader, 2009; Zhang and Xia, 2018; Zhang and Han, 2019), tensor completion (Gandy et al., 2011; Yuan and Zhang, 2016; Zhang, 2019), tensor regression (Zhou et al., 2013; Zhang and Li, 2017; Li et al., 2018; Zhang et al., 2020) and tensor clustering (Han et al., 2022a; Luo and Zhang, 2022).

Previous authors extend traditional linear regression to tensor data, including scalar-on-tensor (Zhou et al., 2013), tensor-on-tensor (Lock, 2018; Liu et al., 2020; Luo and Zhang, 2022), and tensor-on-vector (Li and Zhang, 2017; Sun and Li, 2017), among others. In these problems, the number of unknown parameters can be enormous, often far exceeding the sample size, thus imposing a significant computational burden on the estimation. To address this challenge, dimension reduction tools are necessary for an efficient estimation of tensor coefficients, including the CP decomposition-based regression (Zhou et al., 2013; Sun and Li, 2017; Bi et al., 2018; Hao et al., 2020), the Tucker decomposition-based regression (Li et al., 2018; Guhaniyogi et al., 2017; Zhang et al., 2020; Han et al., 2022b; Luo and Zhang, 2023), and the tensor train decomposition-based regression (Liu et al., 2020, 2024), among others. In addition to aforementioned low rank representation methods, one may further impose a sparsity structure and use regularization in the estimation (Zhou et al., 2013; Zhang et al., 2020; Liu et al., 2020)

Most earlier works only examined a parametric linear relationship between tensors and other variables. To capture non-linear association, varying coefficient model (VCM) offers a more flexible framework (Hastie and Tibshirani, 1993; Cai et al., 2000; Fan and Zhang, 1999). There exist a large body of literature for the development of estimation methods for VCM, such as the local linear method (Zhang et al., 2002; Xia et al., 2004; Fan, 2018; Lu, 2008), the profile least squares method (Fan and Huang, 2005), the average derivatives method (Newey and Stoker, 1993), and the smoothing spline method (Ahmad et al., 2005). VCM is especially useful for addressing dynamic and complex data problems. Cheng et al. (2014) and Cheng et al. (2016) developed a comprehensive framework of VCM in longitudinal/cluster data. Mu et al. (2018) applied a bivariate spline on triangulation to estimate the functional coefficient of space domain data. Yu et al. (2022) used tensor-product splines to evaluate the spatio-temporal non-stationarity. Zhu et al. (2022) studied a kernel based varying coefficient network model.

From our review, very few papers introduced the nonparametric functional effect into tensor data analysis. Chen et al. (2024) proposed a semi-parametric tensor decomposition by assuming a constant core tensor and functional factor matrices generated from tucker decomposition. Han et al. (2023) developed an iterative algorithm to estimate functional factor matrix along one functional axis. Zhou et al. (2024) investigated a non-parametric tensor regression model and used the spline approximation by the scale-adjusted block relaxation algorithm under CP decomposition. None of these works considered varying coefficients for tensor inputs. We propose a new VCM for tensor regression and study its estimation methods in this work. Figure 1 dis-

plays sample images from the Glaucoma Real-world Appraisal Progression Ensemble (GRAPE) dataset (Huang et al., 2023), which forms the basis of our analysis in this paper. The longitudinal image data were collected at the Eye Center of the Second Affiliated Hospital of Zhejiang University and include 1,115 fundus images from 263 eyes, along with accompanying clinical information. In particular, increased intraocular pressure (IOP) is the most important risk factor for glaucoma and we thus focus on accurately predicting IOP based on fundus images to enable timely intervention and prevent the progression of glaucoma. In particular, we examine the complicated age-image interactions in the glaucoma management study and demonstrate the superior prediction performance of the proposed VCM.

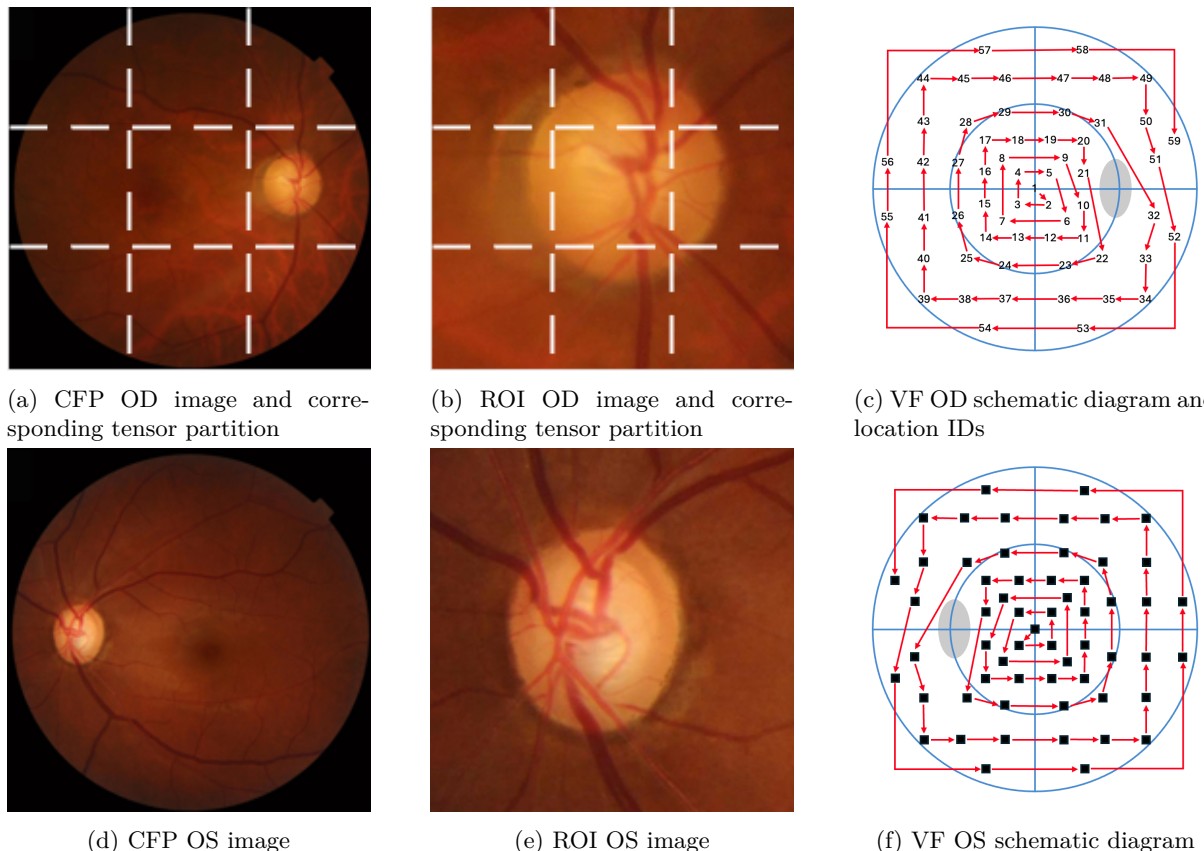

(a) CFP OD image and corresponding tensor partition

(b) ROI OD image and corresponding tensor partition

(c) VF OD schematic diagram and location IDs

(d) CFP OS image

(e) ROI OS image

(f) VF OS schematic diagram

Figure 1: Fundus image analysis: illustrations for different images in the GRAPE data set. (a) and (d): original color fundus photograph (CFP). (b) and (e): processed region of interest (ROI) image. (c) and (f): view field (VF) schematic diagram. Dashed lines in (a) and (b) show the proposed tensor partition on CFP and ROI images. The red arrowed curves in (c) and (f) show the extraction order of VF values of oculus dexter (OD) and oculus sinister(OS) respectively, and detailed location IDs reported in (c).

The varying-coefficient tensor regression model formulated in this paper can be viewed as a combination of tensor partition, tensor decomposition and varying coefficient partial linear model. Under our framework, we initially partition the whole tensor data into distinct, non-overlapping parts based on prior knowledge or distribution principles. The CP decomposition technique is applied to each partitioned tensor to extract low-rank factor representations. We only need to focus on the factor matrix along the subject dimension and with some linear algebra operation, we convert the high-dimensional functional coefficient tensor to a parsimonious functional coefficient matrix. Finally, standard nonparametric estimation methods of VCM can be implemented by vectorizing the transformed matrix covariate. Through extensive simulation

studies we demonstrate the advantage of our proposed model. We finally apply this new model to examine the GRAPE data set and make some new findings for medical research.

The rest of this article is organized as follows. In section 2, we introduce the methodology of our varying coefficient tensor regression model. In section 3, we consider variable selection and model identification using a penalization estimation. In section 4, we present the theory of our model, showing the asymptotic distribution and consistency for estimated functions and coefficients. In section 5, simulation studies are conducted to examine the performance of the proposed model. In section 6, we analyze the GRAPE data using the new model and report our findings. Technical proofs are provided in the Appendix.

## 2 Methodology

### 2.1 Notations and Preliminaries

We first review some basic mathematical facts on tensor (Kolda and Bader (2009)). In this paper, we use lowercase letters (e.g. $x$), lowercase boldface letters (e.g. $\mathbf{x}$), uppercase boldface letters (e.g. $\mathbf{X}$), and uppercase calligraphic letters (e.g. $\mathcal{X}$) to denote scalars, vectors, matrices, and order-3-or-higher tensors respectively. An order-D **tensor** $\mathcal{X} = (x_{j_1 j_2 \cdots j_D}) \in \mathbb{R}^{p_1 \times p_2 \times \cdots \times p_D}$, $j_d = 1, \cdots, p_d$, $d = 1, \cdots, D$, is a $D$-dimensional array with $\Pi_{d=1}^{D} p_d$ elements. A vector or a matrix may be regarded as order-1 or order-2 tensor respectively.

We denote the **inner product** of two tensors $\mathcal{X}, \mathcal{Y} \in \mathbb{R}^{p_1 \times p_2 \times \cdots \times p_D}$ as

$$\langle \mathcal{X}, \mathcal{Y} \rangle = \sum_{j_1=1}^{p_1} \sum_{j_2=1}^{p_2} \cdots \sum_{j_D=1}^{p_D} x_{j_1, j_2, \cdots, j_D} y_{j_1, j_2, \cdots, j_D} \tag{1}$$

An **outer product** for $D$ vectors $\mathbf{x}^{(d)} \in \mathbb{R}^{p_d}$, $d = 1, 2, \cdots, D$, is an order-D tensor denoted as $\mathcal{X} = \mathbf{x}^{(1)} \circ \mathbf{x}^{(2)} \circ \cdots \circ \mathbf{x}^{(D)}$, and such $\mathcal{X}$ is called rank-1 tensor.

The parallel factor analysis, also known as **CP decomposition**, factorizes a tensor into a linear combination of rank-1 tensors. In particular, a tensor satisfying a rank $R$ CP decomposition has the following structure

$$\mathcal{X} = \sum_{r=1}^{R} \lambda_r \mathbf{x}_r^{(1)} \circ \mathbf{x}_r^{(2)} \circ \cdots \circ \mathbf{x}_r^{(D)} \tag{2}$$

where $\mathbf{x}_r^{(d)} = (x_{j_d r}^{(d)}) \in \mathbb{R}^{p_d}$, $j_d = 1, \cdots, p_d$, $d = 1, \cdots, D$, $r = 1, \cdots, R$, are normalized rank-1 tensor, and $\lambda_1 \geq \cdots \geq \lambda_R \geq 0$ are the weights.

In this paper, suppose we observe data $\{(y_i, \mathcal{X}_i, \mathbf{z}_i, t_i) : i = 1, \cdots, n\}$ from a sample of $n$ units, consisting of $n$ independent copies of $(y, \mathcal{X}, \mathbf{z}, t)$, where for the $i$-th unit, $y_i \in \mathbb{R}$ is the response, $\mathcal{X}_i = (x_{i, j_1 \cdots j_D}) \in \mathbb{R}^{p_1 \times \cdots \times p_D}$ is an order-$D$ tensor covariate, $\mathbf{z}_i = (z_{i,1}, \cdots, z_{i,p_0})^\top \in \mathbb{R}^{p_0}$ is a $p_0-$vector of scalar covariates, and $t_i \in [0, 1]$ is an index variable. We concatenate all tensor $\mathcal{X}_i$'s into a $(D+1)$ dimensional tensor $\tilde{\mathcal{X}} = \{\mathcal{X}_i : i = 1, \cdots, n\} \in \mathbb{R}^{n \times p_1 \times \cdots \times p_D}$.

## 2.2 Tensor partition varying-coefficient model

Our interest is to develop a regression model for the association between the scalar response $y$, and its corresponding tensor covariate $\mathcal{X}$ and one-way covariates $\mathbf{z}$. We shall assume that $y$, $\mathcal{X}$ and $\mathbf{z}$ are centered. The regression coefficients for the tensor covariate are allowed to vary with an index variable $t \in [0,1]$. Specifically, for the $i$-th unit, we assume

$$y_i = \langle \mathcal{X}_i, \mathcal{A}(t_i) \rangle + \mathbf{z}_i^\top \boldsymbol{\beta} + \epsilon_i. \tag{3}$$

In model (3), the effect of tensor covariate on response is treated as varying tensor function $\mathcal{A}(t_i) = (\alpha_{j_1, \cdots, j_D}(t_i))$, where each $\alpha_{j_1, \cdots, j_D}(t_i) : [0,1] \to \mathbb{R}$ is an unknown mapping. The coefficient for one-way covariate $\mathbf{z}_i$ is a constant vector $\boldsymbol{\beta} \in \mathbb{R}^{p_0}$. In addition, we assume $\epsilon_i \sim N(0, \sigma^2)$ for $i = 1, \cdots n$, and $\mathrm{Cov}(\epsilon_i, \epsilon_j) = 0$ for any $i \neq j$. In this model, the index variable $t_i$ can be any continuous covariate. For example, in the glaucoma study, investigators are interested in examining how the effects of fundus images on the intraocular pressure change with age and we thus choose $t$ to be the patient's age in this study.

A commonly observed image tensor $\mathcal{X}$ would require the estimation of enormous unknown varying-coefficients. It is computationally prohibitive to directly estimate $\mathcal{A}(t)$ due to the high dimensionality. However, not all elements of $\mathcal{X}$ contribute to the response of interest. Often, regions of interest are confined to a small portion of the entire tensor. Therefore, it is more sensible to focus on sub-tensors that capture those local features. To this end, we consider a partition model that divides the concatenated full-sample tensor $\tilde{\mathcal{X}}$ into $S$ disjoint sub-tensor covariates $\tilde{\mathcal{X}}^{(s)}$, that is

$$\tilde{\mathcal{X}} = \cup_{s=1}^S \tilde{\mathcal{X}}^{(s)} \quad \text{and} \quad \tilde{\mathcal{X}}^{(s)} \cup \tilde{\mathcal{X}}^{(s')} = \emptyset \quad \forall s \neq s'. \tag{4}$$

Without loss of generality, we assume that the size of each sub-tensor $\tilde{\mathcal{X}}^{(s)} \in \mathbb{R}^{n \times p'_1 \times \cdots \times p'_D}$ is fixed for any $s$ such that $S = \Pi_{d=1}^D (p_d/p'_d) \equiv \Pi_{d=1}^D S^{(d)}$, where $S^{(d)}$ is the number of partitions along the $d$-th order. We denote $\mathcal{X}_i^{(s)} \in \mathbb{R}^{p'_1 \times \cdots \times p'_D}$ as partition $s$ of the $i$-th tensor covariate accordingly, and $\tilde{\mathcal{X}}^{(s)} = \{\mathcal{X}_i^{(s)} : i = 1, \cdots, n\}$.

Corresponding to the partition pattern of $\tilde{\mathcal{X}}$, we can also divide $\mathcal{A}(t)$ into $S$ disjoint sub-tensor functions $\mathcal{A}^{(s)}(t)$, that is

$$\mathcal{A}(t) = \cup_{s=1}^S \mathcal{A}^{(s)}(t) \quad \text{and} \quad \mathcal{A}^{(s)}(t) \cup \mathcal{A}^{(s')}(t) = \emptyset \quad \forall s \neq s'. \tag{5}$$

With such tensor partition technique, we can rewrite the tensor effect in (3) as $\langle \mathcal{X}_i, \mathcal{A}(t_i) \rangle = \sum_{s=1}^S \langle \mathcal{X}_i^{(s)}, \mathcal{A}^{(s)}(t_i) \rangle$.

Next we adopt a rank-$R$ CP decomposition model on each partition to further reduce the sub-tensor $\tilde{\mathcal{X}}^{(s)}$ to lower dimensional feature matrices. Specifically, we have

$$\tilde{\mathcal{X}}^{(s)} \approx \sum_{r=1}^R \lambda_r^{(s)} \mathbf{x}_{rs}^* \circ \mathbf{x}_r^{(s1)} \circ \cdots \circ \mathbf{x}_r^{(sD)} \tag{6}$$

where $\lambda_r^{(s)}$ is the weight of the $r$-th rank of the decomposition, $\mathbf{x}_r^{(sd)} = (x_{1r}^{(sd)}, \cdots, x_{p'_d r}^{(sd)})^\top \in \mathbb{R}^{p'_d}$ is the common factor along the $d$-th order of the $r$-th rank, and $\mathbf{x}_{rs}^* = (x_{1rs}^*, \cdots, x_{nrs}^*)^\top \in \mathbb{R}^n$

is the factor along the subject dimension. The elements of $\mathbf{x}_{rs}^*$ contain the primary variation in the tensor variables due to subject differences, while the common structure among the subjects is absorbed into factors $\mathbf{x}_r^{(sd)}$, $r = 1, \cdots, R$, $d = 1, \cdots, D$ (Miranda et al., 2018). If we focus on the $i$-th individual tensor covariate $\mathcal{X}_i$, following (6), we have

$$\mathcal{X}_i^{(s)} \approx \sum_{r=1}^{R} \lambda_r^{(s)} x_{irs}^* \mathbf{x}_r^{(s1)} \circ \cdots \circ \mathbf{x}_r^{(sD)} \equiv \sum_{r=1}^{R} x_{irs}^* \mathcal{U}_r^{(s)} \tag{7}$$

where $\mathcal{U}_r^{(s)} = \lambda_r^{(s)} \mathbf{x}_r^{(s1)} \circ \cdots \circ \mathbf{x}_r^{(sD)} \in \mathbb{R}^{p_1' \times \cdots \times p_D'}$. We note that $\mathcal{U}_r^{(s)}$ contains only the shared information among subjects and thus can be dropped from the regression analysis.

After the above partitioning and decomposition, essentially only $x_{irs}^*$ may contribute to the response variation and thus serve as an appropriate covariate. Consequently, the total number of measurements for an individual tensor is reduced from $p_1 \times \cdots \times p_D$ to $R \times S$. A similar method is adopted in Tang et al. (2020) to learn individualized tensor construction. If we denote $a_{rs}(t) \equiv \langle \mathcal{U}_r^{(s)}, \mathcal{A}^{(s)}(t) \rangle$ as the corresponding coefficient function of $x_{irs}^*$, $i = 1, \cdots, n$, and $\mathbf{A}(t) = (a_{rs}(t))$, model (3) now becomes

$$\begin{aligned}
y_i &= \langle \mathcal{X}_i, \mathcal{A}(t_i) \rangle + \mathbf{z}_i^\top \boldsymbol{\beta} + \epsilon_i \\
&= \sum_{s=1}^{S} \langle \mathcal{X}_i^{(s)}, \mathcal{A}^{(s)}(t_i) \rangle + \mathbf{z}_i^\top \boldsymbol{\beta} + \epsilon_i \\
&\approx \sum_{s=1}^{S} \left\langle \sum_{r=1}^{R} x_{irs}^* \mathcal{U}_r^{(s)}, \mathcal{A}^{(s)}(t_i) \right\rangle + \mathbf{z}_i^\top \boldsymbol{\beta} + \epsilon_i \\
&= \sum_{s=1}^{S} \sum_{r=1}^{R} x_{irs}^* \langle \mathcal{U}_r^{(s)}, \mathcal{A}^{(s)}(t_i) \rangle + \mathbf{z}_i^\top \boldsymbol{\beta} + \epsilon_i \\
&= \langle \mathbf{X}_i^*, \mathbf{A}(t_i) \rangle + \mathbf{z}_i^\top \boldsymbol{\beta} + \epsilon_i, \tag{8}
\end{aligned}$$

where $\mathbf{X}_i^* = (x_{irs}^*) \in \mathbb{R}^{R \times S}$. It is important to emphasize that retaining the subject-specific factor matrices $x_{irs}^*$ does not mean we ignore the shared structural information $\mathcal{U}_r^{(s)}$ across subjects. Rather, this shared spatial structure is absorbed directly into the functional coefficients as the estimating $a_{rs}(t)$ inherently accounts for the interaction between the individual-specific low-rank features $x_{irs}^*$ and the shared spatial structure $\mathcal{U}_r^{(s)}$. Throughout this paper, we shall assume that $R$ and $S$ are constants. Instead of applying the decomposition on coefficient tensor $\mathcal{A}$ (Zhou et al., 2013; Zhang and Li, 2017; Li et al., 2018), we place the low-rank assumption on the tensor covariate $\mathcal{X}$ and make no specific assumptions regarding the structure of the tensor function $\mathcal{A}(t)$. Furthermore, we do not need more assumptions on tensor covariate $\mathcal{X}$, such as Gaussian ensemble design (Zhang et al., 2020; Han et al., 2022b) or tensor restricted isometry property (Luo and Zhang, 2023; Tong et al., 2022). The model considered in this paper thus presents a novel introduction to the literature.

## 2.3 Estimating functions and parameters

For the unknown functional coefficients, we only require the usual smoothness condition which enables the local linear approximation by a first order Taylor expansion. Specifically, for $t_i$ is in

a small neighborhood of $t$, we have

$$\mathbf{A}(t_i) \approx \mathbf{A}(t) + \dot{\mathbf{A}}(t)(t_i - t), \tag{9}$$

where $\dot{\mathbf{A}}(t)$ is the first derivative of $\mathbf{A}(t)$. We may estimate the unknown functions $\mathbf{A}(t)$ and parameters $\boldsymbol{\beta}$ by minimizing the local least squares

$$\sum_{i=1}^{n} \left[ y_i - \mathbf{v}_i^{\top} \left( \mathsf{a}_t + \mathsf{b}_t(t_i - t) \right) - \mathbf{z}_i^{\top} \boldsymbol{\beta} \right]^2 K_h(t_i - t)$$
$$= \sum_{i=1}^{n} \left[ y_i - \tilde{\mathbf{v}}_i^{\top}(t) \boldsymbol{\theta}_t \right]^2 K_h(t_i - t) \tag{10}$$

where $\mathbf{v}_i = \text{vec}(\mathbf{X}_i^*) \in \mathbb{R}^{RS}$, $\mathsf{a}_t = \text{vec}(\mathbf{A}(t))$, $\mathsf{b}_t = \text{vec}(\dot{\mathbf{A}}(t))$, $\tilde{\mathbf{v}}_i^{\top}(t) = (\mathbf{z}_i^{\top}, \mathbf{v}_i^{\top}, (t_i - t)\mathbf{v}_i^{\top})^{\top} \in \mathbb{R}^{2RS+p_0}$, $\boldsymbol{\theta}_t = (\boldsymbol{\beta}^{\top}, \mathsf{a}_t^{\top}, \mathsf{b}_t^{\top})^{\top}$ and $K_h(t) = K(t/h)/h$ is a scaled kernel with bandwidth $h$. It is straightforward to derive the closed-form solution of (10) to be

$$\hat{\boldsymbol{\theta}}_t = \left[ \tilde{\mathbf{V}}^{\top}(t) \mathbf{W}(t) \tilde{\mathbf{V}}(t) \right]^{-1} \tilde{\mathbf{V}}^{\top}(t) \mathbf{W}(t) \mathbf{y} \tag{11}$$

where $\mathbf{y} = (y_1, \cdots, y_n)^{\top} \in \mathbb{R}^n$, $\tilde{\mathbf{V}}(t) = (\tilde{\mathbf{v}}_1(t), \cdots, \tilde{\mathbf{v}}_n(t))^{\top} \in \mathbb{R}^{n \times (2RS+p_0)}$, and $\mathbf{W}(t) = \text{diag}(K_h(t_1 - t), \cdots, K_h(t_n - t)) \in \mathbb{R}^{n \times n}$.

Recall that $\mathbf{A}(t)$ is the true underlying tensor coefficient functions; we use $\mathbf{A}^{\dagger}(t)$ to represent the estimator obtained directly via local linear kernel smoothing here. We thus have vectorized estimate $\text{vec}(\hat{\mathbf{A}}^{\dagger}(t)) = \hat{\mathsf{a}}_t^{\dagger} = (\mathbf{0}_{RS \times p_0}, \mathbf{I}_{RS}, \mathbf{0}_{RS \times RS}) \hat{\boldsymbol{\theta}}_t$. In practice, the bandwidth $h$ is chosen by the leave-one-out cross validation.

After function estimate $\hat{\mathbf{A}}^{\dagger}(t)$ is attained, we may evaluate the global estimator of $\boldsymbol{\beta}$ by minimizing

$$\sum_{i=1}^{n} \left[ y_i - \langle \mathbf{X}_i^*, \hat{\mathbf{A}}^{\dagger}(t_i) \rangle - \mathbf{z}_i^{\top} \boldsymbol{\beta} \right]^2. \tag{12}$$

The final estimate $\hat{\boldsymbol{\beta}}^{\dagger}$ is given by

$$\hat{\boldsymbol{\beta}}^{\dagger} = \left[ \mathbf{Z}^{\top} \mathbf{Z} \right]^{-1} \mathbf{Z} \mathbf{y}^{\dagger} \tag{13}$$

where $\mathbf{Z} = (\mathbf{z}_1, \cdots, \mathbf{z}_n)^{\top} \in \mathbb{R}^{n \times p_0}$, and $\mathbf{y}^{\dagger} = (y_1 - \langle \mathbf{X}_1^*, \hat{\mathbf{A}}^{\dagger}(t_1) \rangle, \cdots, y_n - \langle \mathbf{X}_n^*, \hat{\mathbf{A}}^{\dagger}(t_n) \rangle)^{\top} \in \mathbb{R}^n$.

## 3 Variable selection and structure identification

### 3.1 Group penalty procedure

Even after tensor partition and decomposition, there are still $RS$ functions to be estimated in $\mathbf{A}(t) = (a_{rs}(t))$, $r = 1, \cdots, R$, $s = 1, \cdots, S$, in addition to $p_0$ Euclidean parameters in $\boldsymbol{\beta}$. The dimension may still be quite large relative to the available sample size. In practice not all these components are effective to the response and a sparsity structure is usually plausible. Furthermore, not all functional coefficients are varying and need to be estimated nonparamet-

rically. We further adopt a regularization procedure for variable selection and model structure identification.

To implement the shrinkage approach, we consider spline basis approximation method (De Boor, 2001) for functional estimation in this section. While the kernel method can usually provide a better local approximation to the function estimation, it is often less efficient suitable for the penalized computations. In contrast, the spline method is computationally more efficient for our purpose of regularization.

For a chosen set of B-splines basis functions $\{B_l(t), l = 1, \cdots, L\}$, we denote $\mathbf{B}(t) = (B_1(t), \cdots, B_L(t))^T$ where each $B_l(t)$ is a B-spline function, $L = k_n + k_d + 1$ is the number of basis functions, $k_n$ is the number of equispaced internal knots located in $[0, 1]$, and $k_d$ is the degree of the polynomial spline. The spline approximation of the varying coefficient can be written in terms of B-splines as

$$a_{rs}(t) \approx \sum_{l=1}^{L} \gamma_{lrs} B_l(t) = \mathbf{B}^\top(t)\boldsymbol{\gamma}_{rs}, \tag{14}$$

where $\boldsymbol{\gamma}_{rs} = (\gamma_{1rs}, \cdots, \gamma_{Lrs})^T$ are some suitable coefficients.

Here we consider B-spline basis for its desirable properties such as the numerical stability (De Boor, 2001) while other basis functions can be used for a similar purpose. We need to choose $k_n$ to control the smoothness of nonparametric function $a_{rs}(t)$'s and require $k_n \to \infty$ as $n \to \infty$. Although it is possible to allow different number of basis for different functions $a_{rs}(t)$, we specify them all equal to $L$ for the simplicity of notation.

After substituting (14) into (8), we obtain a slightly different tensor regression form:

$$y_i \approx \sum_{l=1}^{L}\sum_{r=1}^{R}\sum_{s=1}^{S} x_{irs}^* B_l(t_i)\gamma_{lrs} + \mathbf{z}_i^\top\boldsymbol{\beta} + \epsilon_i \equiv \langle \mathcal{H}_i(t_i), \mathcal{G}\rangle + \mathbf{z}_i^\top\boldsymbol{\beta} + \epsilon_i, \tag{15}$$

where $\mathcal{H}_i(t) = \mathbf{X}_i^* \circ \mathbf{B}(t) = (x_{irs}^* B_l(t)) \equiv (h_{ilrs}(t)) \in \mathbb{R}^{L \times R \times S}$, and $\mathcal{G} = (\gamma_{lrs}) \in \mathbb{R}^{L \times R \times S}$.

Before we define the penalty term in a projected B-spline space for simultaneous variable selection and structure identification, we introduce an orthogonal decomposition of $a_{rs}(t)$ with respect to the $L_2$ norm by

$$a_{rs}(t) = (a_{rs})_c + (a_{rs})_v(t) \tag{16}$$

where $(a_{rs})_c = \int_0^1 a_{rs}(t)dt \approx \int_0^1 \mathbf{B}^\top(t)dt\,\boldsymbol{\gamma}_{rs} \in \mathbb{R}$ is the mean of $a_{rs}(t)$, and $(a_{rs})_v(t) = a_{rs}(t) - (a_{rs})_c \approx (\mathbf{B}^\top(t) - \int_0^1 \mathbf{B}^\top(t)dt)\boldsymbol{\gamma}_{rs}$ is the deviation from the mean. Then we have

$$\|a_{rs}\|_2^2 = |(a_{rs})_c|^2 + \|(a_{rs})_v\|_2^2 \tag{17}$$

where $|\cdot|$ denotes the absolute value, and $\|\cdot\|_2$ is the $L_2$ norm. The two terms on the right hand side of (17) can be easily evaluated by $|(a_{rs})_c|^2 \approx \boldsymbol{\gamma}_{rs}^\top \int_0^1 \mathbf{B}(t)dt \int_0^1 \mathbf{B}^\top(t)dt\,\boldsymbol{\gamma}_{rs}$, and $\|(a_{rs})_v\|_2^2 \approx \boldsymbol{\gamma}_{rs}^\top [\int_0^1 (\mathbf{B}(t) - \int_0^1 \mathbf{B}(t)dt)(\mathbf{B}^\top(t) - \int_0^1 \mathbf{B}^\top(t)dt)dt]\boldsymbol{\gamma}_{rs} = \boldsymbol{\gamma}_{rs}^\top(\int_0^1 \mathbf{B}(t)\mathbf{B}^\top(t)dt - \int_0^1 \mathbf{B}(t)dt \int_0^1 \mathbf{B}^\top(t)dt)\boldsymbol{\gamma}_{rs}$. It is clear that $\|(a_{rs})_v\|_2 \neq 0$ when $a_{rs}$ is a varying function, $\|(a_{rs})_v\|_2 = 0$ when $a_{rs}$ is a constant function, and furthermore, $|(a_{rs})_c| = \|(a_{rs})_v\|_2 = 0$ when $a_{rs}$ is a con-

stant zero function.

We then propose the following penalized loss function to simultaneously select important variables and identify the model structure with group regularized estimator $\hat{\mathcal{G}}^{\ddagger}$ and $\hat{\boldsymbol{\beta}}^{\ddagger}$:

$$
\frac{1}{2}\sum_{i=1}^{n}\left[y_i - \langle \mathcal{H}_i(t_i), \mathcal{G}\rangle - \mathbf{z}_i^{\top}\boldsymbol{\beta}\right]^2 + n\sum_{k=1}^{p_0} P_{\omega_{\beta}}(|\beta_k|)
$$
$$
+ n\sum_{r=1}^{R}\sum_{s=1}^{S}\left\{P_{\omega_{\gamma}^c}(|(a_{rs})_c|) + P_{\omega_{\gamma}^v}(\|(a_{rs})_v\|_2)\right\} \tag{18}
$$

where $P_{\omega}(\cdot)$'s are all SCAD penalty functions with a group tuning parameter $\omega$.

To obtain the estimate $\mathcal{G}$ and $\boldsymbol{\beta}$, and further obtain $\mathbf{A}(t)$, we develop an iterative algorithm to minimizing (18), which is shown in Algorithm 1. And we describe the computational detail below.

---

**Algorithm 1** Penalized varying coefficient tensor regression

---

**Input**: penalty parameters $\omega_{\beta}$, $\omega_{\gamma}^c$, $\omega_{\gamma}^v$, number of iteration $C_{\text{iter}}$, threshold $\delta_{\text{iter}}$

1: Initialize $\mathcal{G}^{(0)}$ and $\boldsymbol{\beta}^{(0)}$ from $\mathbf{A}^{\dagger}(t)$ and $\boldsymbol{\beta}^{\dagger}$ in (11) and (13), or a least square estimation of (15). Set $c = 0, \delta_{\text{iter}}^{(0)} = 1$,

2: **while** $c < C_{\text{iter}}$ and $\delta_{\text{iter}}^{(c)} > \delta_{\text{iter}}$ **do**

3:     Set $c = c + 1$,

4:     Update $\boldsymbol{\beta}^{(c)}$ by

$$
\min_{\boldsymbol{\beta}} \frac{1}{2}\sum_{i=1}^{n}\left[y_i - \langle \mathcal{H}_i(t_i), \mathcal{G}^{(c-1)}\rangle - \mathbf{z}_i^{\top}\boldsymbol{\beta}\right]^2 + n\sum_{k=1}^{p_0} P_{\omega_{\beta}}(|\beta_k|). \tag{19}
$$

5:     Update $\mathcal{G}^{(c)}$ by

$$
\min_{\mathcal{G}} \frac{1}{2}\sum_{i=1}^{n}\left[y_i - \langle \mathcal{H}_i(t_i), \mathcal{G}\rangle - \mathbf{z}_i^{\top}\boldsymbol{\beta}^{(c)}\right]^2
$$
$$
+ n\sum_{r=1}^{R}\sum_{s=1}^{S} P_{\omega_{\gamma}^c}(|(a_{rs})_c|) + n\sum_{r=1}^{R}\sum_{s=1}^{S} P_{\omega_{\gamma}^v}(\|(a_{rs})_v\|_2). \tag{20}
$$

6:     Compute $\delta_{\text{iter}}^{(c)} = \|\boldsymbol{\beta}^{(c)} - \boldsymbol{\beta}^{(c-1)}\|_2^2 + \|\mathcal{G}^{(c)} - \mathcal{G}^{(c-1)}\|_2^2$.

7: **end while**

8: Set $\hat{\mathcal{G}}^{\ddagger} = \mathcal{G}^{(c)}$, $\hat{\boldsymbol{\beta}}^{\ddagger} = \boldsymbol{\beta}^{(c)}$. And compute $\hat{\mathbf{A}}^{\ddagger}(t) = (\hat{a}_{rs}^{\ddagger}(t)) = (\mathbf{B}^{\top}(t)\hat{\boldsymbol{\gamma}}_{rs}^{\ddagger})$.

---

In general the SCAD function is defined through its first derivative as

$$
\dot{P}_{\omega}(\theta) = \omega\left[I\{|\theta| \leq \omega\} + \frac{(a_0\omega - |\theta|)_+}{(a_0 - 1)\omega}I\{|\theta| > \omega\}\right] \tag{21}
$$

with $a_0 > 2$, where $I\{\cdot\}$ is an indicator function. In our numerical studies, we set $a_0 = 3.7$ as suggested by Fan and Li (2001).

Following the iterative local quadratic approximation algorithm in Fan and Li (2001), given $\theta^{(c-1)}$ as the estimator of $\theta$ in $(c-1)$-th iteration, with the Taylor expansion, we can approximate

the regularization term $P_\omega(\theta)$ by

$$P_\omega(\theta) \approx P_\omega(\theta^{(c-1)}) + \frac{1}{2}\frac{\dot{P}_\omega(\theta^{(c-1)})}{\theta^{(c-1)}}(\theta^2 - (\theta^{(c-1)})^2). \tag{22}$$

For (19), $\mathcal{G}^{(c-1)}$ is the estimate of $\mathcal{G}$ in the $(c-1)$-th iteration. Its approximate solution is

$$\boldsymbol{\beta}^{(c)} = \left(\mathbf{Z}^\top\mathbf{Z} + n\boldsymbol{\Omega}_0(\boldsymbol{\beta}^{(c-1)},\omega_\beta)\right)^{-1}\mathbf{Z}^\top\mathbf{y}_{\mathbf{Z}}^{(c-1)} \tag{23}$$

where $\mathbf{y}_{\mathbf{Z}}^{(c-1)} = (y_1 - \langle\mathcal{H}_1(t_1),\mathcal{G}^{(c-1)}\rangle,\cdots,y_n - \langle\mathcal{H}_n(t_n),\mathcal{G}^{(c-1)}\rangle)^\top \in \mathbb{R}^n$, $\mathbf{Z} = (\mathbf{z}_1,\cdots,\mathbf{z}_n)^\top \in \mathbb{R}^{n\times p_0}$, $\boldsymbol{\Omega}_0(\boldsymbol{\beta}^{(c-1)},\omega_\beta) = \mathrm{diag}\left(\frac{\dot{P}_{\omega_\beta}(|\beta_1^{(c-1)}|)}{|\beta_1^{(c-1)}|},\cdots,\frac{\dot{P}_{\omega_\beta}(|\beta_{p_0}^{(c-1)}|)}{|\beta_{p_0}^{(c-1)}|}\right) \in \mathbb{R}^{p_0\times p_0}$ is a diagonal matrix, and $\boldsymbol{\beta}^{(c-1)} = (\beta_1^{(c-1)},\cdots,\beta_{p_0}^{(c-1)}) \in \mathbb{R}^{p_0}$ is the estimate of $\boldsymbol{\beta}$ in the $(c-1)$-th iteration.

Then, for (20), given the current estimate $\boldsymbol{\beta}^{(c)}$, $\mathcal{G}^{(c)}$ can be obtained through

$$\mathrm{vec}(\mathcal{G}^{(c)}) = \left(\mathbf{H}^\top\mathbf{H} + n\boldsymbol{\Omega}_1(\mathcal{G}^{(c-1)},\omega_\gamma^c) + n\boldsymbol{\Omega}_2(\mathcal{G}^{(c-1)},\omega_\gamma^v)\right)^{-1}\mathbf{H}^\top\mathbf{y}_{\mathbf{H}}^{(c)} \tag{24}$$

where $\mathbf{y}_{\mathbf{H}}^{(c)} = (y_1 - \mathbf{z}_1^\top\boldsymbol{\beta}^{(c)},\cdots,y_n - \mathbf{z}_n^\top\boldsymbol{\beta}^{(c)})^\top \in \mathbb{R}^n$, $\mathbf{H} = (\mathrm{vec}(\mathcal{H}_1(t_1),\cdots,\mathrm{vec}(\mathcal{H}_n(t_n))^\top \in \mathbb{R}^{n\times LRS}$, and $\mathrm{vec}(\mathcal{H}_i(t_i)) = (h_{ilrs}(t_i)) = (h_{i111}(t_i),\cdots,h_{iL11}(t_i),\cdots,h_{iLR1}(t_i),\cdots,h_{iLRS}(t_i))^\top \in \mathbb{R}^{LRS}$. As for two penalized terms, we have

$$\boldsymbol{\Omega}_1(\mathcal{G}^{(c-1)},\omega_\gamma^c) = \mathrm{diag}(\frac{\dot{P}_{\omega_\gamma^c}(|(a_{11}^{(c-1)})_c|)}{|(a_{11}^{(c-1)})_c|}\mathcal{B}_1,\cdots,\frac{\dot{P}_{\omega_\gamma^c}(|(a_{R1}^{(c-1)})_c|)}{|(a_{R1}^{(c-1)})_c|}\mathcal{B}_1,\cdots,\frac{\dot{P}_{\omega_\gamma^c}(|(a_{RS}^{(c-1)})_c|)}{|(a_{RS}^{(c-1)})_c|}\mathcal{B}_1)$$

is an $LRS$-by-$LRS$ block diagonal matrix, where $|(a_{rs}^{(c-1)})_c| = \sqrt{\boldsymbol{\gamma}_{rs}^{(c-1)\top}\mathcal{B}_1\boldsymbol{\gamma}_{rs}^{(c-1)}}$ with $\boldsymbol{\gamma}_{rs} = (\gamma_{lrs}) \in \mathbb{R}^L$, and $\mathcal{B}_1 \in \mathbb{R}^{L\times L}$ is a matrix given by

$$\mathcal{B}_1 = \int_0^1 \mathbf{B}(t)dt\int_0^1\mathbf{B}^\top(t)dt = \begin{bmatrix} \left(\int_0^1 B_1(t)dt\right)^2 & \cdots & \int_0^1 B_1(t)dt\int_0^1 B_L(t)dt \\ \vdots & \ddots & \vdots \\ \int_0^1 B_L(t)dt\int_0^1 B_1(t)dt & \cdots & \left(\int_0^1 B_L(t)dt\right)^2 \end{bmatrix},$$

and

$$\boldsymbol{\Omega}_2(\mathcal{G}^{(c-1)},\omega_\gamma^v) = \mathrm{diag}(\frac{\dot{P}_{\omega_\gamma^v}(\|(a_{11}^{(c-1)})_v\|_2)}{\|(a_{11}^{(c-1)})_v\|_2}\mathcal{B}_2,\cdots,\frac{\dot{P}_{\omega_\gamma^v}(\|(a_{R1}^{(c-1)})_v\|_2)}{\|(a_{R1}^{(c-1)})_v\|_2}\mathcal{B}_2,\cdots,\frac{\dot{P}_{\omega_\gamma^v}(\|(a_{RS}^{(c-1)})_v\|_2)}{\|(a_{RS}^{(c-1)})_v\|_2}\mathcal{B}_2)$$

is also an $LRS$-by-$LRS$ block diagonal matrix, where $\|(a_{rs}^{(c-1)})_v\|_2 = \sqrt{\boldsymbol{\gamma}_{rs}^{(c-1)\top}\mathcal{B}_2\boldsymbol{\gamma}_{rs}^{(c-1)}}$ and $\mathcal{B}_2 \in \mathbb{R}^{L\times L}$ is a matrix given by

$$\mathcal{B}_2 = \int_0^1 \mathbf{B}(t)\mathbf{B}^\top(t)dt - \int_0^1 \mathbf{B}(t)dt\int_0^1\mathbf{B}^\top(t)dt$$

$$= \begin{bmatrix} \int_0^1 B_1^2(t)dt - \left(\int_0^1 B_1(t)dt\right)^2 & \cdots & \int_0^1 B_1(t)B_L(t)dt - \int_0^1 B_1(t)dt\int_0^1 B_L(t)dt \\ \vdots & \ddots & \vdots \\ \int_0^1 B_L(t)B_1(t)dt - \int_0^1 B_L(t)dt\int_0^1 B_1(t)dt & \cdots & \int_0^1 B_L^2(t)dt - \left(\int_0^1 B_L(t)dt\right)^2 \end{bmatrix}.$$

The vector $\text{vec}(\mathcal{G}^{(c)})$ can be easily tensorized to produce the corresponding $\mathcal{G}^{(c)}$.

We iteratively update $\boldsymbol{\beta}$ and $\mathcal{G}$ until convergence, yielding the final estimates $\hat{\mathcal{G}}^{\ddagger} = \mathcal{G}^{(c)}$ and $\hat{\boldsymbol{\beta}}^{\ddagger} = \boldsymbol{\beta}^{(c)}$. We use $\mathbf{A}^{\ddagger}(t)$ to denote the final refined penalized estimator derived from our iterative procedure, Specifically, we have $\hat{\mathbf{A}}^{\ddagger}(t) = (\hat{a}_{rs}^{\ddagger}(t)) = (\mathbf{B}^{\top}(t)\hat{\boldsymbol{\gamma}}_{rs}^{\ddagger})$.

**Remark 1** Instead of shrinking $|(a_{rs})_c|$ and $\|(a_{rs})_v\|_2$ to zero, we could also operationally replace them by $|(\boldsymbol{\gamma}_{rs})_c|$ and $\|(\boldsymbol{\gamma}_{rs})_v\|_2$ where $(\boldsymbol{\gamma}_{rs})_c = \frac{1}{L}\sum_{l=1}^{L}\gamma_{lrs}$ is the mean and $(\boldsymbol{\gamma}_{rs})_v = \boldsymbol{\gamma}_{rs} - (\boldsymbol{\gamma}_{rs})_c \in \mathbb{R}^L$ represents the deviation function. These two pairs are quite similar in both theory and practice since we can choose the order of spline to ensure a one-to-one corresponding projection. More specifically, we can expand $|(\boldsymbol{\gamma}_{rs})_c|^2 = \frac{1}{L^2}\boldsymbol{\gamma}_{rs}^{\top}\mathbf{1}\mathbf{1}^{\top}\boldsymbol{\gamma}_{rs}$, and $\|(\boldsymbol{\gamma}_{rs})_v\|_2^2 = \boldsymbol{\gamma}_{rs}^{\top}(\mathbf{I} - \frac{1}{L}\mathbf{1}\mathbf{1}^{\top})(\mathbf{I} - \frac{1}{L}\mathbf{1}\mathbf{1}^{\top})^{\top}\boldsymbol{\gamma}_{rs} = \boldsymbol{\gamma}_{rs}^{\top}(\mathbf{I} - \frac{1}{L}\mathbf{1}\mathbf{1}^{\top})\boldsymbol{\gamma}_{rs}$, where $\mathbf{1} \in \mathbb{R}^L$ is a vector of ones and $\mathbf{I} \in \mathbb{R}^{L \times L}$ is an identity matrix.

**Remark 2** To identify constant coefficients in $\mathbf{A}(t)$, we can also use the term $\|\dot{a}_{rs}\|_2 = \sqrt{\boldsymbol{\gamma}_{rs}^{\top}\dot{\mathcal{B}}\boldsymbol{\gamma}_{rs}}$ to replace $\|(a_{rs})_v\|_2$, where $\dot{\mathcal{B}} = \{\int_0^1 \dot{\mathbf{B}}_l(t)^{\top}\dot{\mathbf{B}}_{l'}(t)dt : l,l' = 1,\cdots,L\} \in \mathbb{R}^{L \times L}$. The idea is to shrink the first derivative $\|\dot{a}_{rs}\|_2$ to zero, then $a_{rs}(t)$ is a constant function. This penalty term will produce a similar result as $\|(a_{rs})_v\|_2$ (Guo and Li, 2022).

**Remark 3** In numerical studies, we have also used Lasso (Tibshirani, 1996) and MCP (Zhang, 2010) penalty functions in place of $P_{\omega}(\cdot)$ to show the versatility of our methodology. For Lasso penalty, $\dot{P}_{\omega}(\theta) = \omega\text{sign}(\theta)$ and for MCP penalty, $\dot{P}_{\omega}(\theta) = (\omega - \frac{|\theta|}{a_0'})_{+}\text{sign}(\theta)$ where $a_0' = 3$, which is also adopted in Breheny and Huang (2015).

## 3.2 Tuning penalty parameters and model refinement

After obtaining $\hat{\mathbf{A}}^{\ddagger}(t)$ and $\hat{\boldsymbol{\beta}}^{\ddagger}$, we can select variables (whether $\beta_k$ can be shrunk to zero), and identify constant coefficients (whether $a_{rs}(t)$ is a zero constant function, non-zero constant function or varying function) simultaneously. To this end we need tune penalty parameters $\omega_{\beta}$, $\omega_{\gamma}^c$ and $\omega_{\gamma}^v$ to achieve accurate results. We thus choose the optimal tuning parameters $\omega_{\beta}^*$, $\omega_{\gamma}^{c,*}$ and $\omega_{\gamma}^{v,*}$ by a data-driven grid search method minimizing the Bayesian information criterion (BIC). Other criteria such as extended Bayesian information criterion (eBIC, Chen and Chen 2008) and generalised information criterion (GIC, Fan and Tang 2013) can also be adopted. More specifically, we select the best tuning parameters by minimizing

$$\text{BIC}(\omega_{\beta}, \omega_{\gamma}^c, \omega_{\gamma}^v) = \log\left(\frac{\sum_{i=1}^{n}[y_i - \langle \mathcal{H}_i(t_i), \hat{\mathcal{G}}^{\ddagger}\rangle - \mathbf{z}_i^{\top}\hat{\boldsymbol{\beta}}^{\ddagger}]^2}{n}\right) + (d_{\beta} + d_c + d_v L)\frac{\log(n)}{n} \qquad (25)$$

where $\hat{\mathcal{G}}^{\ddagger}$ and $\hat{\boldsymbol{\beta}}^{\ddagger}$ are the solution of (18) given $\omega_{\beta}$, $\omega_{\gamma}^c$ and $\omega_{\gamma}^v$, $d_c$ and $d_v$ are the numbers of functions estimated as nonzero constant functions and varying functions respectively, and $d_{\beta}$ is the number of non-zero coefficients in $\hat{\boldsymbol{\beta}}^{\ddagger}$. $L$ is equivalent number of constant coefficients for a varying function, and exactly equal to the number of spline basis in (18).

## 4 Asymptotic theory

We first introduce some notations. Denote the random covariates to be $\{t, \mathbf{X}^*, \mathbf{z}\}$ and its sample version as $\{t_i, \mathbf{X}_i^*, \mathbf{z}_i\}$. Let $\mathbf{A}^*(t) = (a_{rs}^*(t))$, and $\boldsymbol{\beta}^* = (\beta_k^*)$ be the ground true value of $\mathbf{A}(t)$ and $\boldsymbol{\beta}$.

Denote $\mu_r = \int t^r K(t)dt$ and $\nu_r = \int t^r K^2(t)dt$ for any integer $r$. For any smooth functions $f(t)$, denote $\dot{f}(t) = df(t)/dt$, and $\ddot{f}(t) = d^2 f(t)/dt^2$. Moreover, we do not distinguish the differentiation and continuation at the boundary points from those in the interior of $[0,1]$. For instance, a continuous function at the boundary of $[0,1]$ means that this function is left continuous at 0 and right continuous at 1.

The following technical conditions are assumed for the theoretical results.

(C1) The index variable $\{t_i\}$ are i.i.d. with a density function $f(t)$, which is twice continuously differentiable and is bounded away from 0 in the support $t \in [0,1]$. Further $t_i$ is independent to the covariates $\mathbf{X}_i^*$ and $\mathbf{z}_i$.

(C2) The kernel function $K(t)$ is a symmetric (i.e., $K(-t) = K(t)$) and continuous density function and there exists a constant $s > 2$ such that $\int_{-1}^{1} K(u)^s u^j du < \infty$ for $j \leq 6$. In addition, there exist a large enough constant $K_0$ such that $\sup_{t \in \mathbb{R}} K(t) \leq K_0$ and $\sup_{t \in \mathbb{R}} K'(t) \leq K_0$.

(C3) The elements in $\mathbf{A}^*(\cdot)$ have continuous third derivatives in $[0,1]$. The parameters $\boldsymbol{\beta}$ fall in a compact space in $\mathbb{R}^{p_0}$.

(C4) Assume the vector $\tilde{\mathbf{v}}_i(t)$, $i = 1, \cdots, n$ are i.i.d. with conditional covariance matrix $\boldsymbol{\Omega}(t) = \mathrm{Cov}(\tilde{\mathbf{v}}_i | T = t)$. Assume that the entries in $\boldsymbol{\Omega}(t)$ are Lipschitz continuous in $t \in [0,1]$, and there exists a $\lambda > 1$ such that $\lambda^{-1} \leq \inf_{t \in [0,1]} \lambda_{\min}(\boldsymbol{\Omega}(t)) \leq \sup_{t \in [0,1]} \lambda_{\max}(\boldsymbol{\Omega}(t)) \leq \lambda$. Here $\lambda_{\min}(\boldsymbol{\Omega}(t))$ and $\lambda_{\max}(\boldsymbol{\Omega}(t))$ are the smallest and largest eigenvalue of $\boldsymbol{\Omega}(t)$ respectively. Moreover, denote $\boldsymbol{\Omega}_{\mathbf{z},\mathbf{v}} = \mathrm{E}((\mathbf{z}^\top, \mathbf{v}^\top)^\top (\mathbf{z}^\top, \mathbf{v}^\top))$, $\Omega_{\mathbf{v}} = \mathrm{E}(\mathbf{v}^\top \mathbf{v})$, $\Omega_{\mathbf{z}} = \mathrm{E}(\mathbf{z}^\top \mathbf{z})$, $\Omega_{\mathbf{vz}} = \mathrm{E}(\mathbf{v}^\top \mathbf{z})$, $\Omega_{\mathbf{zv}} = \mathrm{E}(\mathbf{z}^\top \mathbf{v})$. We assume that $\tilde{\mathbf{v}}_i(t)$ are uniformly bounded and $\boldsymbol{\Omega}_{\mathbf{z},\mathbf{v}}$ is invertible with eigenvalues bounded away from zero and infinity.

(C5) For simplicity we assume that the entries of the tensor covariate $\mathbf{X}^*$ and the one-way covariate $\mathbf{z}$ are centered (i.e., with mean $\mathbf{0}$) and bounded random variables.

(C6) Let $c_1, \cdots, c_{k_n}$ be the interior knots of $[0,1]$. Furthermore, let $c_0 = 0, c_{k_n+1} = 1, \kappa_i = c_i - c_{i-1}$ such that $\max_{1 \leq i \leq k_n+1}\{\kappa_i\} = O(\min_{1 \leq i \leq k_n+1}\{\kappa_i\})$ and $\max_{1 \leq i \leq k_n}\{\kappa_{i+1} - \kappa_i\} = o(k_n^{-1})$.

## 4.1 Asymptotic properties for regression estimates

**Theorem 1** *Suppose that Assumptions (C1)-(C4) hold, if $n \to \infty$ and $h \to 0$ in such a way that $nh \to \infty$ and $nh^5 = O(1)$, then the asymptotic conditional distribution of $\hat{\boldsymbol{\theta}} = (\hat{\boldsymbol{\beta}}^\top, \mathrm{vec}(\hat{\mathbf{A}}(t))^\top, \mathrm{vec}(\dot{\hat{\mathbf{A}}}(t))^\top)^\top$ in (10) is given by*

$$\sqrt{nh} \left\{ \begin{bmatrix} \hat{\boldsymbol{\beta}} - \boldsymbol{\beta}^* \\ \mathrm{vec}(\hat{\mathbf{A}}(t)) - \mathrm{vec}(\mathbf{A}^*(t)) \\ \mathrm{vec}(\dot{\hat{\mathbf{A}}}(t)) - \mathrm{vec}(\dot{\mathbf{A}}^*(t)) \end{bmatrix} - \frac{1}{2} \begin{bmatrix} \mathbf{0}_{p_0} \\ \mu_2 h^2 \mathrm{vec}(\ddot{\mathbf{A}}^*(t)) \\ -h^2 f(t)^{-1}\dot{f}(t)(\mu_2^{-1}\mu_4 - \mu_2)\mathrm{vec}(\ddot{\mathbf{A}}^*(t)) \end{bmatrix} \right\}$$

$$\xrightarrow{d} N\left( \mathbf{0}, \sigma^2 \begin{bmatrix} \nu_0 \boldsymbol{\Omega}_{\mathbf{z},\mathbf{v}}^{-1} f(t)^{-1} & -\nu_0 f(t)^{-2}\dot{f}(t)\boldsymbol{\Omega}_{\mathbf{z},\mathbf{v}}^{-1}\begin{pmatrix}\Omega_{\mathbf{zv}} \\ \Omega_{\mathbf{v}}\end{pmatrix}\Omega_{\mathbf{v}}^{-1} \\ -\nu_0 f(t)^{-2}\dot{f}(t)\Omega_{\mathbf{v}}^{-1}(\Omega_{\mathbf{vz}}, \Omega_{\mathbf{v}})\boldsymbol{\Omega}_{\mathbf{z},\mathbf{v}}^{-1} & h^{-2}f(t)^{-1}\mu_2^{-1}\nu_2\Omega_{\mathbf{v}}^{-1} \end{bmatrix} \right). \tag{26}$$

In particular, let $\tilde{\Omega}_{\mathbf{v}}(t) = \Omega_{\mathbf{v}} - \Omega_{\mathbf{vz}}\Omega_{\mathbf{z}}^{-1}\Omega_{\mathbf{zv}}$, the estimated $\mathrm{vec}(\hat{\mathbf{A}}^{\dagger}(t))$ in $\hat{\boldsymbol{\theta}}$ has asymptotic conditional distribution of

$$\sqrt{nh}\Big[\big[\mathrm{vec}(\hat{\mathbf{A}}^{\dagger}(t)) - \mathrm{vec}(\mathbf{A}^{*}(t))\big] - \frac{1}{2}\mu_2 h^2 \mathrm{vec}(\ddot{\mathbf{A}}^{*}(t))\Big] \xrightarrow{d} N\Big(0, \frac{\nu_0\sigma^2}{f(t)}\tilde{\Omega}_{\mathbf{v}}^{-1}\Big). \tag{27}$$

**Theorem 2** *Let $\hat{\boldsymbol{\beta}}^{\dagger}$ be the estimate that minimizes* (12). *Suppose that Assumptions (C1)-(C5) hold, if $n \to \infty$ and $h \to 0$ in such a way that $nh^4 \to 0$ and $nh^3/\log(n) \to \infty$, we have*

$$\sqrt{n}(\hat{\boldsymbol{\beta}}^{\dagger} - \boldsymbol{\beta}^{*}) \xrightarrow{d} N(0, \sigma^2\Omega_{\mathbf{z}}^{-1}) \tag{28}$$

The optimal bandwidth in Theorem 1 is $h \sim n^{-1/5}$. This bandwidth does not satisfy the condition in Theorem 2. Interestingly, in order to obtain the $\sqrt{n}$ consistency and asymptotic normality for $\hat{\boldsymbol{\beta}}^{\dagger}$, undersmoothing for $\hat{\mathbf{A}}^{\dagger}(t)$ is necessary for controlling the bias of the local smoothing estimators. On the other hand, $h$ should also be large enough such that $nh^3/\log(n) \to \infty$, in order to control the variance of the remainder terms. This is a common requirement in varying coefficient partial linear models; see Carroll et al. (1997) for a detailed discussion.

## 4.2 Asymptotic properties for penalized functions and parameters

According to the result in De Boor (2001), for $a_{rs}(t)$ satisfying Assumption (C3) and (C6), there exists a B-spline function $\mathbf{B}^{\top}(t)\boldsymbol{\gamma}_{rs}^{*}$, where $\boldsymbol{\gamma}_{rs}^{*}$ is the best spline approximating function for $a_{rs}^{*}(t)$, such that

$$R_{rs} \equiv \sup_{t\in[0,1]} |\mathbf{B}^{\top}(t)\boldsymbol{\gamma}_{rs}^{*} - a_{rs}^{*}(t)| = O(k_n^{-2}) \tag{29}$$

where $k_n$ is number of interior knots, $k_n \to \infty$ as $n \to \infty$.

Note in Eq. (18), we actually optimize it through $\mathcal{G}$ and $\boldsymbol{\beta}$. With Assumption (C3) and (C6) hold, we may approximate $|(a_{rs})_c| \approx \sqrt{\boldsymbol{\gamma}_{rs}^{\top}\mathcal{B}_1\boldsymbol{\gamma}_{rs}} = \|\boldsymbol{\gamma}_{rs}\|_{\mathcal{B}_1}$ and $\|(a_{rs})_v\|_2 \approx \sqrt{\boldsymbol{\gamma}_{rs}^{\top}\mathcal{B}_2\boldsymbol{\gamma}_{rs}} = \|\boldsymbol{\gamma}_{rs}\|_{\mathcal{B}_2}$ respectively. Now we rewrite Eq. (18) as

$$L(\mathcal{G}, \boldsymbol{\beta}) = \sum_{i=1}^{n} \big[y_i - \langle\mathcal{H}_i(t_i), \mathcal{G}\rangle - \mathbf{z}_i^{\top}\boldsymbol{\beta}\big]^2 + n\sum_{k=1}^{p_0} P_{\omega_{\beta}}(|\beta_k|)$$
$$+ n\sum_{r=1}^{R}\sum_{s=1}^{S}\big\{P_{\omega_{\gamma}^{c}}(\|\boldsymbol{\gamma}_{rs}\|_{\mathcal{B}_1}) + P_{\omega_{\gamma}^{v}}(\|\boldsymbol{\gamma}_{rs}\|_{\mathcal{B}_2})\big\} \tag{30}$$

Similar with local linear smoothing method in section 2.3, we denote $\mathbf{u}_i = \mathrm{vec}(\mathcal{H}_i(t_i)) = (h_{ilrs}(t_i)) \in \mathbb{R}^{LRS}$, $\tilde{\mathbf{u}}_i = (\mathrm{vec}(\mathcal{H}_i(t_i))^{\top}, \mathbf{z}_i^{\top})^{\top} \in \mathbb{R}^{LRS+p_0}$. Let $\omega_{\max} = \max\{\omega_{\beta}, \omega_{\gamma}^{c}, \omega_{\gamma}^{v}\}$, and $\omega_{\min} = \min\{\omega_{\beta}, \omega_{\gamma}^{c}, \omega_{\gamma}^{v}\}$.

Recall that the number of basis functions $L = k_n + k_d + 1$. As $k_d$ is usually chosen to be very small, we shall assume that $L$ is having the same order as $k_n$. In addition, we assume that

(C7) $p_0 = O(n^{\tau})$ for some constant $\tau < 1$, and $k_n = C_k \min\left\{\left(\frac{n}{\log n}\right)^{1/5}, \left(\frac{n^{1-\tau}}{\log n}\right)^{1/4}\right\}$ for some constant $C_k > 0$.

We also impose the following assumptions for the tuning parameters and the minimum signal

strength.

(C8) There exists a large enough constant $a > 2$ such that $|\beta_k^*| > a\omega_\beta$ for $k \in \{k : \beta_k^* \neq 0\}$, $\|\gamma_{ij}^*\|_{\mathcal{B}_1} \geq a\omega_\gamma^c$ for $(i,j) \in \{(r,s) : \|\gamma_{rs}^*\|_{\mathcal{B}_1} \neq 0\}$, and $\|\gamma_{rs}^*\|_{\mathcal{B}_2} \geq a\omega_\gamma^v$ for $(i,j) \in \{(r,s) : \|\gamma_{ij}^*\|_{\mathcal{B}_2} \neq 0\}$.

**Theorem 3** *Suppose that Conditions (C1), (C3), (C5), (C6) and (C7) hold, and set $\omega_\beta = c_1 \max\left\{\left(\frac{\log n}{n}\right)^{2/5}, \left(\frac{\log n}{n^{1-\tau}}\right)^{1/2}\right\}$, $\omega_\gamma^v = c_2 \max\left\{\left(\frac{\log n}{n}\right)^{1/2}, \left(\frac{\log n}{n^{1-\tau}}\right)^{5/8}\right\}$ and $\omega_\gamma^c = c_3 \max\left\{\left(\frac{\log n}{n}\right)^{1/2}, \left(\frac{\log n}{n^{1-\tau}}\right)^{5/8}\right\}$ for some large enough constants $c_1, c_2, c_3 > 0$. We have:*

*(i) $\|\hat{\beta}_k^\ddagger - \beta_k^*\| = O_p\left(\max\left\{\left(\frac{\log n}{n}\right)^{2/5}, \left(\frac{\log n}{n^{1-\tau}}\right)^{1/2}\right\}\right)$, for $k = 1, \cdots p_0$,*

*(ii) $\|\hat{a}_{rs}^\ddagger(t) - a_{rs}^*(t)\| = O_p\left(\max\left\{\left(\frac{\log n}{n}\right)^{2/5}, \left(\frac{\log n}{n^{1-\tau}}\right)^{1/2}\right\}\right)$, for $r = 1, \cdots, R$, and $s = 1, \cdots, S$.*

**Theorem 4** *Suppose that Conditions (C1), (C3), (C5), (C6), (C7) and (C8) hold, and suppose $\omega_\beta, \omega_\gamma^c, \omega_\gamma^v \to 0$, and $\min\left\{\left(\frac{n}{\log n}\right)^{2/5}, \left(\frac{n^{1-\tau}}{\log n}\right)^{1/2}\right\}\omega_\beta \to \infty$, $\min\left\{\left(\frac{n}{\log n}\right)^{1/2}, \left(\frac{n^{1-\tau}}{\log n}\right)^{5/8}\right\}\omega_\gamma^v \to \infty$ and $\min\left\{\left(\frac{n}{\log n}\right)^{1/2}, \left(\frac{n^{1-\tau}}{\log n}\right)^{5/8}\right\}\omega_\gamma^c \to \infty$, we have, with probability tending to 1,*

*(i) $\hat{\beta}_k^\ddagger = 0$, for $k \in \{k, |\beta_k| = 0\}$,*

*(ii) $(\hat{a}_{rs}^\ddagger)_c = 0$, for $(r,s) \in \{(r,s) : |(a_{rs})_c| = 0\}$,*

*(iii) $(\hat{a}_{rs}^\ddagger)_v(t) \equiv 0$, for $(r,s) \in \{(r,s) : \|(a_{rs})_v\|_2 = 0\}$.*

In Theorem 3, the convergence rate is bounded by $O_p\left(\max\left\{\left(\frac{\log n}{n}\right)^{2/5}, \left(\frac{\log n}{n^{1-\tau}}\right)^{1/2}\right\}\right)$. This rate elegantly balances two components: 1. The term $\left(\frac{\log n}{n}\right)^{2/5}$ matches the optimal nonparametric rate for estimating smooth functions using spline approximations (Stone, 1982). 2. The term $\left(\frac{\log n}{n^{1-\tau}}\right)^{1/2}$ corresponds to the parametric rate in high-dimensional settings with a diverging number of parameters (where $p_0 = O(n^\tau)$). This mirrors the optimal minimax rates established in classical high-dimensional partially linear models and varying coefficient models (Fan and Li, 2001; Cheng et al., 2014).

## 5 Simulation study

### 5.1 Estimation consistency

The data were generated according to (8), where $t_i$, $i = 1, \cdots, n$, are randomly sampled from the uniform distribution $U(0,1)$. We first consider the following two settings:

Case I : Each subject $\mathcal{X}_i$ is a $p_1 \times p_2$ tensor with order $D = 2$, $p_1 = p_2 = 80$ for $i = 1, \cdots, n$. Evenly divide $\mathcal{X}_i$ into $S = 16$ parts with $p_1' = p_2' = 20$, by partitioning it into $S^{(1)} = S^{(2)} = 4$ equal segments along length and width separately. Set $R = 10$. For each part $\tilde{\mathcal{X}}^{(s)}$ expressed in the form of (6), $\lambda_r^{(s)} = 1$, $\mathbf{x}_{rs}^* \in \mathbb{R}^n$ is a Gaussian ensemble vector whose entries are generated i.i.d. from the standard normal distribution, and any $\mathbf{x}_r^{(sd)}, \mathbf{x}_{r'}^{(sd)} \in \mathbb{R}^{p_d'}$, $r \neq r'$ are orthonormal and obtained from the QR decomposition of a random matrix. The tensor function $\mathcal{A}(t)$ has the same dimension of $\mathcal{X}_i$, and for each part, corresponding $\mathcal{A}^{(s)}(t) = \sum_{r=1}^R a_{rs}(t)\mathbf{x}_r^{(s1)} \circ \mathbf{x}_r^{(s2)}$.

Case II : Each subject $\mathcal{X}_i$ is a $p_1 \times p_2 \times p_3$ tensor with order $D = 3$, $p_1 = p_2 = p_3 = 40$ for $i = 1, \cdots, n$. Evenly divide $\mathcal{X}_i$ into $S = 64$ parts by partitioning it into $S^{(1)} = S^{(2)} = S^{(3)} = 4$

equal segments along length, width, and height separately, with $p'_1 = p'_2 = p'_3 = 10$. Set $R = 5$. For each $\tilde{\mathcal{X}}^{(s)}$ in (6), $\lambda_r^{(s)} = 1$, and $\mathbf{x}_{rs}^* \in \mathbb{R}^n, \mathbf{x}_r^{(sd)} \in \mathbb{R}^{p'_d}$ are generated in the same way as Case I. Tensor function $\mathcal{A}(t)$ has the same dimension of $\mathcal{X}_i$, and for each part, corresponding $\mathcal{A}^{(s)}(t) = \sum_{r=1}^R a_{rs}(t)\mathbf{x}_r^{(s1)} \circ \mathbf{x}_r^{(s2)} \circ \mathbf{x}_r^{(s3)}$.

In both cases, we consider four different functions for $a_{rs}(t)$: $a_{1rs}(t) = 4\sqrt{rs/RS}(t-0.5)^2$, $a_{2rs}(t) = \sqrt{rs/RS}t^{0.5}$, $a_{3rs}(t) = 1.75\sqrt{rs/RS}(\exp-(3t-1)^2 + \exp-(4t-3)^2 - 0.75)$, $a_{4rs}(t) = \sqrt{rs/RS}\sin(2\pi(t-0.5))$, $1 \leq r \leq R; 1 \leq s \leq S$. We set $p_0 = 2$ and $\boldsymbol{\beta} = (3,3)^\top$. $\mathbf{Z} \in \mathbb{R}^{n \times p_0}$ is a Gaussian ensemble matrix. The random error $\epsilon_i$ follows the standard normal distribution $N(0,1)$.

For each simulated data, we carry out the estimation procedure introduced in Section 2 to fit the model. After 500 simulations, we summarize the performance of our varying coefficient tensor regression (VCTR) model in Table 1 with sample size $n = 2000$ and 5000. The estimation errors for functions are integrated over the range of $t$. The errors for $\mathbf{A}(t)$ and $\boldsymbol{\beta}$ are averaged across all their components. These empirical results affirm the estimation consistency of the semi-parametric estimators for regression coefficients.

We also show the prediction errors of Case I and II with 10-fold cross validation in Figure 2. We compare prediction errors of our VCTR model with two constant coefficient tensor models, based on Miranda et al. (2018) and Zhou et al. (2013), respectively. When the true model involves varying coefficients, our model clearly demonstrates an advantage over those existing models.

Table 1: Estimation results of Case I and II for 500 simulations. The accuracy of $\hat{\mathbf{A}}^\dagger(t)$ is measured by MIAE and RMISE, where MIAE $= \mathrm{E}\big[\int_0^1 |\hat{\mathbf{A}}^\dagger(t) - \mathbf{A}(t)|dt\big]$, and RMISE $= \mathrm{E}^{1/2}\big[\int_0^1 (\hat{\mathbf{A}}^\dagger(t) - \mathbf{A}(t))^2 dt\big]$. The accuracy of $\hat{\boldsymbol{\beta}}^\dagger$ is measured by MAE and MSE, where MAE $= \frac{1}{p_0}\sum_{j=1}^{p_0} |\hat{\beta}_j^\dagger - \beta_j|$, and RMSE $= \big[\frac{1}{p_0}\sum_{j=1}^{p_0}(\hat{\beta}_j^\dagger - \beta_j)^2\big]^{1/2}$.

| Case | Function | n | $\hat{\mathbf{A}}^\dagger(t)$ | | $\hat{\boldsymbol{\beta}}^\dagger$ | |
| | | | MIAE | RMISE | MAE | RMSE |
|------|----------|---|------|-------|-----|------|
| I | $\mathbf{A}_1(t)$ | 2000 | 0.0622(0.0011) | 0.0780(0.0012) | 0.0343(0.0263) | 0.0371(0.0267) |
| | | 5000 | 0.0272(0.0011) | 0.0341(0.0013) | 0.0200(0.0144) | 0.0218(0.0145) |
| | $\mathbf{A}_2(t)$ | 2000 | 0.0636(0.0023) | 0.0799(0.0029) | 0.0455(0.0274) | 0.0518(0.0302) |
| | | 5000 | 0.0279(0.0007) | 0.0349(0.0009) | 0.0270(0.0137) | 0.0314(0.0158) |
| | $\mathbf{A}_3(t)$ | 2000 | 0.0648(0.0016) | 0.0817(0.0021) | 0.0441(0.0301) | 0.0485(0.0337) |
| | | 5000 | 0.0398(0.0006) | 0.0392(0.0008) | 0.0259(0.0168) | 0.0394(0.0154) |
| | $\mathbf{A}_4(t)$ | 2000 | 0.0678(0.0018) | 0.0852(0.0021) | 0.0340(0.0181) | 0.0364(0.0179) |
| | | 5000 | 0.0358(0.0016) | 0.0448(0.0019) | 0.0152(0.0086) | 0.0165(0.0084) |
| II | $\mathbf{A}_1(t)$ | 2000 | 0.0687(0.0031) | 0.0865(0.0038) | 0.0510(0.0254) | 0.0592(0.0305) |
| | | 5000 | 0.0232(0.0009) | 0.0293(0.0010) | 0.0211(0.0084) | 0.0217(0.0084) |
| | $\mathbf{A}_2(t)$ | 2000 | 0.0709(0.0022) | 0.0889(0.0028) | 0.0409(0.0188) | 0.0461(0.0206) |
| | | 5000 | 0.0239(0.0007) | 0.0299(0.0010) | 0.0235(0.0101) | 0.0244(0.0109) |
| | $\mathbf{A}_3(t)$ | 2000 | 0.0854(0.0055) | 0.1103(0.0071) | 0.0401(0.0381) | 0.0425(0.0386) |
| | | 5000 | 0.0407(0.0009) | 0.0535(0.0011) | 0.0256(0.0152) | 0.0286(0.0175) |
| | $\mathbf{A}_4(t)$ | 2000 | 0.1117(0.0049) | 0.1430(0.0064) | 0.0509(0.0245) | 0.0541(0.0258) |
| | | 5000 | 0.0578(0.0012) | 0.0727(0.0013) | 0.0305(0.0162) | 0.0362(0.0205) |

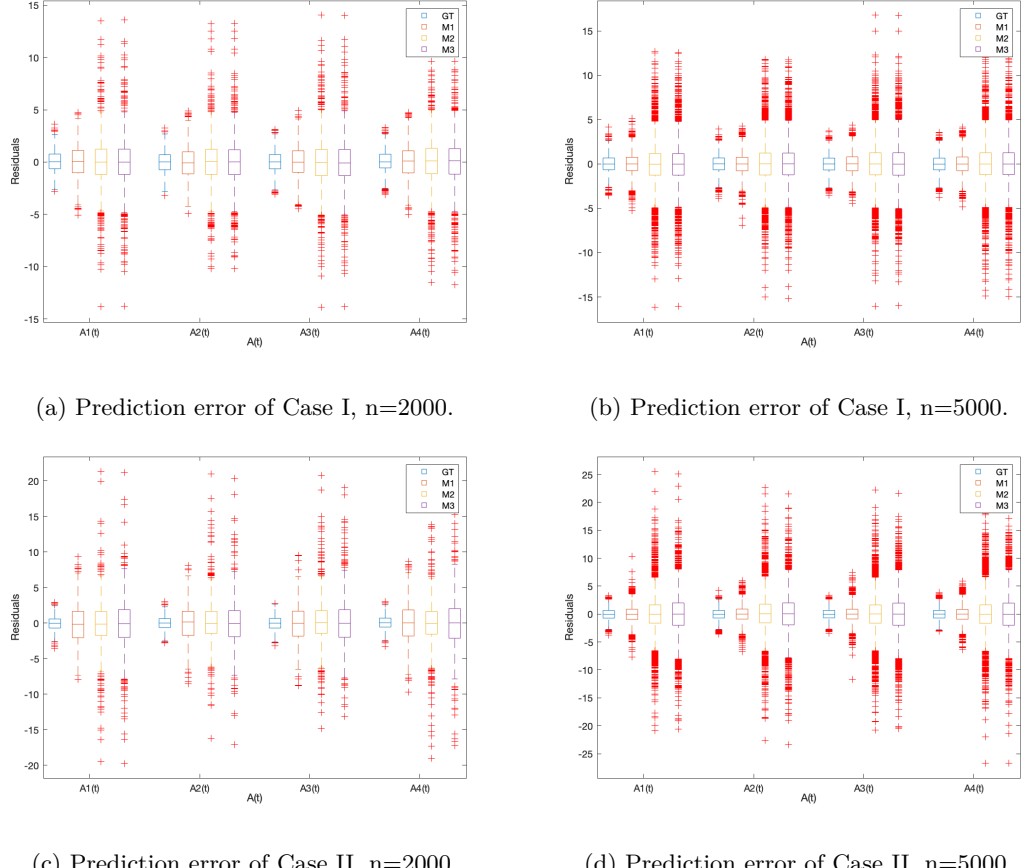

(a) Prediction error of Case I, n=2000.   (b) Prediction error of Case I, n=5000.

(c) Prediction error of Case II, n=2000.   (d) Prediction error of Case II, n=5000.

Figure 2: Box plots of prediction errors of Case I and II with 10-fold cross validation. Model $M_1$ is our VCTR model. Models $M_2$ and $M_3$ are constant coefficient tensor models, estimated by Miranda et al. (2018) and Zhou et al. (2013), respectively.

## 5.2 Selection accuracy

We now evaluate the accuracy of the penalization method for high-dimensional covariates introduced in Section 3. Let $\mathcal{F}_1 = \{(r,s) : \|(a_{rs})_v\|_2 > 0\}$, $\mathcal{F}_2 = \{(r,s) : |(a_{rs})_c| > 0, \|(a_{rs})_v\|_2 = 0\}$ and $\mathcal{F}_3 = \{(r,s) : |(a_{rs})_c| = 0, \|(a_{rs})_v\|_2 = 0\}$ be the index set of varying, constant non-zero and constant zero coefficients in tensor function $\mathbf{A}(t)$. Obviously, $\{(r,s) : r \in \{1, \cdots, R\}, s \in \{1, \cdots, S\}\} = \mathcal{F}_1 \cup \mathcal{F}_2 \cup \mathcal{F}_3$ and $\mathcal{F}_i \cap \mathcal{F}_j = \emptyset$ for $i,j \in \{1,2,3\}$. Also let $\mathcal{F}_4 = \{k, |\beta_k| > 0\}$ and $\mathcal{F}_5 = \{k, |\beta_k| = 0\}$ be the index set for significant and sparsed coefficients in one-way parameter $\boldsymbol{\beta}$. Similarly, we have $\{1, \cdots, p_0\} = \mathcal{F}_4 \cup \mathcal{F}_5$ and $\mathcal{F}_4 \cap \mathcal{F}_5 = \emptyset$. We denote $|\mathcal{F}|$ as the number of elements in a set $\mathcal{F}$. In this subsection, we denote sets $\hat{\mathcal{F}}_i$, $i = 1, \cdots, 5$ as the estimator of ground truth $\mathcal{F}_i$.

In Case III, we consider the selection performance under various spatial correlation among tensor covariates. In case IV, we consider tensor covariates with unknown coefficients greater than the sample size. In both cases, we adopt three penalty functions: Lasso, SCAD, and MCP. The tuning parameters are chosen by BIC. Following Algorithm 1, the penalized regression can be initialized via (15), which requires specifying the spline order $k_d$ and the number of interior knots $k_n$. In our implementation, we consider $k_d \in \{2,3,4\}$ and $k_n \in \{1,2,\ldots,6\}$. The optimal combination is selected using a BIC-based grid search, where for each candidate pair $(k_d, k_n)$,

the model is fitted and the corresponding BIC is evaluated. And the pair minimizing the BIC is then chosen for subsequent penalized tensor regression.

Case III : Each subject $\mathcal{X}_i$ follows the same dimension as Case I. To allow the spatial correlation among different parts, for any two subject factor matrices $\mathbf{X}^{*(s)}$ in different partitions, entries have a first order auto-regressive covariance matrix with covariance $\rho^{|s^{(1)}-s'^{(1)}|+|s^{(2)}-s'^{(2)}|}$ for $1 \leq s^{(1)}, s'^{(1)} \leq S^{(1)}$ and $1 \leq s^{(2)}, s'^{(2)} \leq S^{(2)}$. Tensor function $\mathcal{A}(t)$ is designed similarly as Case I with $\mathcal{A}^{(s)}(t) = \sum_r^R a_{rs}(t)\mathbf{x}_r^{(s1)} \circ \mathbf{x}_r^{(s2)}$. If $1 \leq r \leq s \leq 10$, $a_{rs}(t) = \sqrt{rs/RS}\sin(2\pi(t-0.5))$; if $1 \leq s < r \leq 10$, $a_{rs}(t) = \sqrt{rs/RS}$, and the rest of the tensor functions are all 0's.

Case IV : Each subject $\mathcal{X}_i$ is an $p_1 \times p_2 \times p_3$ tensor with order $D = 3$, $p_1 = p_2 = p_3 = 80$ for $i = 1, \cdots, n$. Evenly divide $\mathcal{X}_i$ into $S = 64$ parts by partitioning it into $S^{(1)} = S^{(2)} = S^{(3)} = 4$ equal segments along length, width and height separately, with $p'_1 = p'_2 = p'_3 = 20$. Set $R = 20$. For each $\tilde{\mathcal{X}}^{(s)}$ in (6), $\lambda_r^{(s)} = 1$, and $\mathbf{x}_{rs}^* \in \mathbb{R}^n, \mathbf{x}_r^{(sd)} \in \mathbb{R}^{p'_d}$ are generated in the same way as Case II. Tensor function $\mathcal{A}(t)$ has the same shape as $\mathcal{X}$, and for each part, corresponding $\mathcal{A}^{(s)}(t) = \sum_r^R a_{rs}(t)\mathbf{x}_r^{(s1)} \circ \mathbf{x}_r^{(s2)} \circ \mathbf{x}_r^{(s3)}$. If $1 \leq r \leq s \leq 10$, $a_{rs}(t) = \sqrt{rs/RS}\sin(2\pi(t-0.5))$; if $1 \leq s < r \leq 10$, $a_{rs}(t) = \sqrt{rs/RS}$ , and the rest of the tensor functions are all 0's.

In both cases, We set $p_0 = 5$ and $\boldsymbol{\beta} = (1,1,0,0,0)^\top$. $\mathbf{Z} \in \mathbb{R}^{n \times p_0}$ is a Gaussian ensemble matrix. The random error $\epsilon_i$ is generated in the same way as Case I and II.

The results for Case III are summarized in Table 2 after 500 simulations. The penalized estimation correctly identifies the true zero coefficients and non-zero coefficients with high probability. The covariate correlation slightly affects the accuracy where wrong selection occurs more often with higher auto-correlation. All three penalty functions perform quite well.

The results for Case IV are summarized in Table 3. We note that this case is more complicated than previous cases since the tensor coefficient has a greater dimension $RS = 20 \times 64$. The un-penalized estimation approach is infeasible for this challenging case. In general the selection accuracy improves with increasing sample size $n$ while the three penalty functions perform similarly well. We also plot prediction errors of Case III and IV with 10-fold cross validation in Figure 3.

# 6 Glaucoma management with fundus images

The GRAPE (Huang et al., 2023) dataset was collected in the Eye Center at the Second Affiliated Hospital of Zhejiang University, which contains 1115 records of 263 eyes from 144 glaucoma patients from 2015 to 2022, with ages ranging from 18 to 81 years. One patient may visit hospital multiple times for follow-up examinations during this period and information obtained from oculus dexter (OD) or oculus sinister(OS) during each visit will be recorded. To avoid dependence among repeated measurements, we randomly select one record from all available visits for each patient and treat it as a single independent sample in our VCTR model, which is composed of a list of visual fields (VF) value from 59 points, excluding 2 points located in blind points, a color fundus photograph (CFP), an optical coherence tomography (OCT)

Table 2: Estimation results of 500 simulations for Case III. We use sensitivity (se), specificity (sp), positive predictive value (ppv), and negative predictive value (npv) to evaluate identification accuracy, where $se = P(\mathcal{F}_i \cap \hat{\mathcal{F}}_i | \mathcal{F}_i)$, $ppv = P(\mathcal{F}_i \cap \hat{\mathcal{F}}_i | \hat{\mathcal{F}}_i)$, $sp = P(\mathcal{F}_i^c \cap \hat{\mathcal{F}}_i^c | \mathcal{F}_i^c)$ and $npv = P(\mathcal{F}_i^c \cap \hat{\mathcal{F}}_i^c | \hat{\mathcal{F}}_i^c)$. Sample size $n = 5000$. Penalty parameters of Lasso, SCAD, and MCP are chosen by BIC.

| | Lasso | | | SCAD | | | MCP | | |
|---|---|---|---|---|---|---|---|---|---|
| $\rho$ | 0.1 | 0.5 | 0.9 | 0.1 | 0.5 | 0.9 | 0.1 | 0.5 | 0.9 |
| | Zero coefficient in $\mathbf{A}(t)$ | | | | | | | | |
| se | 0.9965 | 0.9948 | 0.9926 | 0.9961 | 0.9952 | 0.9930 | 0.9974 | 0.9917 | 0.9913 |
| ppv | 0.9811 | 0.9849 | 0.9793 | 0.9922 | 0.9918 | 0.9827 | 0.9939 | 0.9917 | 0.9810 |
| sp | 0.9516 | 0.9615 | 0.9473 | 0.9802 | 0.9791 | 0.9560 | 0.9846 | 0.9791 | 0.9516 |
| npv | 0.9910 | 0.9865 | 0.9807 | 0.9901 | 0.9878 | 0.9820 | 0.9934 | 0.9792 | 0.9775 |
| | Constant non-zero coefficient in $\mathbf{A}(t)$ | | | | | | | | |
| se | 0.9712 | 0.9788 | 0.9558 | 0.9942 | 0.9827 | 0.9404 | 0.9885 | 0.9827 | 0.9481 |
| ppv | 0.9981 | 0.9923 | 0.9881 | 0.9868 | 0.9884 | 0.9781 | 0.9962 | 0.9848 | 0.9725 |
| sp | 0.9996 | 0.9985 | 0.9978 | 0.9974 | 0.9978 | 0.9959 | 0.9993 | 0.9970 | 0.9948 |
| npv | 0.9944 | 0.9959 | 0.9915 | 0.9989 | 0.9967 | 0.9886 | 0.9978 | 0.9967 | 0.9900 |
| | Varying coefficient in $\mathbf{A}(t)$ | | | | | | | | |
| se | 0.9256 | 0.9385 | 0.9359 | 0.9615 | 0.9744 | 0.9359 | 0.9795 | 0.9744 | 0.9333 |
| ppv | 0.9816 | 0.9787 | 0.9712 | 0.9948 | 0.9873 | 0.9463 | 0.9899 | 0.9720 | 0.9608 |
| sp | 0.9975 | 0.9972 | 0.9961 | 0.9993 | 0.9982 | 0.9925 | 0.9986 | 0.9961 | 0.9947 |
| npv | 0.9898 | 0.9915 | 0.9912 | 0.9947 | 0.9965 | 0.9911 | 0.9972 | 0.9964 | 0.9908 |
| | Prediction error of $\hat{\mathbf{A}}^{\ddagger}(t)$ | | | | | | | | |
| MAE | 0.0109 | 0.0111 | 0.0134 | 0.0114 | 0.0122 | 0.0127 | 0.0136 | 0.0175 | 0.0195 |
| RMISE | 0.0198 | 0.0217 | 0.0219 | 0.0192 | 0.0234 | 0.0240 | 0.0245 | 0.0397 | 0.0414 |
| | Prediction error of $\hat{\boldsymbol{\beta}}^{\ddagger}$ | | | | | | | | |
| MAE | 0.0047 | 0.0082 | 0.0092 | 0.0045 | 0.0063 | 0.0136 | 0.0046 | 0.0058 | 0.0141 |
| RMSE | 0.0077 | 0.0133 | 0.0149 | 0.0067 | 0.0101 | 0.0209 | 0.0080 | 0.0096 | 0.0207 |

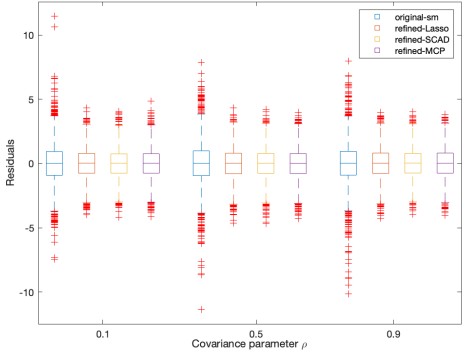

(a) Prediction error of Case III.

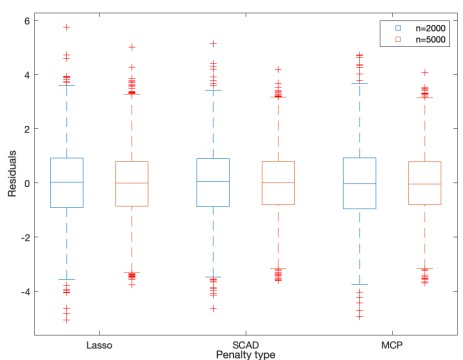

(b) Prediction error of Case IV.

Figure 3: Box plots of prediction error of Case III and IV with 10-fold cross validation. (a) shows prediction errors of different covariance parameter $\rho$'s in Case III, 'Oracle' is our tensor regression model with true model structure, 'Unpenalized' is the un-penalized tensor regression model (section 2.3), and 'Lasso', 'SCAD', and 'MCP' are our final tensor regression model (section 3.2) after model structure identification with Lasso, SCAD and MCP penalties (section 3.1). (b) shows prediction errors of the final tensor regression model (section 3.2) after model structure identification under overfitting situation (Case IV), with different sample size $n = 2000, 5000$. In case IV, un-penalized tensor regression model cannot work.

Table 3: Estimation results of 500 simulations for Case IV. We use sensitivity (se), specificity (sp), positive predictive value (ppv), and negative predictive value (npv) to evaluate identification accuracy, where se $= \mathrm{P}(\mathcal{F}_i \cap \hat{\mathcal{F}}_i | \mathcal{F}_i)$, ppv $= \mathrm{P}(\mathcal{F}_i \cap \hat{\mathcal{F}}_i | \hat{\mathcal{F}}_i)$, sp $= \mathrm{P}(\mathcal{F}_i^c \cap \hat{\mathcal{F}}_i^c | \hat{\mathcal{F}}_i^c)$ and npv $= \mathrm{P}(\mathcal{F}_i^c \cap \hat{\mathcal{F}}_i^c | \mathcal{F}_i^c)$. All tuning parameters for Lasso, SCAD, and MCP are chosen by BIC.

| | Lasso | | | SCAD | | | MCP | | |
|---|---|---|---|---|---|---|---|---|---|
| $n$ | 2000 | 3500 | 5000 | 2000 | 3500 | 5000 | 2000 | 3500 | 5000 |
| | Zero coefficient in $\mathbf{A}(t)$ | | | | | | | | |
| se | 0.9916 | 0.9971 | 0.9975 | 0.9944 | 0.9962 | 0.9971 | 0.9913 | 0.9947 | 0.9947 |
| ppv | 0.9822 | 0.9891 | 0.9929 | 0.9954 | 0.9970 | 0.9982 | 0.9972 | 0.9967 | 0.9981 |
| sp | 0.6271 | 0.7729 | 0.8525 | 0.9051 | 0.9373 | 0.9627 | 0.9424 | 0.9322 | 0.9610 |
| npv | 0.7934 | 0.9295 | 0.9430 | 0.8879 | 0.9249 | 0.9423 | 0.8421 | 0.8750 | 0.8981 |
| | Constant non-zero coefficient in $\mathbf{A}(t)$ | | | | | | | | |
| se | 0.6737 | 0.7895 | 0.8553 | 0.8974 | 0.9316 | 0.9658 | 0.9026 | 0.9263 | 0.9684 |
| ppv | 0.9217 | 0.9754 | 0.9739 | 0.9259 | 0.9332 | 0.9687 | 0.8989 | 0.9273 | 0.9497 |
| sp | 0.9981 | 0.9994 | 0.9993 | 0.9977 | 0.9979 | 0.9990 | 0.9968 | 0.9985 | 0.9977 |
| npv | 0.9901 | 0.9936 | 0.9956 | 0.9969 | 0.9979 | 0.9990 | 0.9970 | 0.9977 | 0.9990 |
| | Varying coefficient in $\mathbf{A}(t)$ | | | | | | | | |
| se | 0.5143 | 0.7429 | 0.8476 | 0.9143 | 0.9476 | 0.9571 | 0.9048 | 0.9429 | 0.9476 |
| ppv | 0.5814 | 0.8564 | 0.8968 | 0.8265 | 0.8984 | 0.9129 | 0.7917 | 0.8161 | 0.8529 |
| sp | 0.9932 | 0.9978 | 0.9983 | 0.9967 | 0.9984 | 0.9982 | 0.9939 | 0.9963 | 0.9971 |
| npv | 0.9919 | 0.9957 | 0.9975 | 0.9986 | 0.9991 | 0.9993 | 0.9984 | 0.9990 | 0.9991 |
| | Prediction error of $\hat{\mathbf{A}}^{\ddagger}(t)$ | | | | | | | | |
| MAE | 0.0242 | 0.0135 | 0.0108 | 0.0132 | 0.0055 | 0.0040 | 0.0097 | 0.0058 | 0.0041 |
| RMISE | 0.0375 | 0.0231 | 0.0194 | 0.0256 | 0.0140 | 0.0117 | 0.0223 | 0.0145 | 0.0115 |
| | Prediction error of $\hat{\boldsymbol{\beta}}^{\ddagger}$ | | | | | | | | |
| MAE | 0.0361 | 0.0148 | 0.0090 | 0.0141 | 0.0073 | 0.0065 | 0.0218 | 0.0085 | 0.0064 |
| RMSE | 0.0514 | 0.0225 | 0.0133 | 0.0221 | 0.0124 | 0.0107 | 0.0325 | 0.0121 | 0.0101 |

measurements and clinical information, such as age, gender, and intraocular pressure (IOP). In Figure 4, we show the distribution of subjects' age and the corresponding IOPs, and in Figure 1, we give an example of two different fundus images (CFP and ROI), together with 59 locations of VF points.

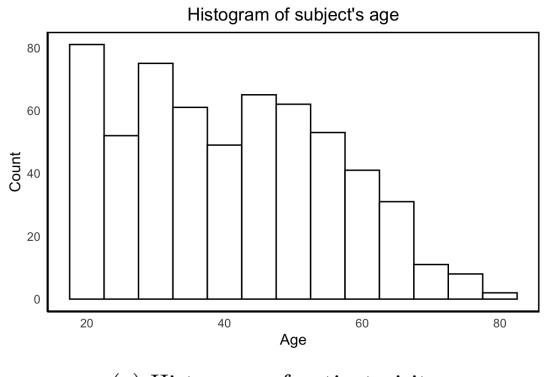

(a) Histogram of patient visits

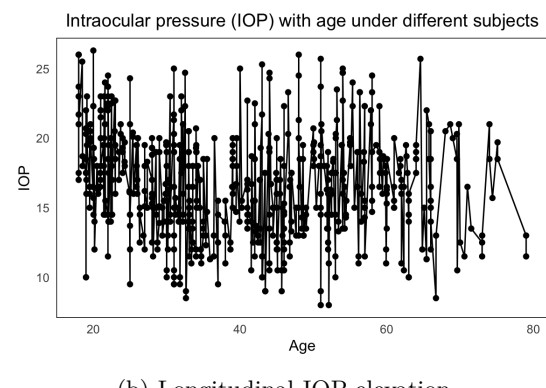

(b) Longitudinal IOP elevation

Figure 4: Fundus image analysis: statistical summary for original GRAPE dataset. (a) shows the histogram of subject's ages in the whole dataset. (b) shows the intraocular pressure (IOP) value with different ages.

The scientific objective is to quantify the relationship between IOP and fundus images of

glaucoma patients, along with their clinical data. Figure 1 displays an example of fundus images (CFP and ROI), before being fed into our model estimation process. During data processing, we first drop samples without fundus images. Then we regard IOPs beyond 2 standard deviations as outliers and also drop the corresponding samples. Finally, $n = 141$ samples remains (72 from male glaucoma patients and 69 from female). In dealing with VF values, we impute missing data and fix outliers with the mean value of the corresponding VF location. We standardized all continuous variables to be mean 0 and variance 1. We resize all remaining images into $\mathcal{X}_i \in \mathbb{R}^{p_1 \times p_2 \times p_3}$, where $p_1 = 192$, $p_2 = 192$ and $p_3 = 3$, representing length, width and number of color channels respectively. In this dataset, both oculus dexter (OD) and oculus sinister (OS) can be included as samples, in which the optic disc and macula are positioned oppositely and blood vessels may have different orientations. To avoid these discrepancies during the tensor partitioning process, we flipped all OS images horizontally and concatenate unified tensor together as $\tilde{\mathcal{X}} \in \mathbb{R}^{n \times p_1 \times p_2 \times p_3}$, and an even division is applied to the length and the width, resulting with $S^{(1)}$ and $S^{(2)}$ equal segments and a total of $S = S^{(1)} \times S^{(2)}$ partitions. For each partition, we denote them as $\tilde{\mathcal{X}}^{(s)} \in \mathbb{R}^{n \times p_1^{(s)} \times p_2^{(s)} \times p_3}$ for $s = 1 \cdots S$. A unified CP decomposition with the same rank $R$ introduced in (6) is applied on $\tilde{\mathcal{X}}^{(s)}$ to obtain factor matrices $\mathbf{X}_s^*$. We choose optic disc area as our ROI since the fundus manifestation of glaucoma contains optic disc rim narrowing, cup-to-disc ratio (CDR) increasing, large extent of parapapillary atrophy, and RNFL defect (Prum et al., 2016), which are mainly focused on the area around optic disc.

Our hypothesis is that the association between IOP and fundus images may vary with different ages, thus we consider model (3) with age being the index variable $t_i$. The one-way covariate $\mathbf{z}_i$ includes subject's gender and the corresponding 5 view field values (cf. location ID 1, 22, 25, 28 and 31 in Figure 1.(f)). To achieve consistency under the same location for OD and OS image, we record VF value, as shown in (c) and (f) in Figure 1. 10-fold cross validation is used to select $S$ and $R$, and the best partition is $S = 3 \times 3 \times 1 = 9$ with the best CP rank $R = 2$ for both CFP and ROI; see Table 4 below for further details. With such specifications, we implement our VCTR model, and the estimated varying coefficient $\hat{\mathbf{A}}(t)$ and one-way constant coefficient $\hat{\boldsymbol{\beta}}$ are shown in Figure 5 and Table 5, respectively. Table 6 shows the estimated constant coefficient $\boldsymbol{\beta}$ for gender and 59 different VF locations with tensor covariate CFP and ROI. We remark that variable selection process is inherently affected by the partition boundary. However, as detailed in Table 4, while extreme choices of partitions or ranks lead to performance degradation, moderate deviations from the optimal hyperparameters result in highly stable prediction errors, confirming the robustness of the model's predictive capabilities. In practice, when prior knowledge or domain expertise is available, the partition strategy can be specified accordingly, which effectively determines S. Otherwise, we suggest to specify a candidate range for both CP ranks and feasible tensor partition schemes, and conduct a grid search using 10-fold cross-validation to select the optimal combination. This selection procedure is standard in high-dimensional tensor modeling and has been implemented in our real data analysis, with code publicly available at `https://github.com/JasonHe95/VCTR`. Empirically, we observe a clear trade-off between the partition granularity and the required CP rank. Specifically, when the tensor is partitioned into fewer partitions (smaller S), a larger CP rank R is typically needed to capture the underlying structure. In contrast, when a finer partition is adopted (larger S), the optimal CP rank tends

to decrease. This phenomenon is also reflected in the results reported in Table 4.

Comparing the un-penalized estimator with the penalized estimator $\hat{a}_{rs}(t)$ in Figure 5, we observe that there are more varying coefficients for ROI than CFP, especially for partitions surrounding the optic disc. This suggests the increasing cup-to-disc ratio could lead to greater variability of image region effects on patients' intraocular pressure levels. These new insights would be difficult to uncover without applying the proposed tensor varying coefficient model.

Table 4: Fundus image analysis: RMSE for tensor partition on colored fundus photograph (CFP) and region of interest (ROI) with different $S$ and $R$. All results are generated from 10-fold cross validation.

| $S_1 \times S_2 \times S_3$ | Colored fundus photograph (CFP) | | | | Region of interest (ROI) | | | |
|---|---|---|---|---|---|---|---|---|
| | $R=1$ | $R=2$ | $R=3$ | $R=4$ | $R=1$ | $R=2$ | $R=3$ | $R=4$ |
| $2 \times 2 \times 1$ | 1.4827 | 1.2961 | 1.1087 | 1.2326 | 1.4889 | 1.1754 | 1.1083 | 1.2543 |
| $2 \times 3 \times 1$ | 1.1374 | 1.0611 | 1.3288 | 1.5936 | 1.1821 | 1.0509 | 1.2513 | 1.5195 |
| $3 \times 2 \times 1$ | 1.1584 | 1.0248 | 1.2897 | 1.4458 | 1.2087 | 1.0399 | 1.3050 | 1.5412 |
| $3 \times 3 \times 1$ | 1.0429 | **0.9766** | 1.1185 | 1.3419 | 1.0427 | **0.9711** | 1.1807 | 1.3163 |
| $3 \times 4 \times 1$ | 1.0548 | 1.3743 | 1.5311 | 1.8173 | 1.0725 | 1.3699 | 1.6377 | 1.7911 |
| $4 \times 3 \times 1$ | 1.0826 | 1.3492 | 1.4884 | 1.7748 | 1.0926 | 1.3857 | 1.6668 | 1.8506 |
| $4 \times 4 \times 1$ | 1.2811 | 1.6503 | 1.9263 | / | 1.2555 | 1.8390 | 2.0421 | / |

Table 5: Fundus image analysis: Estimated results of regression coefficients $\hat{\boldsymbol{\beta}}$ for one-way covariate in GRAPE dataset. 95% confidence intervals are based on 500 bootstrap resamples.

| $\mathbf{z}$ | Colored fundus photograph (CFP) | | | Region of interest (ROI) | | |
|---|---|---|---|---|---|---|
| | $\hat{\boldsymbol{\beta}}^{\dagger}$ | $\hat{\boldsymbol{\beta}}^{\ddagger}$ | 95% CI | $\hat{\boldsymbol{\beta}}^{\dagger}$ | $\hat{\boldsymbol{\beta}}^{\ddagger}$ | 95% CI |
| Gender | -0.0835 | 0 | / | -0.0647 | 0 | / |
| $VF_1$ | -0.1422 | -0.0915 | [-0.1891, -0.0774] | -0.1018 | -0.0606 | [-0.1650, -0.0469] |
| $VF_{22}$ | -0.1427 | -0.1501 | [-0.2089, -0.1231] | -0.1570 | -0.0934 | [-0.1964, -0.0833] |
| $VF_{25}$ | 0.0045 | 0 | / | 0.0204 | 0 | / |
| $VF_{28}$ | -0.0790 | 0 | / | 0.0488 | 0 | / |
| $VF_{31}$ | -0.1284 | -0.0877 | [-0.1862, -0.0707] | -0.1791 | -0.1349 | [-0.1840, -0.0950] |

From Table 5, insignificant gender effects are observed on CFP and ROI images. Male patients tend to have slightly higher scaled intraocular pressure than female patients, similar to earlier findings in Liu et al. (2022). For the view field values, only $VF_1$, $VF_{22}$ and $VF_{31}$ are significantly associated with the response with negative effects. These VFs are located near optic disc, an important area for diagnosing glaucoma.

Additional results for the estimated constant coefficient $\boldsymbol{\beta}$ for gender and 59 different VF locations with tensor covariate, and the estimated penalized varying-coefficient functions $a_{rs}(t)$ for CFP and ROI are provided Table 6 and Figure 6 respectively. The corresponding mean and CI curves are generated via bootstrapping.

At last, we compared our VCTR model with other two constant coefficient tensor models consistent with those in section 5, and the experiment results are shown in Table 7. For model $M_3$ proposed in Zhou et al. (2013), due to the relatively small sample size ($n = 141$), we downscale the image dimensions to $p_1 = p_2 = 16$ to avoid excessive number of parameters. Meanwhile, the CP rank is fixed at $R = 2$, consistent with that used in our VCTR model, and ensuring a fair model complexity. However, we can find it is overfitting. While this model has the best performance in in-sample errors, its prediction ability is the worst in out-of-sample error. Our

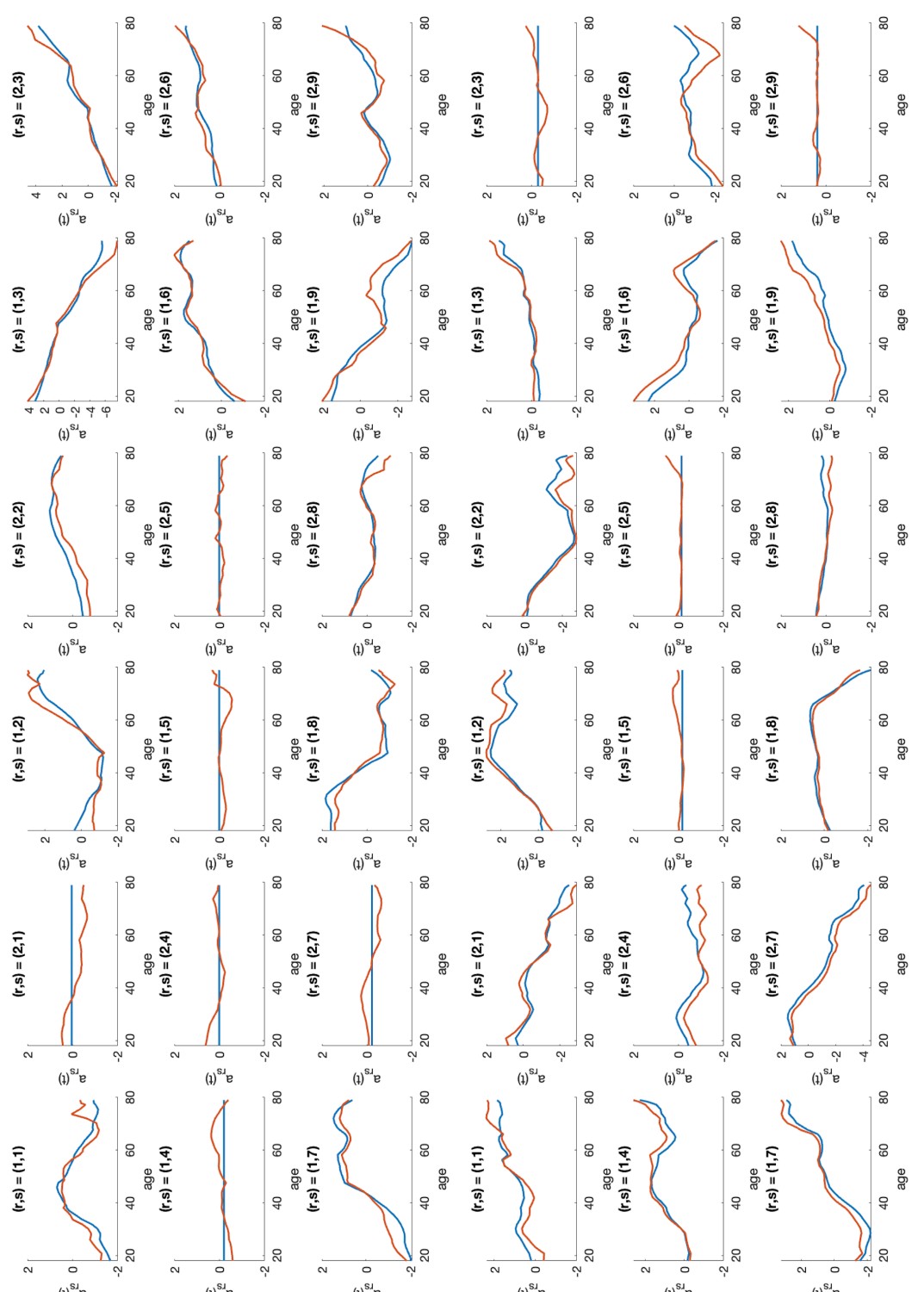

Figure 5: Fundus image analysis: estimated varying coefficient $\hat{a}_{rs}(t)$ for $\mathbf{X}_{\mathrm{CFP}}$ generated from colored fundus photogragh (CFP, the top three rows) and $\mathbf{X}_{\mathrm{ROI}}$ generated from region of interest (ROI, the bottom three rows). Red lines are $\hat{a}_{rs}(t)$ from original kernel smoothing, and blue lines are $\hat{a}_{rs}(t)$ from refined kernel smoothing after identifying coefficients.

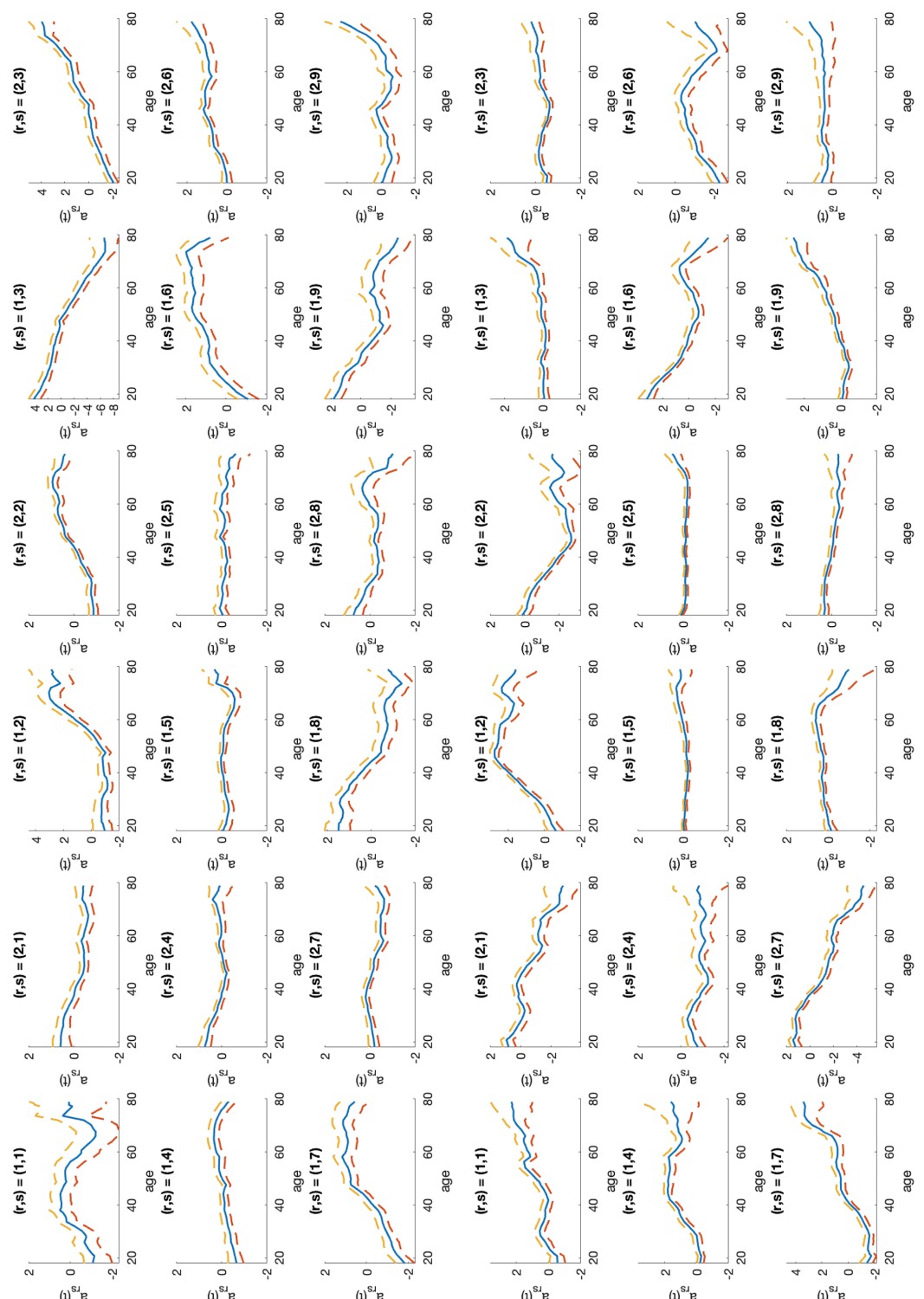

Figure 6: Fundus image analysis: the estimated varying-coefficient functions $\hat{a}_{rs}(t)$ for CFP (top three rows) and ROI (bottom three rows) images. The solid line represents the mean estimated value of $\hat{a}_{rs}(t)$, and the dashed line represents 95% confidence interval, generated from 500 bootstrap resamples.

Table 6: Fundus image analysis: Estimated constant coefficients $\hat{\beta}$ for one-way covariate $\mathbf{z}$. $\hat{\beta}^\dagger$ and $\hat{\beta}^\ddagger$ are estimated from original kernel smoothing, and refined kernel smoothing after identifying coefficients.

|  |  | Gender | VF1 | VF2 | VF3 | VF4 | VF5 | VF6 | VF7 | VF8 | VF9 | VF10 | VF11 |
|---|---|---|---|---|---|---|---|---|---|---|---|---|---|
| CFP | $\hat{\beta}^\dagger$ | 0.0449 | 0.0026 | -0.0703 | 0.0310 | 0.1350 | 0.0214 | -0.0703 | -0.1108 | 0.0382 | -0.0873 | 0.1089 | 0.0061 |
|  | $\hat{\beta}^\ddagger$ | 0.0000 | 0.0000 | 0.0000 | 0.0000 | **-0.0251** | 0.0000 | 0.0000 | 0.0000 | **0.0246** | 0.0000 | 0.0000 | 0.0000 |
| ROI | $\hat{\beta}^\dagger$ | -0.2304 | 0.0224 | -0.0464 | 0.0029 | 0.0925 | 0.0185 | -0.0985 | -0.1165 | 0.0433 | -0.1209 | 0.0817 | -0.0004 |
|  | $\hat{\beta}^\ddagger$ | 0.0000 | 0.0000 | 0.0000 | 0.0000 | **0.0590** | 0.0000 | 0.0000 | 0.0000 | **0.1076** | 0.0000 | 0.0000 | 0.0000 |

|  |  | VF12 | VF13 | VF14 | VF15 | VF16 | VF17 | VF18 | VF19 | VF20 | VF21 | VF22 | VF23 |
|---|---|---|---|---|---|---|---|---|---|---|---|---|---|
| CFP | $\hat{\beta}^\dagger$ | 0.1360 | 0.0351 | -0.0748 | -0.0073 | 0.1046 | -0.0137 | -0.0333 | -0.0365 | -0.0197 | 0.1786 | 0.1801 | -0.0919 |
|  | $\hat{\beta}^\ddagger$ | **0.0858** | 0.0000 | 0.0000 | 0.0000 | 0.0000 | 0.0000 | 0.0000 | 0.0000 | 0.0000 | **-0.1220** | **-0.1390** | 0.0000 |
| ROI | $\hat{\beta}^\dagger$ | 0.2076 | -0.0237 | 0.0496 | 0.0860 | 0.0667 | -0.0394 | -0.0929 | 0.0869 | -0.0122 | 0.0810 | 0.1262 | -0.0546 |
|  | $\hat{\beta}^\ddagger$ | **0.1688** | 0.0000 | 0.0000 | 0.0000 | 0.0000 | 0.0000 | 0.0000 | 0.0000 | 0.0000 | 0.0000 | **-0.0927** | 0.0000 |

|  |  | VF24 | VF25 | VF26 | VF27 | VF28 | VF29 | VF30 | VF31 | VF32 | VF33 | VF34 | VF35 |
|---|---|---|---|---|---|---|---|---|---|---|---|---|---|
| CFP | $\hat{\beta}^\dagger$ | -0.0656 | 0.0598 | 0.0326 | -0.0171 | -0.0551 | -0.0079 | 0.0241 | -0.1621 | 0.0671 | 0.0297 | 0.0169 | -0.0182 |
|  | $\hat{\beta}^\ddagger$ | 0.0000 | 0.0000 | **0.0166** | 0.0000 | 0.0000 | 0.0000 | 0.0000 | **-0.1647** | 0.0000 | **-0.0128** | 0.0000 | 0.0000 |
| ROI | $\hat{\beta}^\dagger$ | -0.1070 | 0.1096 | -0.0591 | -0.1407 | 0.0100 | 0.0075 | 0.0124 | -0.0822 | 0.0928 | -0.0185 | 0.1679 | -0.1079 |
|  | $\hat{\beta}^\ddagger$ | 0.0000 | 0.0000 | **0.0370** | **-0.0613** | 0.0000 | 0.0000 | 0.0000 | **-0.0652** | 0.0000 | 0.0000 | 0.0000 | 0.0000 |

|  |  | VF36 | VF37 | VF38 | VF39 | VF40 | VF41 | VF42 | VF43 | VF44 | VF45 | VF46 | VF47 |
|---|---|---|---|---|---|---|---|---|---|---|---|---|---|
| CFP | $\hat{\beta}^\dagger$ | -0.0050 | 0.0824 | -0.0795 | 0.1766 | -0.0673 | 0.0275 | -0.0827 | 0.1662 | 0.0199 | 0.1022 | -0.0603 | 0.0374 |
|  | $\hat{\beta}^\ddagger$ | 0.0000 | 0.0000 | 0.0000 | 0.0000 | 0.0000 | 0.0000 | 0.0000 | **0.0789** | 0.0000 | 0.0000 | 0.0000 | 0.0000 |
| ROI | $\hat{\beta}^\dagger$ | 0.0211 | -0.0337 | -0.0988 | 0.0973 | -0.0169 | -0.0155 | -0.0347 | 0.1645 | 0.0101 | -0.0616 | 0.0189 | -0.0687 |
|  | $\hat{\beta}^\ddagger$ | 0.0000 | 0.0000 | 0.0000 | 0.0000 | 0.0000 | **0.0075** | 0.0000 | **0.1033** | 0.0000 | 0.0000 | 0.0000 | 0.0000 |

|  |  | VF48 | VF49 | VF50 | VF51 | VF52 | VF53 | VF54 | VF55 | VF56 | VF57 | VF58 | VF59 |
|---|---|---|---|---|---|---|---|---|---|---|---|---|---|
| CFP | $\hat{\beta}^\dagger$ | -0.0805 | 0.0864 | -0.0614 | 0.1146 | -0.1118 | -0.0993 | -0.0722 | -0.0275 | -0.0158 | -0.0052 | -0.1165 | -0.0641 |
|  | $\hat{\beta}^\ddagger$ | **-0.0516** | 0.0000 | 0.0000 | 0.0000 | 0.0000 | **-0.0599** | 0.0000 | 0.0000 | 0.0000 | 0.0000 | 0.0000 | 0.0000 |
| ROI | $\hat{\beta}^\dagger$ | -0.0539 | 0.0949 | 0.0607 | 0.0186 | -0.0853 | -0.0678 | 0.0154 | -0.0166 | 0.0373 | 0.0209 | -0.0487 | -0.0638 |
|  | $\hat{\beta}^\ddagger$ | **-0.0148** | 0.0000 | 0.0000 | 0.0000 | 0.0000 | **-0.0874** | 0.0000 | 0.0000 | 0.0000 | 0.0000 | 0.0000 | 0.0000 |

refined $M_1$ model has slightly worse performance in in-sample error compared to original $M_1$ model. However, due to fewer functions and parameters to estimate resulting from the model structure identification, refined $M_1$ model shows better performance in out-of-sample error than original $M_1$ model. The tensor partition model with constant parameters $M_2$ always has a worse performance comapared with our model in both in-sample error and out-of-sample error.

Table 7: Fundus image analysis: prediction error for different models. Model $M_1$ is our VCPLT model proposed (section 2.3), model refined $M_1$ is our refined tensor regression model (section 3.2) after SCAD penalty (section 3). Model $M_2$ and $M_3$ are two different constant coefficient tensor models consistent with models in Figure 2. In-sample error is calculated without training-test set split. Out-of-sample error is calculated by 10-fold cross validation. MAE is the mean absolute error, RMSE is the root of the mean squared error, and SDE is the standard deviation of error.

| Model | In-sample error | | | Out-of-sample error | | |
|---|---|---|---|---|---|---|
| | MAE | RMSE | SDE | MAE | RMSE | SDE |
| Colored fundus photograph (CFP) | | | | | | |
| $M_1$ | 0.5674 | 0.7268 | 0.5271 | 0.7888 | 1.0125 | 0.9693 |
| Refined $M_1$ | 0.6154 | 0.7533 | 0.5467 | **0.7679** | **0.9398** | **0.8709** |
| $M_2$ | 0.7176 | 0.8759 | 0.7673 | 0.8937 | 1.1725 | 1.1935 |
| $M_3$ | **0.4779** | **0.6182** | **0.3822** | 1.3788 | 1.7685 | 1.8574 |
| Region of interest (ROI) | | | | | | |
| $M_1$ | 0.5605 | 0.7033 | 0.4938 | 0.8228 | 0.9938 | 0.9838 |
| Refined $M_1$ | 0.5835 | 0.7326 | 0.5201 | **0.7555** | **0.8879** | **0.9118** |
| $M_2$ | 0.7054 | 0.8664 | 0.7506 | 0.8749 | 1.1656 | 1.2002 |
| $M_3$ | **0.4602** | **0.5618** | **0.3156** | 1.3380 | 1.7432 | 1.9341 |

# 7  Appendix: Proof of asymptotic theory

## 7.1  Asymptotics of the regression estimates

**Proof of Theorem 1**: For any fixed $t \in [0,1]$, let $(\hat{\boldsymbol{\beta}}, \hat{\mathbf{A}}(t), \dot{\hat{\mathbf{A}}}(t))$ be the minimizer of (10), which implies that

$$
\mathbf{0} = \frac{1}{n} \sum_{i=1}^{n} \left[ \mathbf{v}_i^\top \text{vec}(\hat{\mathbf{A}}(t)) + (t_i - t)\mathbf{v}_i^\top \text{vec}(\dot{\hat{\mathbf{A}}}(t)) + \mathbf{z}_i^\top \hat{\boldsymbol{\beta}} - y_i \right] \begin{bmatrix} \mathbf{z}_i \\ \mathbf{v}_i \\ (t_i - t)\mathbf{v}_i \end{bmatrix} K_h(t_i - t)
$$

$$
= \frac{1}{n} \sum_{i=1}^{n} \left[ \mathbf{v}_i^\top \left[ \text{vec}(\hat{\mathbf{A}}(t)) - \text{vec}(\mathbf{A}^*(t)) \right] + (t_i - t)\mathbf{v}_i^\top \left[ \text{vec}(\dot{\hat{\mathbf{A}}}(t)) - \text{vec}(\dot{\mathbf{A}}^*(t)) \right] + \mathbf{z}_i^\top \left[ \hat{\boldsymbol{\beta}} - \boldsymbol{\beta}^* \right] \right.
$$

$$
\left. - \frac{1}{2}(t_i - t)^2 \mathbf{v}_i^\top \text{vec}(\ddot{\mathbf{A}}^*(t)) - \frac{1}{6}(t_i - t)^3 \mathbf{v}_i^\top \text{vec}(\dddot{\mathbf{A}}^*(\xi_i)) - \epsilon_i \right] \begin{bmatrix} \mathbf{z}_i \\ \mathbf{v}_i \\ (t_i - t)\mathbf{v}_i \end{bmatrix} K_h(t_i - t)
$$

where $\xi_i$ is between $t_i$ and $t$ for $i = 1, \cdots n$.

Denote $\tilde{\mathbf{v}}_i(t) = (\mathbf{z}_i^\top, \mathbf{v}_i^\top, (t_i - t)\mathbf{v}_i^\top)^\top$ for $i = 1, \cdots, n$, and $\tilde{\mathbf{V}} := (\tilde{\mathbf{v}}_1(t), \cdots, \tilde{\mathbf{v}}_n(t))^\top = \begin{bmatrix} \mathbf{z}_1^\top & \mathbf{v}_1^\top & (t_1 - t)\mathbf{v}_1^\top \\ \vdots & \vdots & \vdots \\ \mathbf{z}_n^\top & \mathbf{v}_n^\top & (t_n - t)\mathbf{v}_n^\top \end{bmatrix}$, and $\mathbf{W}(t) = \begin{bmatrix} K_h(t_1 - t) & & \\ & \ddots & \\ & & K_h(t_n - t) \end{bmatrix}$. The above equality is

equivalent to:

$$
\begin{bmatrix}
\hat{\boldsymbol{\beta}} - \boldsymbol{\beta}^* \\
\mathrm{vec}(\hat{\mathbf{A}}(t)) - \mathrm{vec}(\mathbf{A}^*(t)) \\
\mathrm{vec}(\hat{\dot{\mathbf{A}}}(t)) - \mathrm{vec}(\dot{\mathbf{A}}^*(t))
\end{bmatrix}
= \frac{1}{2} [\tilde{\mathbf{V}}^\top(t)\mathbf{W}(t)\tilde{\mathbf{V}}(t)]^{-1}\tilde{\mathbf{V}}^\top(t)\mathbf{W}(t)
\begin{bmatrix}
\mathbf{v}_1^\top (t_1 - t)^2 \mathrm{vec}(\ddot{\mathbf{A}}^*(t)) \\
\vdots \\
\mathbf{v}_n^\top (t_n - t)^2 \mathrm{vec}(\ddot{\mathbf{A}}^*(t))
\end{bmatrix}
$$

$$
+ [\tilde{\mathbf{V}}^\top(t)\mathbf{W}(t)\tilde{\mathbf{V}}(t)]^{-1}\tilde{\mathbf{V}}^\top(t)\mathbf{W}(t)\boldsymbol{\epsilon} + R, \tag{31}
$$

where $R$ is a remainder term involves of $\ddot{\mathbf{A}}^*(\xi_i)$ and $(t_i - t)^3 K_h(t_i - t)$. We shall evaluate the bias induced by the first term on the right hand side of the above equation and $R$ first.

For $\tilde{\mathbf{V}}^\top(t)\mathbf{W}(t)\tilde{\mathbf{V}}(t)$, we have

$$\tilde{\mathbf{V}}^\top(t)\mathbf{W}(t)\tilde{\mathbf{V}}(t) \tag{32}$$

$$
= \begin{bmatrix}
\mathbf{z}_1 & \cdots & \mathbf{z}_n \\
\mathbf{v}_1 & \cdots & \mathbf{v}_n \\
(t_1 - t)\mathbf{v}_1 & \cdots & (t_n - t)\mathbf{v}_n
\end{bmatrix}
\begin{bmatrix}
K_h(t_1 - t) & & \\
& \ddots & \\
& & K_h(t_n - t)
\end{bmatrix}
\begin{bmatrix}
\mathbf{z}_1^\top & \mathbf{v}_1^\top & (t_1 - t)\mathbf{v}_1^\top \\
\vdots & \vdots & \vdots \\
\mathbf{z}_n^\top & \mathbf{v}_n^\top & (t_n - t)\mathbf{v}_n^\top
\end{bmatrix}
$$

$$
= \begin{bmatrix}
K_h(t_1 - t)\mathbf{z}_1 & \cdots & K_h(t_n - t)\mathbf{z}_n \\
K_h(t_1 - t)\mathbf{v}_1 & \cdots & K_h(t_n - t)\mathbf{v}_n \\
(t_1 - t)K_h(t_1 - t)\mathbf{v}_1 & \cdots & (t_n - t)K_h(t_n - t)\mathbf{v}_n
\end{bmatrix}
\begin{bmatrix}
\mathbf{z}_1^\top & \mathbf{v}_1^\top & (t_1 - t)\mathbf{v}_1^\top \\
\vdots & \vdots & \vdots \\
\mathbf{z}_n^\top & \mathbf{v}_n^\top & (t_n - t)\mathbf{v}_n^\top
\end{bmatrix}
$$

$$
= \begin{bmatrix}
\sum_{i=1}^n K_h(t_i - t)\mathbf{z}_i\mathbf{z}_i^\top & \sum_{i=1}^n K_h(t_i - t)\mathbf{z}_i\mathbf{v}_i^\top & \sum_{i=1}^n (t_i - t)K_h(t_i - t)\mathbf{z}_i\mathbf{v}_i^\top \\
\sum_{i=1}^n K_h(t_i - t)\mathbf{v}_i\mathbf{z}_i^\top & \sum_{i=1}^n K_h(t_i - t)\mathbf{v}_i\mathbf{v}_i^\top & \sum_{i=1}^n (t_i - t)K_h(t_i - t)\mathbf{v}_i\mathbf{v}_i^\top \\
\sum_{i=1}^n (t_i - t)K_h(t_i - t)\mathbf{v}_i\mathbf{z}_i^\top & \sum_{i=1}^n (t_i - t)K_h(t_i - t)\mathbf{v}_i\mathbf{v}_i^\top & \sum_{i=1}^n (t_i - t)^2 K_h(t_i - t)\mathbf{v}_i\mathbf{v}_i^\top
\end{bmatrix}.
$$

For brevity, we only provide evaluations on $\sum_{i=1}^n K_h(t_i - t)\mathbf{v}_i\mathbf{v}_i^\top$, $\sum_{i=1}^n (t_i - t)K_h(t_i - t)\mathbf{v}_i\mathbf{v}_i^\top$ and $\sum_{i=1}^n (t_i - t)^2 K_h(t_i - t)\mathbf{v}_i\mathbf{v}_i^\top$ in (32), and other entries can be derived similarly. Note that,

$$
\mathrm{E}\Big[\sum_{i=1}^n K_h(t_i - t)\mathbf{v}_i\mathbf{v}_i^\top\Big] = n\mathrm{E}(\mathbf{v}\mathbf{v}^\top)\mathrm{E}\big[K_h(t_i - t)\big] = n\Omega_{\mathbf{v}}\mathrm{E}\big[K_h(t_i - t)\big],
$$

$$
\mathrm{E}(K_h(t_i - t)) = \frac{1}{h}\int K\left(\frac{t_i - t}{h}\right)f(t_i)dt_i = \frac{1}{h}\int K(u)f(t + hu)d(t + hu) = f(t)(1 + O(h^2)).
$$

We thus have,

$$
\mathrm{E}\big[n^{-1}\sum_{i=1}^n K_h(t_i - t)\mathbf{v}_i\mathbf{v}_i^\top\big] = \Omega_{\mathbf{v}}f(t) + O(h^2).
$$

Similarly, from the facts that

$$
\mathrm{E}\Big[\sum_{i=1}^n (t_i - t)K_h(t_i - t)\mathbf{v}_i\mathbf{v}_i^\top\Big] = n\mathrm{E}(\mathbf{v}\mathbf{v}^\top)\mathrm{E}\big[(t_i - t)K_h(t_i - t)\big] = n\Omega_{\mathbf{v}}\mathrm{E}\big[(t_i - t)K_h(t_i - t)\big],
$$

$$
\mathrm{E}\big[(t_i - t)K_h(t_i - t)\big] = \frac{1}{h}\int (t_i - t)K\left(\frac{t_i - t}{h}\right)f(t_i)dt_i = \int uK(u)f(t + hu)d(t + hu)
$$
$$
= h^2\dot{f}(t)\mu_2 + O(h^3),
$$

we have,

$$\mathrm{E}\big[n^{-1}\sum_{i=1}^{n}(t_i-t)K_h(t_i-t)\mathbf{v}_i\mathbf{v}_i^\top\big]=\Omega(h^2).$$

Similarly, from the facts that,

$$\mathrm{E}\big[\sum_{i=1}^{n}(t_i-t)^2K_h(t_i-t)\mathbf{v}_i\mathbf{v}_i^\top\big]=n\mathrm{E}(\mathbf{v}\mathbf{v}^\top)\mathrm{E}\big[(t_i-t)^2K_h(t_i-t)\big]=n\Omega_{\mathbf{v}}\mathrm{E}\big[(t_i-t)^2K_h(t_i-t)\big],$$

$$\mathrm{E}\big[(t_i-t)^2K_h(t_i-t)\big]=\frac{1}{h}\int(t_i-t)^2K\left(\frac{t_i-t}{h}\right)f(t_i)dt_i=h\int u^2K(u)f(t+hu)d(t+hu)$$
$$=h^2\mu_2 f(t)+O(h^4),$$

we have: $\mathrm{E}\big[n^{-1}\sum_{i=1}^{n}(t_i-t)^2K_h(t_i-t)\mathbf{v}_i\mathbf{v}_i^\top\big]=h^2\mu_2 f(t)\Omega_{\mathbf{v}}+O(h^4)$.

Applying similar calculations to other terms in (32) and use the formula for the inverse of $2\times 2$ block matrix square diagonal partition, we have:

$$[n^{-1}\tilde{\mathbf{V}}^\top(t)\mathbf{W}(t)\tilde{\mathbf{V}}(t)]^{-1}\xrightarrow{p}\begin{bmatrix}\mathbf{\Omega}_{\mathbf{z},\mathbf{v}}f(t)+O(h^2) & h^2\mu_2\dot{f}(t)\begin{pmatrix}\Omega_{\mathbf{zv}}\\\Omega_{\mathbf{v}}\end{pmatrix}+O(h^3)\\ h^2\mu_2\dot{f}(t)(\Omega_{\mathbf{vz}},\Omega_{\mathbf{v}})+O(h^3) & h^2\mu_2\Omega_{\mathbf{v}}f(t)+O(h^4)\end{bmatrix}^{-1}$$

$$=\begin{bmatrix}\mathbf{\Omega}_{\mathbf{z},\mathbf{v}}^{-1}f^{-1}(t)+O(h^2) & -f^{-2}(t)\dot{f}(t)\mathbf{\Omega}_{\mathbf{z},\mathbf{v}}^{-1}\begin{pmatrix}\Omega_{\mathbf{zv}}\\\Omega_{\mathbf{v}}\end{pmatrix}\Omega_{\mathbf{v}}^{-1}+O(h^2)\\ -f^{-2}(t)\dot{f}(t)\Omega_{\mathbf{v}}^{-1}(\Omega_{\mathbf{vz}},\Omega_{\mathbf{v}})\mathbf{\Omega}_{\mathbf{z},\mathbf{v}}^{-1}+O(h^2) & h^{-2}\mu_2^{-1}\Omega_{\mathbf{v}}^{-1}f(t)^{-1}+O(1)\end{bmatrix} \quad (33)$$

Similarly, for $n^{-1}\tilde{\mathbf{V}}^\top(t)\mathbf{W}(t)\begin{bmatrix}\mathbf{v}_1^\top(t_1-t)^2\mathrm{vec}(\ddot{\mathbf{A}}^*(t))\\\vdots\\\mathbf{v}_n^\top(t_n-t)^2\mathrm{vec}(\ddot{\mathbf{A}}^*(t))\end{bmatrix}$, we have:

$$n^{-1}\tilde{\mathbf{V}}^\top(t)\mathbf{W}(t)\begin{bmatrix}\mathbf{v}_1^\top(t_1-t)^2\mathrm{vec}(\ddot{\mathbf{A}}^*(t))\\\vdots\\\mathbf{v}_n^\top(t_n-t)^2\mathrm{vec}(\ddot{\mathbf{A}}^*(t))\end{bmatrix}=n^{-1}\begin{bmatrix}K_h(t_1-t)\mathbf{z}_1 & \cdots & K_h(t_n-t)\mathbf{z}_n\\ K_h(t_1-t)\mathbf{v}_1 & \cdots & K_h(t_n-t)\mathbf{v}_n\\ (t_1-t)K_h(t_1-t)\mathbf{v}_1 & \cdots & (t_n-t)K_h(t_n-t)\mathbf{v}_n\end{bmatrix}$$

$$\times\begin{bmatrix}\mathbf{v}_1^\top(t_1-t)^2\mathrm{vec}(\ddot{\mathbf{A}}^*(t))\\\vdots\\\mathbf{v}_n^\top(t_n-t)^2\mathrm{vec}(\ddot{\mathbf{A}}^*(t))\end{bmatrix}$$

$$=\begin{bmatrix}n^{-1}\sum_{i=1}^{n}(t_i-t)^2K_h(t_i-t)\mathbf{z}_i\mathbf{v}_i^\top\mathrm{vec}(\ddot{\mathbf{A}}^*(t))\\ n^{-1}\sum_{i=1}^{n}(t_i-t)^2K_h(t_i-t)\mathbf{v}_i\mathbf{v}_i^\top\mathrm{vec}(\ddot{\mathbf{A}}^*(t))\\ n^{-1}\sum_{i=1}^{n}(t_i-t)^3K_h(t_i-t)\mathbf{v}_i\mathbf{v}_i^\top\mathrm{vec}(\ddot{\mathbf{A}}^*(t))\end{bmatrix}$$

$$\xrightarrow{p}\begin{bmatrix}\mu_2h^2\Omega_{\mathbf{zv}}\mathrm{vec}(\ddot{\mathbf{A}}^*(t))f(t)\\ \mu_2h^2\Omega_{\mathbf{v}}\mathrm{vec}(\ddot{\mathbf{A}}^*(t))f(t)\\ \mu_4h^4\Omega_{\mathbf{v}}\mathrm{vec}(\ddot{\mathbf{A}}^*(t))\dot{f}(t)\end{bmatrix}. \quad (34)$$

Combining (33) and (34) we have

$$[\tilde{\mathbf{V}}^\top(t)\mathbf{W}(t)\tilde{\mathbf{V}}(t)]^{-1}\tilde{\mathbf{V}}^\top(t)\mathbf{W}(t)\begin{bmatrix} \mathbf{v}_1^\top(t_1-t)^2\text{vec}(\ddot{\mathbf{A}}^*(t)) \\ \vdots \\ \mathbf{v}_n^\top(t_n-t)^2\text{vec}(\ddot{\mathbf{A}}^*(t)) \end{bmatrix}$$

$$\xrightarrow{p} \begin{bmatrix} \begin{pmatrix} \mathbf{0}_{p_0} \\ \mu_2 h^2\text{vec}(\ddot{\mathbf{A}}^*(t)) \end{pmatrix} + O(h^4) \\ -\mu_2 h^2 f(t)^{-1}\dot{f}(t)\text{vec}(\ddot{\mathbf{A}}^*(t)) + h^2\mu_2^{-1}\mu_4 f(t)^{-1}\dot{f}(t)\text{vec}(\ddot{\mathbf{A}}^*(t)) + O(h^4) \end{bmatrix}$$

$$= \begin{bmatrix} \mathbf{0}_{p_0} \\ \mu_2 h^2\text{vec}(\ddot{\mathbf{A}}^*(t)) \\ -h^2 f(t)^{-1}\dot{f}(t)(\mu_2^{-1}\mu_4 - \mu_2)\text{vec}(\ddot{\mathbf{A}}^*(t)) \end{bmatrix} + O(h^4).$$

Similarly, using the fact that under Conditions (C1)-(C3), we have

$$\text{E}[(t_i-t)^3 K_h(t_i-t)] = \frac{1}{h}\int(t_i-t)^3 K\left(\frac{t_i-t}{h}\right)f(t_i)dt_i$$

$$= h^2\int u^3 K(u)f(t+hu)d(t+hu)$$

$$= h^4\mu_4\dot{f}(t) + O(h^5),$$

and that the variance of the remainder $R$ is of order:

$$\frac{1}{n}\text{E}[(t_i-t)^6 K_h^2(t_i-t)] = \frac{1}{nh^2}\int(t_i-t)^6 K^2\left(\frac{t_i-t}{h}\right)f(t_i)dt_i$$

$$= n^{-1}h^4\int u^6 K^2(u)f(t+hu)d(t+hu)$$

$$= O(n^{-1}h^5).$$

We have $R = O_p(h^4 + n^{-1/2}h^{5/2})$. Consequently, from the fact that $n^{-1/2}h^{5/2} = o(h^{-4})$ and (31) we have,

$$\begin{bmatrix} \hat{\boldsymbol{\beta}} - \boldsymbol{\beta}^* \\ \text{vec}(\hat{\mathbf{A}}(t)) - \text{vec}(\mathbf{A}^*(t)) \\ \text{vec}(\hat{\dot{\mathbf{A}}}(t)) - \text{vec}(\dot{\mathbf{A}}^*(t)) \end{bmatrix} - \frac{1}{2}\begin{bmatrix} \mathbf{0}_{p_0} \\ \mu_2 h^2\text{vec}(\ddot{\mathbf{A}}^*(t)) \\ -h^2 f(t)^{-1}\dot{f}(t)(\mu_2^{-1}\mu_4 - \mu_2)\text{vec}(\ddot{\mathbf{A}}^*(t)) \end{bmatrix}$$

$$= [\tilde{\mathbf{V}}^\top(t)\mathbf{W}(t)\tilde{\mathbf{V}}(t)]^{-1}\tilde{\mathbf{V}}^\top(t)\mathbf{W}(t)\boldsymbol{\epsilon} + O_p(h^4). \tag{35}$$

Note that,

$$\text{Var}\left\{[\tilde{\mathbf{V}}^\top(t)\mathbf{W}(t)\tilde{\mathbf{V}}(t)]^{-1}\tilde{\mathbf{V}}^\top(t)\mathbf{W}(t)\boldsymbol{\epsilon}\right\}$$

$$= \text{E}\sigma^2[\tilde{\mathbf{V}}^\top(t)\mathbf{W}(t)\tilde{\mathbf{V}}(t)]^{-1}\tilde{\mathbf{V}}^\top(t)\mathbf{W}^2(t)\tilde{\mathbf{V}}(t)[\tilde{\mathbf{V}}^\top(t)\mathbf{W}(t)\tilde{\mathbf{V}}(t)]^{-1} \tag{36}$$

We have already evaluated $[\tilde{\mathbf{V}}^\top(t)\mathbf{W}(t)\tilde{\mathbf{V}}(t)]^{-1}$ in (33). For $\tilde{\mathbf{V}}^\top(t)\mathbf{W}^2(t)\tilde{\mathbf{V}}(t)$, we have

$$\tilde{\mathbf{V}}^\top(t)\mathbf{W}^2(t)\tilde{\mathbf{V}}(t) \tag{37}$$

$$= \begin{bmatrix} \mathbf{z}_1 & \cdots & \mathbf{z}_n \\ \mathbf{v}_1 & \cdots & \mathbf{v}_n \\ (t_1 - t)\mathbf{v}_1 & \cdots & (t_n - t)\mathbf{v}_n \end{bmatrix} \begin{bmatrix} K_h^2(t_1 - t) & & \\ & \ddots & \\ & & K_h^2(t_n - t) \end{bmatrix} \begin{bmatrix} \mathbf{z}_1^\top & \mathbf{v}_1^\top & (t_1 - t)\mathbf{v}_1^\top \\ \vdots & \vdots & \vdots \\ \mathbf{z}_n^\top & \mathbf{v}_n^\top & (t_n - t)\mathbf{v}_n^\top \end{bmatrix}$$

$$= \begin{bmatrix} \sum_{i=1}^n K_h^2(t_i - t)\mathbf{z}_i\mathbf{z}_i^\top & \sum_{i=1}^n K_h^2(t_i - t)\mathbf{z}_i\mathbf{v}_i^\top & \sum_{i=1}^n (t_i - t)K_h^2(t_i - t)\mathbf{z}_i\mathbf{v}_i^\top \\ \sum_{i=1}^n K_h^2(t_i - t)\mathbf{v}_i\mathbf{z}_i^\top & \sum_{i=1}^n K_h^2(t_i - t)\mathbf{v}_i\mathbf{v}_i^\top & \sum_{i=1}^n (t_i - t)K_h^2(t_i - t)\mathbf{v}_i\mathbf{v}_i^\top \\ \sum_{i=1}^n (t_i - t)K_h^2(t_i - t)\mathbf{v}_i\mathbf{z}_i^\top & \sum_{i=1}^n (t_i - t)K_h^2(t_i - t)\mathbf{v}_i\mathbf{v}_i^\top & \sum_{i=1}^n (t_i - t)^2 K_h^2(t_i - t)\mathbf{v}_i\mathbf{v}_i^\top \end{bmatrix}$$

Similar to (32), we focus on $\sum_{i=1}^n K_h^2(t_i - t)\mathbf{v}_i\mathbf{v}_i^\top$, $\sum_{i=1}^n (t_i - t)K_h^2(t_i - t)\mathbf{v}_i\mathbf{v}_i^\top$ and $\sum_{i=1}^n (t_i - t)^2 K_h^2(t_i - t)\mathbf{v}_i\mathbf{v}_i^\top$, and other entries in (37) can be derived similarly. Note that

$$\mathrm{E}\Big[n^{-1}\sum_{i=1}^n K_h^2(t_i - t)\mathbf{v}_i\mathbf{v}_i^\top\Big] = \Omega_\mathbf{v}\mathrm{E}\big[K_h^2(t_i - t)\big],$$

and

$$\mathrm{E}[K_h^2(t_i - t)] = \frac{1}{h^2}\int K^2\Big(\frac{t_i - t}{h}\Big)f(t_i)dt_i$$
$$= \frac{1}{h^2}\int K^2(u)f(t + hu)d(t + hu)$$
$$= \frac{1}{h}f(t)\nu_0 + O(h).$$

Therefore we have, $n^{-1}\sum_{i=1}^n K_h^2(t_i - t)\mathbf{v}_i\mathbf{v}_i^\top \xrightarrow{p} \frac{1}{h}f(t)\nu_0\Omega_\mathbf{v} + O(h)$. Similarly, from the facts that

$$\mathrm{E}\Big[n^{-1}\sum_{i=1}^n (t_i - t)K_h^2(t_i - t)\mathbf{v}_i\mathbf{v}_i^\top\Big] = \Omega_\mathbf{v}\mathrm{E}\big[(t_i - t)K_h^2(t_i - t)\big],$$

and

$$\mathrm{E}\big[(t_i - t)K_h^2(t_i - t)\big] = \frac{1}{h^2}\int (t_i - t)K^2\Big(\frac{t_i - t}{h}\Big)f(t_i)dt_i$$
$$= \frac{1}{h}\int uK^2(u)f(t + hu)d(t + hu)$$
$$= h\dot{f}(t)\nu_2 + O(h^2),$$

we have, $n^{-1}\sum_{i=1}^n (t_i - t)K_h^2(t_i - t)\mathbf{v}_i\mathbf{v}_i^\top \xrightarrow{p} h\nu_2\dot{f}(t)\Omega_\mathbf{v} + O(h^2)$. Similarly, from

$$\mathrm{E}\Big[n^{-1}\sum_{i=1}^n (t_i - t)^2 K_h^2(t_i - t)\mathbf{v}_i\mathbf{v}_i^\top\Big] = \Omega_\mathbf{v}\mathrm{E}\big[(t_i - t)^2 K_h^2(t_i - t)\big],$$

and

$$\mathrm{E}[(t_i - t)^2 K_h^2(t_i - t)] = \frac{1}{h^2}\int (t_i - t)^2 K^2\Big(\frac{t_i - t}{h}\Big)f(t_i)dt_i$$

$$= \int u^2 K^2(u)f(t + hu)d(t + hu)$$

$$= h\nu_2 f(t) + O(h^3),$$

we have, $n^{-1}\sum_{i=1}^n (t_i - t)^2 K_h^2(t_i - t)\mathbf{v}_i\mathbf{v}_i^\top \xrightarrow{p} h\nu_2 f(t)\Omega_{\mathbf{v}} + O(h^3)$.

Summarizing above, we can obtain

$$n^{-1}\tilde{\mathbf{V}}^\top(t)\mathbf{W}^2(t)\tilde{\mathbf{V}}(t) \xrightarrow{p} \begin{bmatrix} h^{-1}f(t)\nu_0\mathbf{\Omega}_{\mathbf{z},\mathbf{v}} + O(h) & h\nu_2\dot{f}(t)\begin{pmatrix}\Omega_{\mathbf{zv}}\\ \Omega_{\mathbf{v}}\end{pmatrix} + O(h^2) \\ h\nu_2\dot{f}(t)(\Omega_{\mathbf{vz}}, \Omega_{\mathbf{v}}) + O(h^2) & h\nu_2\Omega_{\mathbf{v}}f(t) + O(h^3) \end{bmatrix}$$

Together with (35) and (36) we have:

$$\mathrm{Var}\left\{\sqrt{nh}\begin{bmatrix}\hat{\boldsymbol{\beta}} - \boldsymbol{\beta}^* \\ \mathrm{vec}(\hat{\mathbf{A}}(t)) - \mathrm{vec}(\mathbf{A}^*(t)) \\ \mathrm{vec}(\hat{\dot{\mathbf{A}}}(t)) - \mathrm{vec}(\dot{\mathbf{A}}^*(t))\end{bmatrix}\right\}$$

$$= \sigma^2 \begin{bmatrix} \mathbf{\Omega}_{\mathbf{z},\mathbf{v}}^{-1}f(t)^{-1} + O(h^2) & -f(t)^{-2}\dot{f}(t)\mathbf{\Omega}_{\mathbf{z},\mathbf{v}}^{-1}\begin{pmatrix}\Omega_{\mathbf{zv}}\\ \Omega_{\mathbf{v}}\end{pmatrix}\Omega_{\mathbf{v}}^{-1} + O(h^2) \\ -f(t)^{-2}\dot{f}(t)\Omega_{\mathbf{v}}^{-1}(\Omega_{\mathbf{vz}}, \Omega_{\mathbf{v}})\mathbf{\Omega}_{\mathbf{z},\mathbf{v}}^{-1} + O(h^2) & h^{-2}\mu_2^{-1}\Omega_{\mathbf{v}}^{-1}f(t)^{-1} + O(1) \end{bmatrix}$$

$$\times \begin{bmatrix} f(t)\nu_0\mathbf{\Omega}_{\mathbf{z},\mathbf{v}} + O(h^2) & h^2\nu_2\dot{f}(t)\begin{pmatrix}\Omega_{\mathbf{zv}}\\ \Omega_{\mathbf{v}}\end{pmatrix} + O(h^3) \\ h^2\nu_2\dot{f}(t)(\Omega_{\mathbf{vz}}, \Omega_{\mathbf{v}}) + O(h^3) & h^2\nu_2\Omega_{\mathbf{v}}f(t) + O(h^4) \end{bmatrix}$$

$$\times \begin{bmatrix} \mathbf{\Omega}_{\mathbf{z},\mathbf{v}}^{-1}f(t)^{-1} + O(h^2) & -f(t)^{-2}\dot{f}(t)\mathbf{\Omega}_{\mathbf{z},\mathbf{v}}^{-1}\begin{pmatrix}\Omega_{\mathbf{zv}}\\ \Omega_{\mathbf{v}}\end{pmatrix}\Omega_{\mathbf{v}}^{-1} + O(h^2) \\ -f(t)^{-2}\dot{f}(t)\Omega_{\mathbf{v}}^{-1}(\Omega_{\mathbf{vz}}, \Omega_{\mathbf{v}})\mathbf{\Omega}_{\mathbf{z},\mathbf{v}}^{-1} + O(h^2) & h^{-2}\mu_2^{-1}\Omega_{\mathbf{v}}^{-1}f(t)^{-1} + O(1) \end{bmatrix}$$

$$= \sigma^2 \begin{bmatrix} \nu_0\mathbf{I} + O(h^2) & O(h^3) \\ (\mu_2^{-1}\nu_2 - \nu_0)f(t)^{-1}\dot{f}(t)\Omega_{\mathbf{v}}^{-1}(\Omega_{\mathbf{vz}}, \Omega_{\mathbf{v}}) + O(h^2) & \mu_2^{-1}\nu_2\mathbf{I} + O(h^2) \end{bmatrix}$$

$$\times \begin{bmatrix} \mathbf{\Omega}_{\mathbf{z},\mathbf{v}}^{-1}f(t)^{-1} + O(h^2) & -f(t)^{-2}\dot{f}(t)\mathbf{\Omega}_{\mathbf{z},\mathbf{v}}^{-1}\begin{pmatrix}\Omega_{\mathbf{zv}}\\ \Omega_{\mathbf{v}}\end{pmatrix}\Omega_{\mathbf{v}}^{-1} + O(h^2) \\ -f(t)^{-2}\dot{f}(t)\Omega_{\mathbf{v}}^{-1}(\Omega_{\mathbf{vz}}, \Omega_{\mathbf{v}})\mathbf{\Omega}_{\mathbf{z},\mathbf{v}}^{-1} + O(h^2) & h^{-2}\mu_2^{-1}\Omega_{\mathbf{v}}^{-1}f(t)^{-1} + O(1) \end{bmatrix}$$

$$= \sigma^2 \begin{bmatrix} \nu_0\mathbf{\Omega}_{\mathbf{z},\mathbf{v}}^{-1}f(t)^{-1} + O(h^2) & -\nu_0 f(t)^{-2}\dot{f}(t)\mathbf{\Omega}_{\mathbf{z},\mathbf{v}}^{-1}\begin{pmatrix}\Omega_{\mathbf{zv}}\\ \Omega_{\mathbf{v}}\end{pmatrix}\Omega_{\mathbf{v}}^{-1} + O(h^2) \\ -\nu_0 f(t)^{-2}\dot{f}(t)\Omega_{\mathbf{v}}^{-1}(\Omega_{\mathbf{vz}}, \Omega_{\mathbf{v}})\mathbf{\Omega}_{\mathbf{z},\mathbf{v}}^{-1} + O(h^2) & h^{-2}f(t)^{-1}\mu_2^{-1}\nu_2\Omega_{\mathbf{v}}^{-1} + O(1) \end{bmatrix}.$$

$$(38)$$

The first statement in Theorem 1 can then be concluded from (35) and (38), and the second statement in Theorem 1 can be easily derived by applying the formula for the inverse of a $2 \times 2$ block matrix to $\mathbf{\Omega}_{\mathbf{z},\mathbf{v}}^{-1}$.

**Lemma 1** *Let $\{X_i, Y_i\}_{i=1}^n$ be i.i.d. samples such that $\mathrm{E}|Y_i|^s < \infty$ and $\sup_x \int |y|^s f(x, y)dy < \infty$*

*for some $s > 2$, where $f$ is the joint density of $(X, Y)$. Let $K$ be a bounded function with a compact support, satisfying a Lipschitz condition. Given that $n^{2d-1}h \to \infty$ for some $d < 1 - s^{-1}$, we have*

$$\sup_x \left| n^{-1} \sum_{i=1}^n [K_h(X_i - x)Y_i - \mathrm{E}\{K_h(X_i - x)Y_i\}] \right| = O_p\left(\left\{\frac{\log(1/h)}{nh}\right\}^{1/2}\right).$$

**Proof of Lemma 1**: This proof can be attained from the results in Mack and Silverman (1982) or using a maximal inequality in empirical process theory Pollard (1991). See Stone (1982) for a detailed discussion on uniform convergence rates for non-parametric regression.

**Proof of Theorem 2**: In (12), our target is to use all local estimate $\hat{\mathbf{A}}(t_i)$, $i = 1, \cdots, n$ to obtain a global estimate $\hat{\boldsymbol{\beta}}$. From the independence of $t_i$ and $\mathbf{X}_i^*$, $\mathbf{z}_i$, Condition (C5), the assumption that $nh^4 \to 0$ and Lemma 1, it can be shown that

$$\sup_{t \in [0,1]} \left| \mathrm{vec}(\hat{\mathbf{A}}(t)) - \mathrm{vec}(\mathbf{A}^*(t)) \right| = O_p\left(\left\{\frac{\log(1/h)}{nh}\right\}^{1/2}\right). \tag{39}$$

By (12), we can compute our global estimate through

$$
\begin{aligned}
\hat{\boldsymbol{\beta}}^\dagger &= [\mathbf{Z}^\top \mathbf{Z}]^{-1} \mathbf{Z}^\top \begin{bmatrix} y_1 - \mathbf{v}_1^\top \mathrm{vec}(\hat{\mathbf{A}}(t_1)) \\ \vdots \\ y_n - \mathbf{v}_n^\top \mathrm{vec}(\hat{\mathbf{A}}(t_n)) \end{bmatrix} \\
&= [\mathbf{Z}^\top \mathbf{Z}]^{-1} \mathbf{Z}^\top \begin{bmatrix} \mathbf{v}_1^\top \mathrm{vec}(\mathbf{A}^*(t_1)) + \mathbf{z}_1^\top \boldsymbol{\beta} + \epsilon_1 - \mathbf{v}_1^\top \mathrm{vec}(\hat{\mathbf{A}}(t_1)) \\ \vdots \\ \mathbf{v}_n^\top \mathrm{vec}(\mathbf{A}^*(t_n)) + \mathbf{z}_n^\top \boldsymbol{\beta} + \epsilon_n - \mathbf{v}_n^\top \mathrm{vec}(\hat{\mathbf{A}}(t_n)) \end{bmatrix} \\
&= [\mathbf{Z}^\top \mathbf{Z}]^{-1} \mathbf{Z}^\top \begin{bmatrix} \mathbf{v}_1^\top [\mathrm{vec}(\mathbf{A}^*(t_1)) - \mathrm{vec}(\hat{\mathbf{A}}(t_1))] \\ \vdots \\ \mathbf{v}_n^\top [\mathrm{vec}(\mathbf{A}^*(t_n)) - \mathrm{vec}(\hat{\mathbf{A}}(t_n))] \end{bmatrix} + \boldsymbol{\beta} + [\mathbf{Z}^\top \mathbf{Z}]^{-1} \mathbf{Z}^\top \boldsymbol{\epsilon}.
\end{aligned}
$$

So we have

$$\hat{\boldsymbol{\beta}}^\dagger - \boldsymbol{\beta} = [\mathbf{Z}^\top \mathbf{Z}]^{-1} \mathbf{Z}^\top \begin{bmatrix} \mathbf{v}_1^\top [\mathrm{vec}(\mathbf{A}^*(t_1)) - \mathrm{vec}(\hat{\mathbf{A}}(t_1))] \\ \vdots \\ \mathbf{v}_n^\top [\mathrm{vec}(\mathbf{A}^*(t_n)) - \mathrm{vec}(\hat{\mathbf{A}}(t_n))] \end{bmatrix} + [\mathbf{Z}^\top \mathbf{Z}]^{-1} \mathbf{Z}^\top \boldsymbol{\epsilon}.$$

For the first term on the right hand side of the above equation, we shall establish its asymptotic property using a cross-validation arguments. Specifically, suppose the index $\{1, 2, \ldots, n\}$ are divided into $h^{-2}$ non-overlapping subsets with size $nh^2$. For simplicity we shall assume that $D := h^{-2}$ and $nh^2$ are integers and denoted the $D$ disjoint set as $\mathcal{S}_1, \ldots, \mathcal{S}_D$. Further, we denote $\mathcal{S}_{-d} := \{1, 2, \ldots, n\}/\mathcal{S}_d$, and for any index set $\mathcal{S}$ and a given matrix $\mathbf{M}$, we use $\mathbf{M}_{\mathcal{S}}$ to denote the submatrix of $\mathbf{M}$ by keeping the rows indexed by the index set $\mathcal{S}$ only. For $d \in \{1, 2, \ldots, D\}$, let $\mathrm{vec}(\hat{\mathbf{A}}^{(-d)}(t))$ be the local least squares estimator obtained by leaving the samples indexed by $S_d$ out. It is easy to see that $\mathrm{vec}(\hat{\mathbf{A}}^{(-d)}(t)) - \mathrm{vec}(\hat{\mathbf{A}}(t)) = O_p(h^2)$ for any $t \in [0, 1]$. To show

this, we apply Neumann series on $\hat{\boldsymbol{\theta}}^{(-d)}$

$$
\begin{aligned}
\hat{\boldsymbol{\theta}}^{(-d)} =& \Big[\sum_{i\notin\mathcal{S}_d} K_h(t_i-t)\tilde{\mathbf{v}}_i\tilde{\mathbf{v}}_i^\top\Big]^{-1}\Big[\sum_{i\notin\mathcal{S}_d} K_h(t_i-t)\tilde{\mathbf{v}}_i y_i\Big] \\
=& \Big[\sum_{i=1}^{n} K_h(t_i-t)\tilde{\mathbf{v}}_i\tilde{\mathbf{v}}_i^\top - \sum_{i\in\mathcal{S}_d} K_h(t_i-t)\tilde{\mathbf{v}}_i\tilde{\mathbf{v}}_i^\top\Big]^{-1}\Big[\sum_{i\notin\mathcal{S}_d} K_h(t_i-t)\tilde{\mathbf{v}}_i y_i\Big] \\
=& \sum_{p=0}^{\infty}\Big[\big[\sum_{i=1}^{n} K_h(t_i-t)\tilde{\mathbf{v}}_i\tilde{\mathbf{v}}_i^\top\big]^{-1}\big[\sum_{i\in\mathcal{S}_d} K_h(t_i-t)\tilde{\mathbf{v}}_i\tilde{\mathbf{v}}_i^\top\big]\Big]^{p} \\
& \times \Big[\sum_{i=1}^{n} K_h(t_i-t)\tilde{\mathbf{v}}_i\tilde{\mathbf{v}}_i^\top\Big]^{-1}\Big[\sum_{i\notin\mathcal{S}_d} K_h(t_i-t)\tilde{\mathbf{v}}_i y_i\Big]
\end{aligned}
$$

Given the result in (33), we have

$$
\Big[\sum_{i=1}^{n} K_h(t_i-t)\tilde{\mathbf{v}}_i\tilde{\mathbf{v}}_i^\top\Big]^{-1}\Big[\sum_{i\in\mathcal{S}_d} K_h(t_i-t)\tilde{\mathbf{v}}_i\tilde{\mathbf{v}}_i^\top\Big]
$$

$$
\xrightarrow{p} n^{-1}
\begin{bmatrix}
\boldsymbol{\Omega}_{\mathbf{z,v}}^{-1}f^{-1}(t)+O(h^2) & -f^{-2}(t)\dot{f}(t)\boldsymbol{\Omega}_{\mathbf{z,v}}^{-1}\begin{pmatrix}\boldsymbol{\Omega}_{\mathbf{zv}}\\ \boldsymbol{\Omega}_{\mathbf{v}}\end{pmatrix}\boldsymbol{\Omega}_{\mathbf{v}}^{-1}+O(h^2) \\
-f^{-2}(t)\dot{f}(t)\boldsymbol{\Omega}_{\mathbf{v}}^{-1}(\boldsymbol{\Omega}_{\mathbf{vz}},\boldsymbol{\Omega}_{\mathbf{v}})\boldsymbol{\Omega}_{\mathbf{z,v}}^{-1}+O(h^2) & h^{-2}\mu_2^{-1}\boldsymbol{\Omega}_{\mathbf{v}}^{-1}f(t)^{-1}+O(1).
\end{bmatrix}
$$

$$
\times\, nh^2
\begin{bmatrix}
\boldsymbol{\Omega}_{\mathbf{z,v}}f(t)+O(h^2) & h^2\mu_2\dot{f}(t)\begin{pmatrix}\boldsymbol{\Omega}_{\mathbf{zv}}\\ \boldsymbol{\Omega}_{\mathbf{v}}\end{pmatrix}+O(h^3) \\
h^2\mu_2\dot{f}(t)(\boldsymbol{\Omega}_{\mathbf{vz}},\boldsymbol{\Omega}_{\mathbf{v}})+O(h^3) & h^2\mu_2\boldsymbol{\Omega}_{\mathbf{v}}f(t)+O(h^4)
\end{bmatrix}
$$

$$
= h^2\mathbf{I}_{2RS+p_0}+O(h^4).
$$

And

$$
\begin{aligned}
& \Big[\sum_{i=1}^{n} K_h(t_i-t)\tilde{\mathbf{v}}_i\tilde{\mathbf{v}}_i^\top\Big]^{-1}\Big[\sum_{i\notin\mathcal{S}_d} K_h(t_i-t)\tilde{\mathbf{v}}_i y_i\Big] \\
=& \Big[\sum_{i=1}^{n} K_h(t_i-t)\tilde{\mathbf{v}}_i\tilde{\mathbf{v}}_i^\top\Big]^{-1}\Big[\sum_{i=1}^{n} K_h(t_i-t)\tilde{\mathbf{v}}_i y_i - \sum_{i\notin\mathcal{S}_d} K_h(t_i-t)\tilde{\mathbf{v}}_i y_i\Big] \\
\xrightarrow{p}& \Big[\sum_{i=1}^{n} K_h(t_i-t)\tilde{\mathbf{v}}_i\tilde{\mathbf{v}}_i^\top\Big]^{-1}\Big[\sum_{i=1}^{n} K_h(t_i-t)\tilde{\mathbf{v}}_i y_i\Big](1+O(h^2)) \\
=& \hat{\boldsymbol{\theta}}+O(h^2).
\end{aligned}
$$

So $\hat{\boldsymbol{\theta}}^{(-d)} = \sum_{p=0}^{\infty}[h^2\mathbf{I}_{2RS+p_0}+O_p(h^4)]^p[\hat{\boldsymbol{\theta}}+O_p(h^2)] = \hat{\boldsymbol{\theta}}+O_p(h^2)$, and thus $\text{vec}(\hat{\mathbf{A}}^{(-d)}(t)) = (\mathbf{0}_{RS\times p_0},\mathbf{I}_{RS},\mathbf{0}_{RS\times RS})\hat{\boldsymbol{\theta}} = \text{vec}(\hat{\mathbf{A}}(t))+O_p(h^2)$.

Therefore, we have,

$$
[\mathbf{Z}^\top\mathbf{Z}]^{-1}\mathbf{Z}^\top
\begin{bmatrix}
\mathbf{v}_1^\top[\text{vec}(\mathbf{A}^*(t_1)) - \text{vec}(\hat{\mathbf{A}}(t_1))] \\
\vdots \\
\mathbf{v}_n^\top[\text{vec}(\mathbf{A}^*(t_n)) - \text{vec}(\hat{\mathbf{A}}(t_n))]
\end{bmatrix}
$$

$$
= \sum_{d=1}^{D} [\mathbf{Z}^\top\mathbf{Z}]^{-1}\mathbf{Z}_{\mathcal{S}_d}^\top
\begin{bmatrix}
\mathbf{v}_1^\top[\text{vec}(\mathbf{A}^*(t_1)) - \text{vec}(\hat{\mathbf{A}}^{(-d)}(t_1))] + \mathbf{v}_1^\top[\text{vec}(\hat{\mathbf{A}}^{(-d)}(t_1)) - \text{vec}(\hat{\mathbf{A}}(t_1))] \\
\vdots \\
\mathbf{v}_n^\top[\text{vec}(\mathbf{A}^*(t_n)) - \text{vec}(\hat{\mathbf{A}}^{(-d)}(t_n)) + \mathbf{v}_n^\top[\text{vec}(\hat{\mathbf{A}}^{(-d)}(t_n)) - \text{vec}(\hat{\mathbf{A}}(t_n))]
\end{bmatrix}_{\mathcal{S}_d}
$$

$$
= \sum_{d=1}^{D} [\mathbf{Z}^\top\mathbf{Z}]^{-1}\mathbf{Z}_{\mathcal{S}_d}^\top
\begin{bmatrix}
\mathbf{v}_1^\top[\text{vec}(\mathbf{A}(t_1)) - \text{vec}(\hat{\mathbf{A}}^{(-d)}(t_1))] \\
\vdots \\
\mathbf{v}_n^\top[\text{vec}(\mathbf{A}(t_n)) - \text{vec}(\hat{\mathbf{A}}^{(-d)}(t_n))]
\end{bmatrix}_{\mathcal{S}_d}
+ O_p(h^2).
$$

Notice that given $t_i$, $\mathbf{z}_i$ and $\mathbf{v}_i$, for $i \in \mathcal{S}_{-d}$, $\{\text{vec}(\hat{\mathbf{A}}^{(-d)}(t_j)), j \in \mathcal{S}_d\}$ are conditionally independent. By conditional on $t_i$, $\mathbf{z}_i$ and $\mathbf{v}_i$, for $i \in S_{-d}$ first, and applying the classical concentration inequality on $\frac{1}{n/D}\mathbf{Z}_{\mathcal{S}_d}^\top \begin{bmatrix} \mathbf{v}_1^\top[\text{vec}(\mathbf{A}^*(t_1)) - \text{vec}(\hat{\mathbf{A}}^{(-d)}(t_1))] \\ \vdots \\ \mathbf{v}_n^\top[\text{vec}(\mathbf{A}^*(t_n)) - \text{vec}(\hat{\mathbf{A}}^{(-d)}(t_n))] \end{bmatrix}_{\mathcal{S}_d}$, and together with the bound in (39), we have, there exists a large enough constant $C > 0$ such that, with probability larger than $1 - O(n^{-1})$,

$$
\frac{1}{n/D}\mathbf{Z}_{\mathcal{S}_d}^\top
\begin{bmatrix}
\mathbf{v}_1^\top[\text{vec}(\mathbf{A}^*(t_1)) - \text{vec}(\hat{\mathbf{A}}(t_1))] \\
\vdots \\
\mathbf{v}_n^\top[\text{vec}(\mathbf{A}^*(t_n)) - \text{vec}(\hat{\mathbf{A}}(t_n))]
\end{bmatrix}_{\mathcal{S}_d}
\leq C\left(h^2 + \sqrt{\frac{1}{nh}}\sqrt{\frac{\log n}{nh^2}}\right).
$$

Here in the last step we have used the facts that $\text{E}[\text{vec}(\mathbf{A}(t)) - \text{vec}(\hat{\mathbf{A}}^{(-d)}(t))|t] = O(h^2)$, and $\text{Var}[\text{vec}(\mathbf{A}(t)) - \text{vec}(\hat{\mathbf{A}}^{(-d)}(t))|t] = O((nh)^{-1})$. Consequently, we have

$$
\sqrt{n}[\mathbf{Z}^\top\mathbf{Z}]^{-1}\mathbf{Z}^\top
\begin{bmatrix}
\mathbf{v}_1^\top[\text{vec}(\mathbf{A}^*(t_1)) - \text{vec}(\hat{\mathbf{A}}(t_1))] \\
\vdots \\
\mathbf{v}_n^\top[\text{vec}(\mathbf{A}^*(t_n)) - \text{vec}(\hat{\mathbf{A}}(t_n))]
\end{bmatrix}
= O_p\left(\sqrt{n}h^2 + \sqrt{\frac{1}{h}}\sqrt{\frac{\log n}{nh^2}}\right) = o_p(1).
$$

As a result, Theorem 2 can be concluded by Slutsky's Theorem and the fact that

$$
\text{Var}\left\{\sqrt{n}[\mathbf{Z}^\top\mathbf{Z}]^{-1}\mathbf{Z}^\top\boldsymbol{\epsilon}\right\} = n\sigma^2\text{E}[\mathbf{Z}^\top\mathbf{Z}]^{-1}\mathbf{Z}^\top\mathbf{Z}[\mathbf{Z}^\top\mathbf{Z}]^{-1} = \sigma^2\text{E}[n^{-1}\sum_{i=1}^{n}\mathbf{z}_i\mathbf{z}_i^\top]^{-1} \to \frac{\sigma^2}{n}\Omega_{\mathbf{z}}^{-1}.
$$

## 7.2  Asymptotics of the penalized functions and parameters

**Proof of theorem 3**: Let $\delta_n = k_n^{-3/2} + \max\{n^{\tau/2}, k_n^{1/2}\}k_n^{1/2}\sqrt{\frac{\log n}{n}}$, $\mathcal{G} = \mathcal{G}^* + \delta_n\boldsymbol{\Delta}_{\mathcal{G}}$, $\boldsymbol{\beta} = \boldsymbol{\beta}^* + \delta_n\boldsymbol{\Delta}_{\boldsymbol{\beta}}$, where $\boldsymbol{\Delta}_{\mathcal{G}} \in \mathbb{R}^{L\times R\times S}$ and $\boldsymbol{\Delta}_{\boldsymbol{\beta}} \in \mathbb{R}^{p_0}$. Denote $\tilde{\boldsymbol{\Delta}} = (\text{vec}(\boldsymbol{\Delta}_{\mathcal{G}})^\top, k_n^{1/2}\boldsymbol{\Delta}_{\boldsymbol{\beta}}^\top)^\top$. Let $L(\mathcal{G}, \boldsymbol{\beta})$ be the penalized squared loss defined as in (18). By Fan and Li (2001), it suffices to

show that for any arbitrary small $\epsilon > 0$, there exists a large constant $C$ such that

$$P\Big\{ \inf_{\|\tilde{\Delta}\|=C} L(\mathcal{G}, \boldsymbol{\beta}) > L(\mathcal{G}^*, \boldsymbol{\beta}^*)\Big\} \geq 1 - \epsilon. \tag{40}$$

This implies that with probability at least $1 - \epsilon$, there is a local minimum $(\hat{\mathcal{G}}^{\ddagger}, \hat{\boldsymbol{\beta}}^{\ddagger})$ in the ball $\{(\mathcal{G}, \boldsymbol{\beta}) = (\mathcal{G}^* + \delta_n \boldsymbol{\Delta}_{\mathcal{G}}, \boldsymbol{\beta}^* + \delta_n \boldsymbol{\Delta}_{\boldsymbol{\beta}}) : \tilde{\boldsymbol{\Delta}} = (\text{vec}(\boldsymbol{\Delta}_{\mathcal{G}})^{\top}, k_n^{1/2}\boldsymbol{\Delta}_{\boldsymbol{\beta}}^{\top})^{\top}, \|\tilde{\boldsymbol{\Delta}}\| \leq C\}$. That is, there exists a local minimizer such that $\|\hat{\mathcal{G}}^{\ddagger} - \mathcal{G}^*\| = O_p(\delta_n), \|\hat{\boldsymbol{\beta}}^{\ddagger} - \boldsymbol{\beta}^*\| = O_p(k_n^{-1/2}\delta_n)$. Notice that

$$L(\mathcal{G}, \boldsymbol{\beta}) - L(\mathcal{G}^*, \boldsymbol{\beta}^*) = L(\mathcal{G}, \boldsymbol{\beta}) - L(\mathcal{G}, \boldsymbol{\beta}^*) + L(\mathcal{G}, \boldsymbol{\beta}^*) - L(\mathcal{G}^*, \boldsymbol{\beta}^*).$$

To prove (40), we shall evaluate $L(\mathcal{G}, \boldsymbol{\beta}) - L(\mathcal{G}, \boldsymbol{\beta}^*)$ and $L(\mathcal{G}, \boldsymbol{\beta}^*) - L(\mathcal{G}^*, \boldsymbol{\beta}^*)$ separately first.

Recall that we denote $\mathbf{u}_i = \text{vec}(\mathcal{H}_i(t_i)) = (h_{ilrs}(t_i)) \in \mathbb{R}^{LRS}$, $\tilde{\mathbf{u}}_i = (\text{vec}(\mathcal{H}_i(t_i))^{\top}, \mathbf{z}_i^{\top})^{\top} \in \mathbb{R}^{LRS+p_0}$. We have

$$L(\mathcal{G}, \boldsymbol{\beta}) - L(\mathcal{G}, \boldsymbol{\beta}^*)$$
$$= \frac{1}{2}\sum_{i=1}^{n} \Big\{ [y_i - \langle \mathcal{H}_i(t_i), \mathcal{G} \rangle - \mathbf{z}_i^{\top}\boldsymbol{\beta}]^2 - [y_i - \langle \mathcal{H}_i(t_i), \mathcal{G} \rangle - \mathbf{z}_i^{\top}\boldsymbol{\beta}^*]^2 \Big\}$$
$$+ n\sum_{k=1}^{p_0} \{P_{\omega_{\beta}}(|\beta_k|) - P_{\omega_{\beta}}(|\beta_k^*|)\}$$
$$= \frac{1}{2}\delta_n^2 \sum_{i=1}^{n} \mathbf{z}_i^{\top}\boldsymbol{\Delta}_{\boldsymbol{\beta}}\boldsymbol{\Delta}_{\boldsymbol{\beta}}^{\top}\mathbf{z}_i - \delta_n \sum_{i=1}^{n} \mathbf{z}_i^{\top}\boldsymbol{\Delta}_{\boldsymbol{\beta}}[y_i - \langle \mathcal{H}_i(t_i), \mathcal{G} \rangle - \mathbf{z}_i^{\top}\boldsymbol{\beta}^*]$$
$$+ n\sum_{k=1}^{p_0} \{P_{\omega_{\beta}}(|\beta_k|) - P_{\omega_{\beta}}(|\beta_k^*|)\}$$
$$=: I_1 + I_2 + I_3.$$

Similarly,

$$L(\mathcal{G}, \boldsymbol{\beta}^*) - L(\mathcal{G}^*, \boldsymbol{\beta}^*)$$
$$= \frac{1}{2}\sum_{i=1}^{n} \Big\{ [y_i - \langle \mathcal{H}_i(t_i), \mathcal{G} \rangle - \mathbf{z}_i^{\top}\boldsymbol{\beta}^*]^2 - [y_i - \langle \mathcal{H}_i(t_i), \mathcal{G}^* \rangle - \mathbf{z}_i^{\top}\boldsymbol{\beta}^*]^2 \Big\}$$
$$+ n\sum_{r=1}^{R}\sum_{s=1}^{S} \{P_{\omega_{\gamma}^c}(\|\boldsymbol{\gamma}_{rs}\|_{\mathcal{B}_1}) - P_{\omega_{\gamma}^c}(\|\boldsymbol{\gamma}_{rs}^*\|_{\mathcal{B}_1})\} + n\sum_{r=1}^{R}\sum_{s=1}^{S} \{P_{\omega_{\gamma}^v}(\|\boldsymbol{\gamma}_{rs}\|_{\mathcal{B}_2}) - P_{\omega_{\gamma}^v}(\|\boldsymbol{\gamma}_{rs}^*\|_{\mathcal{B}_2})\}$$
$$= \frac{1}{2}\sum_{i=1}^{n} \Big\{ [y_i - \langle \mathcal{H}_i(t_i), \mathcal{G}^* \rangle - \mathbf{z}_i^{\top}\boldsymbol{\beta}^* - \delta_n \mathbf{u}_i^{\top}\boldsymbol{\Delta}_{\mathcal{G}}]^2 - [y_i - \langle \mathcal{H}_i(t_i), \mathcal{G}^* \rangle - \mathbf{z}_i^{\top}\boldsymbol{\beta}^*]^2 \Big\}$$
$$+ n\sum_{r=1}^{R}\sum_{s=1}^{S} \{P_{\omega_{\gamma}^c}(\|\boldsymbol{\gamma}_{rs}\|_{\mathcal{B}_1}) - P_{\omega_{\gamma}^c}(\|\boldsymbol{\gamma}_{rs}^*\|_{\mathcal{B}_1})\} + n\sum_{r=1}^{R}\sum_{s=1}^{S} \{P_{\omega_{\gamma}^v}(\|\boldsymbol{\gamma}_{rs}\|_{\mathcal{B}_2}) - P_{\omega_{\gamma}^v}(\|\boldsymbol{\gamma}_{rs}^*\|_{\mathcal{B}_2})\}$$
$$= \frac{1}{2}\delta_n^2 \sum_{i=1}^{n} \mathbf{u}_i^{\top}\boldsymbol{\Delta}_{\mathcal{G}}\boldsymbol{\Delta}_{\mathcal{G}}^{\top}\mathbf{u}_i - \delta_n \sum_{i=1}^{n} \mathbf{u}_i^{\top}\boldsymbol{\Delta}_{\mathcal{G}}[y_i - \langle \mathcal{H}_i(t_i), \mathcal{G}^* \rangle - \mathbf{z}_i^{\top}\boldsymbol{\beta}^*]$$
$$+ n\sum_{r=1}^{R}\sum_{s=1}^{S} \{P_{\omega_{\gamma}^c}(\|\boldsymbol{\gamma}_{rs}\|_{\mathcal{B}_1}) - P_{\omega_{\gamma}^c}(\|\boldsymbol{\gamma}_{rs}^*\|_{\mathcal{B}_1})\} + n\sum_{r=1}^{R}\sum_{s=1}^{S} \{P_{\omega_{\gamma}^v}(\|\boldsymbol{\gamma}_{rs}\|_{\mathcal{B}_2}) - P_{\omega_{\gamma}^v}(\|\boldsymbol{\gamma}_{rs}^*\|_{\mathcal{B}_2})\}$$
$$=: I_4 + I_5 + I_6 + I_7.$$

For $I_1$, we have with probability tending to 1,

$$I_1 = \frac{1}{2}\delta_n^2\sum_{i=1}^n \mathbf{z}_i^\top \boldsymbol{\Delta}_\beta \boldsymbol{\Delta}_\beta^\top \mathbf{z}_i = \frac{1}{2}\delta_n^2\sum_{i=1}^n \boldsymbol{\Delta}_\beta^\top \mathbf{z}_i\mathbf{z}_i^\top \boldsymbol{\Delta}_\beta = \Omega(nk_n^{-1}\delta_n^2\|k_n^{1/2}\boldsymbol{\Delta}_\beta\|^2).$$

For $I_4$, note that similar to Lemma 3 in the supplementary material of Chen et al. (2023) we can show that with probability tending to 1, the eigenvalues of $n^{-1}\sum_{i=1}^n \tilde{\mathbf{u}}_i\tilde{\mathbf{u}}_i^\top$ are of order $\Omega(k_n^{-1})$. Therefore, we have there exists a constant $C_0 > 0$ such that with probability tending to 1,

$$I_4 = \frac{1}{2}\delta_n^2\sum_{i=1}^n \mathbf{u}_i^\top \boldsymbol{\Delta}_\mathcal{G}\boldsymbol{\Delta}_\mathcal{G}^\top \mathbf{u}_i = \frac{1}{2}\delta_n^2\boldsymbol{\Delta}_\mathcal{G}^\top \sum_{i=1}^n \mathbf{u}_i\mathbf{u}_i^\top \boldsymbol{\Delta}_\mathcal{G} \geq C_0 nk_n^{-1}\delta_n^2\|\boldsymbol{\Delta}_\mathcal{G}\|^2.$$

For $I_2$, using the facts that with probability tending to 1, uniformly for all $\|\tilde{\boldsymbol{\Delta}}\| = C$, we have

$$\sup_{\|\tilde{\boldsymbol{\Delta}}\|=C} n^{-1}\sum_{i=1}^n \mathbf{z}_i^\top \boldsymbol{\Delta}_\beta \epsilon_i = O_p\left(\sqrt{\frac{\log n}{n^{1-\tau}}}\|\boldsymbol{\Delta}_\beta\|\right),$$

and

$$\sup_{\|\tilde{\boldsymbol{\Delta}}\|=C} n^{-1}\sum_{i=1}^n \mathbf{z}_i^\top \boldsymbol{\Delta}_\beta \mathbf{u}_i^\top \boldsymbol{\Delta}_\mathcal{G} = O_p\left(k_n^{-1/2}\sqrt{\frac{\log n}{n^{1-\tau}}}\|\boldsymbol{\Delta}_\beta\|\|\boldsymbol{\Delta}_\mathcal{G}\|\right).$$

Thus we have

$$
\begin{aligned}
I_2 =& -\delta_n\sum_{i=1}^n \mathbf{z}_i^\top \boldsymbol{\Delta}_\beta\left[y_i - \langle\mathcal{H}_i(t_i),\mathcal{G}\rangle - \mathbf{z}_i^\top \boldsymbol{\beta}^*\right]\\
=& -\delta_n\sum_{i=1}^n \mathbf{z}_i^\top \boldsymbol{\Delta}_\beta\left[\langle\mathbf{X}_i^*,\mathbf{A}^*(t_i)\rangle + \mathbf{z}_i^\top \boldsymbol{\beta}^* + \epsilon_i - \langle\mathcal{H}_i(t_i),\mathcal{G}^*\rangle + \langle\mathcal{H}_i(t_i),\mathcal{G}^*-\mathcal{G}\rangle - \mathbf{z}_i^\top \boldsymbol{\beta}^*\right]\\
=& -\delta_n\sum_{i=1}^n \mathbf{z}_i^\top \boldsymbol{\Delta}_\beta\left\{\sum_{r=1}^R\sum_{s=1}^S x_{irs}^*\left[a_{rs}^*(t_i) - \mathbf{B}^\top(t_i)\boldsymbol{\gamma}_{rs}^*\right]\right\} - \delta_n\sum_{i=1}^n \mathbf{z}_i^\top \boldsymbol{\Delta}_\beta \epsilon_i + \delta_n^2\sum_{i=1}^n \mathbf{z}_i^\top \boldsymbol{\Delta}_\beta \mathbf{u}_i^\top \boldsymbol{\Delta}_\mathcal{G}\\
\geq& -\delta_n\sum_{i=1}^n \mathbf{z}_i^\top \boldsymbol{\Delta}_\beta\sum_{r=1}^R\sum_{s=1}^S\left|x_{irs}^* R_{rs}\right| - O_p\left(\delta_n\sqrt{n^{1+\tau}\log n}\|\boldsymbol{\Delta}_\beta\|\right)\\
=& -O_p\left(n\delta_n k_n^{-5/2}\|k_n^{1/2}\boldsymbol{\Delta}_\beta\| + \delta_n k_n^{-1/2}\sqrt{n^{1+\tau}\log n}\|k_n^{1/2}\boldsymbol{\Delta}_\beta\|\right).
\end{aligned}
$$

Similarly, using the fact that with probability tending to 1,

$$\sup_{\|\tilde{\boldsymbol{\Delta}}\|=C} n^{-1}\sum_{i=1}^n \mathbf{u}_i^\top \boldsymbol{\Delta}_\mathcal{G}\epsilon_i = O_p\left(\sqrt{\frac{\log n}{n}}\|\boldsymbol{\Delta}_\mathcal{G}\|\right),$$

we have, for $I_5$,

$$
\begin{aligned}
I_5 = & -\delta_n \sum_{i=1}^n \mathbf{u}_i^\top \boldsymbol{\Delta}_{\mathcal{G}} \left[ y_i - \langle \mathcal{H}_i(t_i), \mathcal{G}^* \rangle - \mathbf{z}_i^\top \boldsymbol{\beta}^* \right] \\
= & -\delta_n \sum_{i=1}^n \mathbf{u}_i^\top \boldsymbol{\Delta}_{\mathcal{G}} \left[ \langle \mathbf{X}_i^*, \mathbf{A}^*(t_i) \rangle + \mathbf{z}_i^\top \boldsymbol{\beta}^* + \epsilon_i - \langle \mathcal{H}_i(t_i), \mathcal{G}^* \rangle - \mathbf{z}_i^\top \boldsymbol{\beta}^* \right] \\
= & -\delta_n \sum_{i=1}^n \mathbf{u}_i^\top \boldsymbol{\Delta}_{\mathcal{G}} \left\{ \sum_{r=1}^R \sum_{s=1}^S x_{irs}^* \left[ a_{rs}^*(t_i) - \mathbf{B}^\top(t_i) \boldsymbol{\gamma}_{rs}^* \right] \right\} - 2\delta_n \sum_{i=1}^n \mathbf{u}_i^\top \boldsymbol{\Delta}_{\mathcal{G}} \epsilon_i \\
\geq & -\delta_n \sum_{i=1}^n \mathbf{u}_i^\top \boldsymbol{\Delta}_{\mathcal{G}} \sum_{r=1}^R \sum_{s=1}^S \left| x_{irs}^* R_{rs} \right| - O_p \left( n\delta_n \sqrt{\frac{\log n}{n}} \| \boldsymbol{\Delta}_{\mathcal{G}} \| \right) \\
= & -O_p \left( n\delta_n k_n^{-5/2} \| \boldsymbol{\Delta}_{\mathcal{G}} \| + \delta_n \sqrt{n \log n} \| \boldsymbol{\Delta}_{\mathcal{G}} \| \right)
\end{aligned}
$$

With $\| k_n^{1/2} \boldsymbol{\Delta}_\beta \| + \| \boldsymbol{\Delta}_{\mathcal{G}} \| \geq \| \tilde{\boldsymbol{\Delta}} \|$, we have, there exists a constant $C > 0$ such that with probability tending to 1,

$$
I_1 + I_4 + I_2 + I_5 \geq C_1 \left( n\delta_n^2 k_n^{-1} \| \tilde{\boldsymbol{\Delta}} \|^2 - n\delta_n k_n^{-5/2} \| \tilde{\boldsymbol{\Delta}} \| - \delta_n \max \left\{ n^{\tau/2} k_n^{-1/2}, 1 \right\} \sqrt{n \log n} \| \tilde{\boldsymbol{\Delta}} \| \right).
$$

When $\delta_n = k_n^{-3/2} + \max \left\{ n^{\tau/2}, k_n^{1/2} \right\} k_n^{1/2} \sqrt{\frac{\log n}{n}}$ and when the radius $C$ is large enough, we have $I_1 + I_4 + I_2 + I_5 > 0$ with probability tending to 1. In particular, $\delta_n$ is minimized when $k_n = \Omega \left( \min \left\{ \left( \frac{n}{\log n} \right)^{1/5}, \left( \frac{n^{1-\tau}}{\log n} \right)^{1/4} \right\} \right)$.

For $I_3$, notice that Condition (C8) implies $\max \{ P_{\omega_\beta}''(|\beta_k^*|), P_{\omega_\gamma^c}''(\| \boldsymbol{\gamma}^* \|_{\mathcal{B}_1}), P_{\omega_\gamma^v}''(\| \boldsymbol{\gamma}^* \|_{\mathcal{B}_2}), |\beta_k^*| \neq 0, \| \boldsymbol{\gamma}^* \|_{\mathcal{B}_1} \neq 0, \| \boldsymbol{\gamma}^* \|_{\mathcal{B}_2} \neq 0 \} \to 0$. By the fact that $\omega_\beta = O(k_n^{-1/2} \delta_n)$, we have

$$
\begin{aligned}
I_3 = & n \sum_{k=1}^{p_0} \left\{ P_{\omega_\beta}(|\beta_k|) - P_{\omega_\beta}(|\beta_k^*|) \right\} \\
= & n \sum_{k=1}^{p_0} \left\{ P_{\omega_\beta}(|\beta_k^* + \delta_n \boldsymbol{\Delta}_{\beta,k}|) - P_{\omega_\beta}(|\beta_k^*|) \right\} \\
\geq & - \left| n \sum_{k:\beta_k^* \neq 0} \left\{ \delta_n \boldsymbol{\Delta}_{\beta,k} P_{\omega_\beta}'(|\beta_k^*|) \operatorname{sign}(|\beta_k^*|) + \frac{1}{2} \delta_n^2 \boldsymbol{\Delta}_{\beta,k}^2 P_{\omega_\beta}''(|\beta_k^*|)[1 + o(1)] \right\} \right| \\
= & O \left( n\delta_n k_n^{-1/2} \omega_\beta \| k_n^{1/2} \boldsymbol{\Delta}_\beta \| \right) \\
\geq & - C_2 n\delta_n^2 k_n^{-1} \| k_n^{1/2} \boldsymbol{\Delta}_\beta \|,
\end{aligned}
$$

for some $C_2 > 0$. Similarly, for $I_6$ and $I_7$, by the fact that $\omega_\gamma^v = O(k_n^{-1} \delta_n), \omega_\gamma^c = O(k_n^{-1} \delta_n)$, we can show that there exists a constant $C_3 > 0$ such that

$$
I_6 + I_7 \geq -C_3 n\delta_n^2 k_n^{-1} \| \boldsymbol{\Delta}_{\mathcal{G}} \|.
$$

Therefore, when $C$ is large enough, we have $I_1 + I_4 > |I_2 + I_5| + |I_3 + I_6 + I_7|$, which implies that $\sum_{\ell=1}^7 I_\ell > 0$. Therefore, we conclude that $\| \hat{\mathcal{G}}^\ddagger - \mathcal{G}^* \| = O_p(\delta_n) = O \left( k_n^{-3/2} + \max \left\{ n^{\tau/2}, k_n^{1/2} \right\} k_n^{1/2} \sqrt{\frac{\log n}{n}} \right), \| \hat{\boldsymbol{\beta}}^\ddagger - \boldsymbol{\beta}^* \| = O_p(k_n^{-1/2} \delta_n) = O \left( k_n^{-2} + \max \left\{ n^{\tau/2}, k_n^{1/2} \right\} \sqrt{\frac{\log n}{n}} \right)$. The theorem is concluded by $\| \hat{a}_{rs}^\ddagger(t) - \hat{a}_{rs}^*(t) \| = O_p(k_n^{-1/2}) \| \hat{\mathcal{G}}^\ddagger - \mathcal{G}^* \|$ and plugging in $k_n =$

$$\Omega\Big(\min\Big\{\big(\tfrac{n}{\log n}\big)^{1/5}, \big(\tfrac{n^{1-\tau}}{\log n}\big)^{1/4}\Big\}\Big).$$

**Proof of theorem 4**: This can be proved using the results in the previous proof and the arguments in the proof of Lemma 1 of Fan and Li (2001).

## 8 Additional simulation results

To better illustrate our estimation procedure, we present Figure 7 to provide some further insight. Panel (a) is the ground truth of $\mathbf{A}(t)$ in case III with $\rho = 0.1$; Panel (b) is estimated from the un-penalized kernel smoothing method using (11); Panel (c) is the penalized estimator $\hat{\mathbf{A}}(t)$ using (18); Panel (d) is the a further refined smooth estimation $\hat{\mathbf{A}}(t)$ after identifying model structure. Clearly the initial kernel smoothing method cannot accurately estimate coefficients that vary too rapidly, particularly in the lower-right space of $\hat{\mathbf{A}}(t)$. After applying the penalized estimation, the fine details of $\mathbf{A}(t)$ are estimated more precisely but exhibit some local wrinkles. Finally implementing another kernel smoothing with the identified model structure, the estimated tensor becomes smoother and closer to the truth.

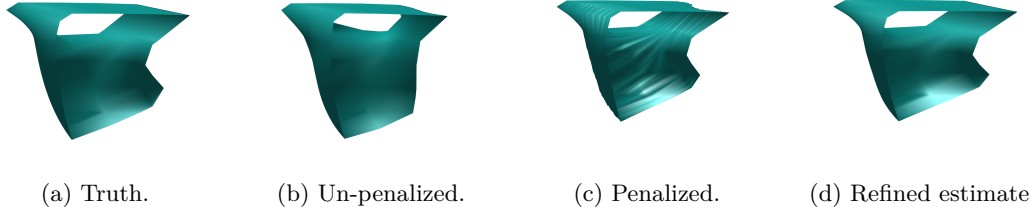

(a) Truth.  (b) Un-penalized.  (c) Penalized.  (d) Refined estimate.

Figure 7: Ground truth of $\mathbf{A}(t)$ and different estimates $\hat{\mathbf{A}}(t)$ in Case III, $\rho = 0.1$.

In the following, we provide additional simulation results to illustrate our model performance under simpler partition strategies and lower CP rank with less sample size. For Case I, given tensors $\mathcal{X}_i \in \mathbb{R}^{p_1 \times p_2}$ for $i = 1, 2, \cdots, n$, a partition strategy and a CP rank are chosen from $(S^{(1)}, S^{(2)}, R) \in \{(3,3,3), (3,3,6), (4,4,3)\}$. For Case II, given tensors $\mathcal{X}_i \in \mathbb{R}^{p_1 \times p_2 \times p_3}$ for $i = 1, 2, \cdots, n$, a partition strategy and a CP rank are chosen from $(S^{(1)}, S^{(2)}, S^{(3)}, R) \in \{(3,3,3,3), (3,3,3,6), (4,4,4,3)\}$. We set the sample size $n = 1200$, and 1600. Consider i.i.d. noise generated from two different distributions: $\Pi_1 = N(0,1)$, a standard Gaussian distribution; $\Pi_2 = t(5)$, a t-distribution with 5 degrees of freedom, representing a heavy-tailed noise. Other settings are consistent with those in section 5. Estimation accuracy of $\mathbf{A}(t)$ and $\boldsymbol{\beta}$ are summarized in Table 8 and 9.

From Table 8 and 9, we can easily see our model learns unknown function $\mathbf{A}(t)$ and coefficient $\boldsymbol{\beta}$ better with larger sample size. Compared to the white noise setting, the estimation accuracy of our model decreases under heavy-tailed noise. However, it still maintains a relatively low level of error. Due to the increased number of parameters to be estimated in Case II introduced by high dimension compared to Case I, the estimation accuracy shown in Table 9 is lower than that in Table 8. Furthermore, with increasing dimensionality, the influence of the partition strategy on parameter estimation accuracy becomes more significant than that of the CP rank.

Table 8: Estimation consistency of 200 simulations for Case I: $\hat{\mathbf{A}}^{\dagger}(t)$'s accuracy is measured by RMISE = $\mathrm{E}^{1/2}\big[\int_0^1(\hat{\mathbf{A}}^{\dagger}(t)-\mathbf{A}(t))^2dt\big]$, and $\hat{\boldsymbol{\beta}}^{\dagger}$'s accuracy is measured by RMSE = $\big[\frac{1}{p_0}\sum_{j=1}^{p_0}(\hat{\beta}_j^{\dagger}-\beta_j)^2\big]^{1/2}$.

| $n$ | 1200 | | | | 1600 | | | |
|---|---|---|---|---|---|---|---|---|
| Noise | $\Pi_1$ | | $\Pi_2$ | | $\Pi_1$ | | $\Pi_2$ | |
| Estimator | $\hat{\mathbf{A}}^{\dagger}(t)$ | $\hat{\boldsymbol{\beta}}^{\dagger}$ | $\hat{\mathbf{A}}^{\dagger}(t)$ | $\hat{\boldsymbol{\beta}}^{\dagger}$ | $\hat{\mathbf{A}}^{\dagger}(t)$ | $\hat{\boldsymbol{\beta}}^{\dagger}$ | $\hat{\mathbf{A}}^{\dagger}(t)$ | $\hat{\boldsymbol{\beta}}^{\dagger}$ |
| $(S^{(1)}, S^{(2)}, R) = (3, 3, 3)$ | | | | | | | | |
| $\mathbf{A}_1(t)$ | 0.0536 | 0.0528 | 0.0667 | 0.0725 | 0.0462 | 0.0502 | 0.0575 | 0.0647 |
| $\mathbf{A}_2(t)$ | 0.0537 | 0.0470 | 0.0704 | 0.0619 | 0.0477 | 0.0516 | 0.0603 | 0.0630 |
| $\mathbf{A}_3(t)$ | 0.0705 | 0.0540 | 0.0836 | 0.0865 | 0.0650 | 0.0516 | 0.0773 | 0.0634 |
| $\mathbf{A}_4(t)$ | 0.0873 | 0.0446 | 0.0967 | 0.0796 | 0.0844 | 0.0475 | 0.0923 | 0.0640 |
| $(S^{(1)}, S^{(2)}, R) = (3, 3, 6)$ | | | | | | | | |
| $\mathbf{A}_1(t)$ | 0.0579 | 0.0535 | 0.0731 | 0.0596 | 0.0462 | 0.0308 | 0.0614 | 0.0545 |
| $\mathbf{A}_2(t)$ | 0.0577 | 0.0592 | 0.0745 | 0.0477 | 0.0481 | 0.0506 | 0.0611 | 0.0798 |
| $\mathbf{A}_3(t)$ | 0.0735 | 0.0538 | 0.0855 | 0.0673 | 0.0646 | 0.0555 | 0.0732 | 0.0712 |
| $\mathbf{A}_4(t)$ | 0.0866 | 0.0587 | 0.1013 | 0.0536 | 0.0806 | 0.0517 | 0.0922 | 0.0625 |
| $(S^{(1)}, S^{(2)}, R) = (4, 4, 3)$ | | | | | | | | |
| $\mathbf{A}_1(t)$ | 0.0557 | 0.0667 | 0.0712 | 0.0751 | 0.0467 | 0.0483 | 0.0620 | 0.0681 |
| $\mathbf{A}_2(t)$ | 0.0543 | 0.0549 | 0.0725 | 0.0693 | 0.0491 | 0.0516 | 0.0636 | 0.0692 |
| $\mathbf{A}_3(t)$ | 0.0716 | 0.0750 | 0.0828 | 0.0898 | 0.0643 | 0.0635 | 0.0756 | 0.0618 |
| $\mathbf{A}_4(t)$ | 0.0908 | 0.0436 | 0.0994 | 0.0628 | 0.0841 | 0.0376 | 0.0919 | 0.0822 |

Table 9: Estimation consistency of 200 simulations for Case II: $\hat{\mathbf{A}}^{\dagger}(t)$'s accuracy is measured by RMISE = $\mathrm{E}^{1/2}\big[\int_0^1(\hat{\mathbf{A}}^{\dagger}(t)-\mathbf{A}(t))^2dt\big]$, and $\hat{\boldsymbol{\beta}}^{\dagger}$'s accuracy is measured by RMSE = $\big[\frac{1}{p_0}\sum_{j=1}^{p_0}(\hat{\beta}_j^{\dagger}-\beta_j)^2\big]^{1/2}$.

| $n$ | 1200 | | | | 1600 | | | |
|---|---|---|---|---|---|---|---|---|
| Noise | $\Pi_1$ | | $\Pi_2$ | | $\Pi_1$ | | $\Pi_2$ | |
| Estimator | $\hat{\mathbf{A}}^{\dagger}(t)$ | $\hat{\boldsymbol{\beta}}^{\dagger}$ | $\hat{\mathbf{A}}^{\dagger}(t)$ | $\hat{\boldsymbol{\beta}}^{\dagger}$ | $\hat{\mathbf{A}}^{\dagger}(t)$ | $\hat{\boldsymbol{\beta}}^{\dagger}$ | $\hat{\mathbf{A}}^{\dagger}(t)$ | $\hat{\boldsymbol{\beta}}^{\dagger}$ |
| $(S^{(1)}, S^{(2)}, S^{(3)}, R) = (3, 3, 3, 3)$ | | | | | | | | |
| $\mathbf{A}_1(t)$ | 0.0615 | 0.0526 | 0.0789 | 0.0672 | 0.0501 | 0.0491 | 0.0660 | 0.0558 |
| $\mathbf{A}_2(t)$ | 0.0613 | 0.0614 | 0.0790 | 0.0730 | 0.0502 | 0.0483 | 0.0641 | 0.0647 |
| $\mathbf{A}_3(t)$ | 0.0775 | 0.0593 | 0.0924 | 0.0721 | 0.0679 | 0.0528 | 0.0793 | 0.0610 |
| $\mathbf{A}_4(t)$ | 0.0948 | 0.0611 | 0.1077 | 0.0734 | 0.0860 | 0.0558 | 0.0953 | 0.0669 |
| $(S^{(1)}, S^{(2)}, S^{(3)}, R) = (3, 3, 3, 6)$ | | | | | | | | |
| $\mathbf{A}_1(t)$ | 0.0877 | 0.0554 | 0.1126 | 0.0664 | 0.0611 | 0.0464 | 0.0791 | 0.0535 |
| $\mathbf{A}_2(t)$ | 0.0872 | 0.0519 | 0.1134 | 0.0736 | 0.0613 | 0.0462 | 0.0793 | 0.0587 |
| $\mathbf{A}_3(t)$ | 0.1048 | 0.0552 | 0.1260 | 0.0776 | 0.0783 | 0.0516 | 0.0922 | 0.0564 |
| $\mathbf{A}_4(t)$ | 0.1252 | 0.0641 | 0.1438 | 0.0760 | 0.0960 | 0.0502 | 0.1093 | 0.0670 |
| $(S^{(1)}, S^{(2)}, S^{(3)}, R) = (4, 4, 4, 3)$ | | | | | | | | |
| $\mathbf{A}_1(t)$ | 0.1119 | 0.0598 | 0.1443 | 0.0794 | 0.0741 | 0.0406 | 0.0874 | 0.0622 |
| $\mathbf{A}_2(t)$ | 0.1135 | 0.0574 | 0.1456 | 0.0796 | 0.0744 | 0.0426 | 0.0877 | 0.0616 |
| $\mathbf{A}_3(t)$ | 0.1375 | 0.0590 | 0.1627 | 0.0760 | 0.0943 | 0.0424 | 0.1043 | 0.0643 |
| $\mathbf{A}_4(t)$ | 0.1642 | 0.0670 | 0.1874 | 0.0923 | 0.1165 | 0.0490 | 0.1227 | 0.0615 |

Next, we conduct additional experiments on our penalized model. As the Lasso and MCP penalties yield similar results to SCAD, only the SCAD penalty is presented.

In Case III, to explore selection performance under various spatial correlation among tensor covariates, we generate different factor matrix $\mathbf{X}^{*(s)}$'s with a same auto-regressive covariance matrix in the main manuscript. For tensor function $\mathcal{A}(t)$ with $\mathcal{A}^{(s)}(t) = \sum_r^R a_{rs}(t)\mathbf{x}_r^{(s1)} \circ \mathbf{x}_r^{(s2)}$. If $1 \leq s^{(1)} < s^{(2)} \leq 3$, $a_{rs}(t) = \sqrt{rs^{(1)}s^{(2)}/RS^{(1)}S^{(2)}}\sin(2\pi(t-0.5))$, and if $1 \leq s^{(2)} \leq s^{(1)} \leq 3$, $a_{rs}(t) = \sqrt{rs^{(1)}s^{(2)}/RS^{(1)}S^{(2)}}$, for $1 \leq r \leq 3$. And the rest of tensor functions are all 0's. Corresponding partition strategy and CP rank are chosen from $(S^{(1)}, S^{(2)}, R) \in \{(3, 3, 6), (4, 4, 3)\}$. All other settings are the same with those in section 5.2. We set the sample size $n = 800, 1200$ and 1600, and correlation parameter $\rho = 0.1, 0.5$ and 0.9. Only SCAD penalty is considered, and all experiment results are summarized in Table 10 and 11.

Table 10: Selection accuracy of 200 simulations for Case III. Partition strategy and CP rank: $(S^{(1)}, S^{(2)}, R) = (3, 3, 6)$. Sensitivity (se), specificity (sp), positive predictive value (ppv), and negative predictive value (npv) are used to evaluate performance, where se $= \mathrm{P}(\mathcal{F}_i \cap \hat{\mathcal{F}}_i | \mathcal{F}_i)$, ppv $= \mathrm{P}(\mathcal{F}_i \cap \hat{\mathcal{F}}_i | \hat{\mathcal{F}}_i)$, sp $= \mathrm{P}(\mathcal{F}_i^c \cap \hat{\mathcal{F}}_i^c | \mathcal{F}_i^c)$ and npv $= \mathrm{P}(\mathcal{F}_i^c \cap \hat{\mathcal{F}}_i^c | \mathcal{F}_i)$. SCAD penalty parameter is selected by BIC.

| $n$ | 800 | | | 1200 | | | 1600 | | |
|---|---|---|---|---|---|---|---|---|---|
| $\rho$ | 0.1 | 0.5 | 0.9 | 0.1 | 0.5 | 0.9 | 0.1 | 0.5 | 0.9 |
| | Zero coefficient in $\mathbf{A}(t)$ | | | | | | | | |
| se | 0.9852 | 0.9733 | 0.8126 | 0.9978 | 0.9978 | 0.9163 | 0.9993 | 0.9993 | 0.9659 |
| ppv | 0.9921 | 0.9934 | 0.9669 | 0.9958 | 0.9914 | 0.9803 | 0.9943 | 0.9950 | 0.9883 |
| sp | 0.9919 | 0.9933 | 0.9719 | 0.9956 | 0.9911 | 0.9807 | 0.9941 | 0.9948 | 0.9881 |
| npv | 0.9858 | 0.9750 | 0.8433 | 0.9979 | 0.9979 | 0.9254 | 0.9993 | 0.9993 | 0.9690 |
| | Constant non-zero coefficient in $\mathbf{A}(t)$ | | | | | | | | |
| se | 0.8344 | 0.8700 | 0.6167 | 0.9600 | 0.9522 | 0.8233 | 0.9900 | 0.9844 | 0.8822 |
| ppv | 0.9852 | 0.9607 | 0.6999 | 0.9978 | 0.9967 | 0.8776 | 0.9989 | 0.9989 | 0.9513 |
| sp | 0.9939 | 0.9822 | 0.8689 | 0.9989 | 0.9983 | 0.9389 | 0.9994 | 0.9994 | 0.9744 |
| npv | 0.9288 | 0.9413 | 0.8265 | 0.9809 | 0.9776 | 0.9174 | 0.9951 | 0.9927 | 0.9462 |
| | Varying coefficient in $\mathbf{A}(t)$ | | | | | | | | |
| se | 0.9911 | 1.0000 | 0.9956 | 0.9956 | 0.9956 | 0.9867 | 0.9978 | 0.9978 | 0.9956 |
| ppv | 0.7924 | 0.8348 | 0.6202 | 0.9382 | 0.9400 | 0.7860 | 0.9960 | 0.9862 | 0.8558 |
| sp | 0.9329 | 0.9502 | 0.8551 | 0.9853 | 0.9853 | 0.9369 | 0.9991 | 0.9964 | 0.9591 |
| npv | 0.9982 | 1.0000 | 0.9989 | 0.9991 | 0.9991 | 0.9972 | 0.9996 | 0.9996 | 0.9991 |
| | Prediction error of $\hat{\mathbf{A}}^{\ddagger}(t)$ | | | | | | | | |
| MAE | 0.0413 | 0.0433 | 0.0862 | 0.0288 | 0.0312 | 0.0548 | 0.0233 | 0.0251 | 0.0425 |
| RMISE | 0.0656 | 0.0694 | 0.1440 | 0.0467 | 0.0504 | 0.0899 | 0.0369 | 0.0404 | 0.0727 |
| | Prediction error of $\hat{\boldsymbol{\beta}}^{\ddagger}$ | | | | | | | | |
| MAE | 0.0181 | 0.0216 | 0.0218 | 0.0140 | 0.0148 | 0.0177 | 0.0110 | 0.0129 | 0.0115 |
| RMSE | 0.0258 | 0.0303 | 0.0294 | 0.0205 | 0.0209 | 0.0236 | 0.0162 | 0.0188 | 0.0168 |

In Case IV, we focus on tensor covariates with unknown parameters greater than the sample size. The partition strategy and CP rank is designed as $(S^{(1)}, S^{(2)}, S^{(3)}, R) = (4, 4, 4, 6)$. Tensor function $\mathcal{A}(t)$ is generated with $\mathcal{A}^{(s)}(t) = \sum_r^R a_{rs}(t) \mathbf{x}_r^{(s1)} \circ \mathbf{x}_r^{(s2)} \circ \mathbf{x}_r^{(s3)}$. If $s^{(1)} + s^{(2)} + s^{(3)} < 6$, $a_{rs}(t) = \sqrt{rs^{(1)}s^{(2)}s^{(3)}/RS^{(1)}S^{(2)}S^{(3)}} \sin(2\pi(t - 0.5))$, and if $s^{(1)} + s^{(2)} + s^{(3)} \geq 6$ and $1 \leq s^{(1)}, s^{(2)}, s^{(3)} \leq 3$, $a_{rs}(t) = \sqrt{rs^{(1)}s^{(2)}s^{(3)}/RS^{(1)}S^{(2)}S^{(3)}}$, for $4 \leq r \leq 6$. And the rest of the tensor functions are all 0's. All other settings are the same with those in section 5.2. We set the sample size $n = 400, 800, 1200$ and $1600$, and consider noise with two different distributions, $\Pi_1$ and $\Pi_2$. All experiment results are summarized in Table 12.

From Table 10 and 11, we can find our penalized model can identify model structure and detect sparsity with high probability in both proposals of partition strategy and CP rank. And selection accuracy improves with larger sample sizes, while it deteriorates with increasing covariate correlation. A same trend is also observed in the prediction errors of the estimated $\mathbf{A}(t)$ and $\boldsymbol{\beta}$.

Table 12 illustrates our penalized model in a more complicated case, where $RS = 6 \times 64$. In this case, un-penalized estimation approach is invalid for sample size $n = 400$ and $800$. As the sample size increases, selection accuracy improves significantly. When sample size $n = 1600$, the penalized model can nearly perfectly distinguish varying, constant non-zero, and zero coefficients.

Table 11: Selection accuracy of 200 simulations for Case III. Partition strategy and CP rank: $(S^{(1)}, S^{(2)}, R) = (4, 4, 3)$. Partition strategy and CP rank: $(S^{(1)}, S^{(2)}, R) = (3, 3, 6)$. Sensitivity (se), specificity (sp), positive predictive value (ppv), and negative predictive value (npv) are used to evaluate performance, where se $= \mathrm{P}(\mathcal{F}_i \cap \hat{\mathcal{F}}_i | \mathcal{F}_i)$, ppv $= \mathrm{P}(\mathcal{F}_i \cap \hat{\mathcal{F}}_i | \hat{\mathcal{F}}_i)$, sp $= \mathrm{P}(\mathcal{F}_i^c \cap \hat{\mathcal{F}}_i^c | \mathcal{F}_i^c)$ and npv $= \mathrm{P}(\mathcal{F}_i^c \cap \hat{\mathcal{F}}_i^c | \mathcal{F}_i^c)$. SCAD penalty parameter is selected by BIC.

| $n$ | 800 | | | 1200 | | | 1600 | | |
|---|---|---|---|---|---|---|---|---|---|
| $\rho$ | 0.1 | 0.5 | 0.9 | 0.1 | 0.5 | 0.9 | 0.1 | 0.5 | 0.9 |
| | Zero coefficient in $\mathbf{A}(t)$ | | | | | | | | |
| se | 0.9752 | 0.9552 | 0.6429 | 0.9962 | 0.9933 | 0.8286 | 1.0000 | 1.0000 | 0.9114 |
| ppv | 0.9962 | 0.9894 | 0.9567 | 0.9964 | 0.9955 | 0.9773 | 0.9982 | 0.9982 | 0.9837 |
| sp | 0.9970 | 0.9919 | 0.9770 | 0.9970 | 0.9963 | 0.9859 | 0.9985 | 0.9985 | 0.9881 |
| npv | 0.9817 | 0.9674 | 0.7850 | 0.9972 | 0.9950 | 0.8863 | 1.0000 | 1.0000 | 0.9372 |
| | Constant non-zero coefficient in $\mathbf{A}(t)$ | | | | | | | | |
| se | 0.8467 | 0.8533 | 0.5789 | 0.9756 | 0.9611 | 0.7833 | 0.9967 | 0.9922 | 0.8833 |
| ppv | 0.9833 | 0.9722 | 0.7187 | 0.9979 | 0.9933 | 0.8671 | 1.0000 | 1.0000 | 0.9269 |
| sp | 0.9913 | 0.9853 | 0.8720 | 0.9987 | 0.9960 | 0.9247 | 1.0000 | 1.0000 | 0.9560 |
| npv | 0.9218 | 0.9244 | 0.7842 | 0.9860 | 0.9783 | 0.8829 | 0.9981 | 0.9955 | 0.9358 |
| | Varying coefficient in $\mathbf{A}(t)$ | | | | | | | | |
| se | 0.9978 | 0.9933 | 0.9889 | 0.9978 | 0.9978 | 0.9978 | 1.0000 | 0.9978 | 0.9978 |
| ppv | 0.8004 | 0.7924 | 0.5910 | 0.9587 | 0.9433 | 0.6831 | 0.9960 | 0.9904 | 0.8198 |
| sp | 0.9328 | 0.9251 | 0.7251 | 0.9892 | 0.9836 | 0.8749 | 0.9990 | 0.9974 | 0.9400 |
| npv | 0.9995 | 0.9984 | 0.9960 | 0.9995 | 0.9995 | 0.9994 | 1.0000 | 0.9995 | 0.9995 |
| | Prediction error of $\hat{\mathbf{A}}^{\ddagger}(t)$ | | | | | | | | |
| MAE | 0.0434 | 0.0486 | 0.1143 | 0.0298 | 0.0329 | 0.0697 | 0.0239 | 0.0269 | 0.0505 |
| RMISE | 0.0686 | 0.0768 | 0.1775 | 0.0474 | 0.0522 | 0.1108 | 0.0378 | 0.0429 | 0.0824 |
| | Prediction error of $\hat{\beta}^{\ddagger}$ | | | | | | | | |
| MAE | 0.0206 | 0.0209 | 0.0220 | 0.0157 | 0.0137 | 0.0156 | 0.0127 | 0.0122 | 0.0123 |
| RMSE | 0.0294 | 0.0294 | 0.0307 | 0.0220 | 0.0201 | 0.0221 | 0.0188 | 0.0177 | 0.0172 |

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

Table 12: Selection accuracy of 200 simulations for Case IV. Sensitivity (se), specificity (sp), positive predictive value (ppv), and negative predictive value (npv) are used to evaluate performance, where se $= P(\mathcal{F}_i \cap \hat{\mathcal{F}}_i | \mathcal{F}_i)$, ppv $= P(\mathcal{F}_i \cap \hat{\mathcal{F}}_i | \hat{\mathcal{F}}_i)$, sp $= P(\mathcal{F}_i^c \cap \hat{\mathcal{F}}_i^c | \mathcal{F}_i^c)$ and npv $= P(\mathcal{F}_i^c \cap \hat{\mathcal{F}}_i^c | \hat{\mathcal{F}}_i^c)$. SCAD penalty parameter is selected by BIC.

| $n$ | 400 | | 800 | | 1200 | | 1600 | |
|---|---|---|---|---|---|---|---|---|
| Noise | $\Pi_1$ | $\Pi_2$ | $\Pi_1$ | $\Pi_2$ | $\Pi_1$ | $\Pi_2$ | $\Pi_1$ | $\Pi_2$ |
| Zero coefficient in $\mathbf{A}(t)$ | | | | | | | | |
| se | 0.9766 | 0.8518 | 0.9884 | 0.9673 | 0.9987 | 0.9947 | 1.0000 | 0.9983 |
| ppv | 0.9966 | 0.9700 | 0.9997 | 0.9970 | 0.9997 | 0.9987 | 0.9997 | 0.9970 |
| sp | 0.9877 | 0.9012 | 0.9988 | 0.9889 | 0.9988 | 0.9951 | 0.9988 | 0.9889 |
| npv | 0.9204 | 0.6200 | 0.9590 | 0.8916 | 0.9879 | 0.9808 | 1.0000 | 0.9938 |
| Constant non-zero coefficient in $\mathbf{A}(t)$ | | | | | | | | |
| se | 0.9333 | 0.9608 | 0.9863 | 0.9745 | 0.9961 | 0.9826 | 1.0000 | 1.0000 |
| ppv | 0.9088 | 0.6147 | 0.9456 | 0.8425 | 0.9828 | 0.9702 | 1.0000 | 0.9904 |
| sp | 0.9847 | 0.8604 | 0.9913 | 0.9712 | 0.9973 | 0.9952 | 1.0000 | 0.9985 |
| npv | 0.9900 | 0.9932 | 0.9978 | 0.9949 | 0.9994 | 0.9961 | 1.0000 | 1.0000 |
| Varying coefficient in $\mathbf{A}(t)$ | | | | | | | | |
| se | 0.9700 | 0.9167 | 0.9967 | 0.9700 | 0.9967 | 0.9786 | 0.9967 | 0.9867 |
| ppv | 0.8631 | 0.6800 | 0.9686 | 0.9448 | 0.9968 | 0.9905 | 1.0000 | 1.0000 |
| sp | 0.9850 | 0.9944 | 0.9972 | 0.9946 | 0.9992 | 0.9997 | 1.0000 | 1.0000 |
| npv | 0.9975 | 0.9735 | 0.9997 | 0.9975 | 0.9997 | 0.9975 | 0.9997 | 0.9989 |
| Prediction error of $\hat{\mathbf{A}}^{\ddagger}(t)$ | | | | | | | | |
| MAE | 0.0378 | 0.0651 | 0.0224 | 0.0297 | 0.0188 | 0.0197 | 0.0115 | 0.0159 |
| RMISE | 0.1107 | 0.1431 | 0.0504 | 0.0646 | 0.0373 | 0.0418 | 0.0255 | 0.0341 |
| Prediction error of $\hat{\beta}^{\ddagger}$ | | | | | | | | |
| MAE | 0.0383 | 0.0657 | 0.0190 | 0.0371 | 0.0175 | 0.0253 | 0.0140 | 0.0213 |
| RMSE | 0.0680 | 0.1102 | 0.0279 | 0.0545 | 0.0267 | 0.0359 | 0.0226 | 0.0282 |

Jiahua Chen and Zehua Chen. Extended bayesian information criteria for model selection with large model spaces. *Biometrika*, 95(3):759–771, 2008.

Jianan Chen, Binyan Jiang, and Jialiang Li. Nonparametric instrument model averaging. *Journal of Nonparametric Statistics*, 35(4):905–926, 2023.

Ming-Yen Cheng, Toshio Honda, Jialiang Li, and Heng Peng. Nonparametric independence screening and structure identification for ultra-high dimensional longitudinal data. *The Annals of Statistics*, 42(5):1819–1849, 2014.

Ming-Yen Cheng, Toshio Honda, and Jialiang Li. Efficient estimation in semivarying coefficient models for longitudinal/clustered data. *The Annals of Statistics*, 44(5):1988–2017, 2016.

Carl De Boor. *A Practical Guide to Splines*. New York: Springer., 2001.

Daniel M Dunlavy, Tamara G Kolda, and Evrim Acar. Temporal link prediction using matrix and tensor factorizations. *ACM Transactions on Knowledge Discovery from Data*, 5(2):1–27, 2011.

Jianqing Fan. *Local Polynomial Modelling and Its Applications: Monographs on Statistics and Applied Probability 66*. Routledge, 2018.

Jianqing Fan and Tao Huang. Profile likelihood inferences on semiparametric varying-coefficient partially linear models. *Bernoulli*, 11(6):1031–1057, 2005.

Jianqing Fan and Runze Li. Variable selection via nonconcave penalized likelihood and its oracle properties. *Journal of the American Statistical Association*, 96(456):1348–1360, 2001.

Jianqing Fan and Wenyang Zhang. Statistical estimation in varying coefficient models. *The Annals of Statistics*, 27(5):1491–1518, 1999.

Yingying Fan and Cheng Yong Tang. Tuning parameter selection in high dimensional penalized likelihood. *Journal of the Royal Statistical Society Series B: Statistical Methodology*, 75(3): 531–552, 2013.

Silvia Gandy, Benjamin Recht, and Isao Yamada. Tensor completion and low-n-rank tensor recovery via convex optimization. *Inverse Problems*, 27(2):025010, 2011.

Rajarshi Guhaniyogi, Shaan Qamar, and David B Dunson. Bayesian tensor regression. *Journal of Machine Learning Research*, 18(79):1–31, 2017.

Chaohui Guo and Jialiang Li. Homogeneity and structure identification in semiparametric factor models. *Journal of Business & Economic Statistics*, 40(1):408–422, 2022.

Rungang Han, Yuetian Luo, Miaoyan Wang, and Anru R Zhang. Exact clustering in tensor block model: Statistical optimality and computational limit. *Journal of the Royal Statistical Society Series B: Statistical Methodology*, 84(5):1666–1698, 2022a.

Rungang Han, Rebecca Willett, and Anru R Zhang. An optimal statistical and computational framework for generalized tensor estimation. *The Annals of Statistics*, 50(1):1–29, 2022b.

Rungang Han, Pixu Shi, and Anru R Zhang. Guaranteed functional tensor singular value decomposition. *Journal of the American Statistical Association*, 119(546):995–1007, 2023.

Botao Hao, Anru Zhang, and Guang Cheng. Sparse and low-rank tensor estimation via cubic sketchings. *IEEE Transactions on Information Theory*, 66(9):5927–5964, 2020.

Trevor Hastie and Robert Tibshirani. Varying-coefficient models. *Journal of the Royal Statistical Society Series B: Statistical Methodology*, 55(4):757–779, 1993.

Christopher J Hillar and Lek-Heng Lim. Most tensor problems are np-hard. *Journal of the Association for Computing Machinery*, 60(6):1–39, 2013.

Victoria Hore, Ana Vinuela, Alfonso Buil, Julian Knight, Mark I McCarthy, Kerrin Small, and Jonathan Marchini. Tensor decomposition for multiple-tissue gene expression experiments. *Nature Genetics*, 48(9):1094–1100, 2016.

Xiaoling Huang, Xiangyin Kong, Ziyan Shen, Jing Ouyang, Yunxiang Li, Kai Jin, and Juan Ye. Grape: A multi-modal dataset of longitudinal follow-up visual field and fundus images for glaucoma management. *Scientific Data*, 10(1):520–520, 2023.

Tamara G Kolda and Brett W Bader. Tensor decompositions and applications. *SIAM Review*, 51(3):455–500, 2009.

Lexin Li and Xin Zhang. Parsimonious tensor response regression. *Journal of the American Statistical Association*, 112(519):1131–1146, 2017.

Xiaoshan Li, Da Xu, Hua Zhou, Lexin Li, et al. Tucker tensor regression and neuroimaging analysis. *Statistics in Biosciences*, 10(3):520–545, 2018.

Jianjun Liu, Zebin Wu, Liang Xiao, Jun Sun, and Hong Yan. Generalized tensor regression for hyperspectral image classification. *IEEE Transactions on Geoscience and Remote Sensing*, 58(2):1244–1258, 2019.

Xuan Liu, Xue Pan, Yuan Ma, Cheng Jin, Bo Wang, and Yi Ning. Variation in intraocular pressure by sex, age, and geographic location in china: a nationwide study of 284,937 adults. *Frontiers in Endocrinology*, 13:949827–949827, 2022.

Yipeng Liu, Jiani Liu, and Ce Zhu. Low-rank tensor train coefficient array estimation for tensor-on-tensor regression. *IEEE Transactions on Neural Networks and Learning Systems*, 31(12): 5402–5411, 2020.

Zihuan Liu, Cheuk Yin Lee, and Heping Zhang. Tensor quantile regression with low-rank tensor train estimation. *The Annals of Applied Statistics*, 18(2):1294–1318, 2024.

Eric F Lock. Tensor-on-tensor regression. *Journal of Computational and Graphical Statistics*, 27(3):638–647, 2018.

Yiqiang Lu. Generalized partially linear varying-coefficient models. *Journal of Statistical Planning and Inference*, 138(4):901–914, 2008.

Yuetian Luo and Anru R Zhang. Tensor clustering with planted structures: Statistical optimality and computational limits. *The Annals of Statistics*, 50(1):584–613, 2022.

Yuetian Luo and Anru R Zhang. Low-rank tensor estimation via riemannian gauss-newton: Statistical optimality and second-order convergence. *The Journal of Machine Learning Research*, 24(1):18274–18321, 2023.

YP Mack and BW Silverman. Weak and strong uniform consistency of kernel regression estimates. *Probability Theory and Related Fields*, 61(3):405–415, 1982.

Michelle F Miranda, Hongtu Zhu, Joseph G Ibrahim, and Alzheimer's Disease Neuroimaging Initiative. Tprm: Tensor partition regression models with applications in imaging biomarker detection. *The Annals of Applied Statistics*, 12(3):1422–1450, 2018.

Jingru Mu, Guannan Wang, and Li Wang. Estimation and inference in spatially varying coefficient models. *Environmetrics*, 29(1):e2485, 2018.

Whitney K Newey and Thomas M Stoker. Efficiency of weighted average derivative estimators and index models. *Econometrica*, 61(5):1199–1223, 1993.

David Pollard. Asymptotics for least absolute deviation regression estimators. *Econometric Theory*, 7(2):186–199, 1991.

Bruce E Prum, Lisa F Rosenberg, Steven J Gedde, Steven L Mansberger, Joshua D Stein, Sayoko E Moroi, Leon W Herndon, Michele C Lim, and Ruth D Williams. Primary open-angle glaucoma preferred practice pattern® guidelines. *Ophthalmology*, 123(1):41–111, 2016.

Charles J Stone. Optimal global rates of convergence for nonparametric regression. *The Annals of Statistics*, 10(4):1040–1053, 1982.

Will Wei Sun and Lexin Li. Store: sparse tensor response regression and neuroimaging analysis. *Journal of Machine Learning Research*, 18(135):1–37, 2017.

Xiwei Tang, Xuan Bi, and Annie Qu. Individualized multilayer tensor learning with an application in imaging analysis. *Journal of the American Statistical Association*, 115(530):836–851, 2020.

Robert Tibshirani. Regression shrinkage and selection via the lasso. *Journal of the Royal Statistical Society Series B: Statistical Methodology*, 58(1):267–288, 1996.

Tian Tong, Cong Ma, Ashley Prater-Bennette, Erin Tripp, and Yuejie Chi. Scaling and scalability: Provable nonconvex low-rank tensor estimation from incomplete measurements. *The Journal of Machine Learning Research*, 23(1):7312–7388, 2022.

Yingcun Xia, Wenyang Zhang, and Howell Tong. Efficient estimation for semivarying-coefficient models. *Biometrika*, 91(3):661–681, 2004.

Shan Yu, Yueying Wang, Li Wang, and Lei Gao. Spatiotemporal autoregressive partially linear varying coefficient models. *Statistica Sinica*, 32(4):2119–2146, 2022.

Ming Yuan and Cun-Hui Zhang. On tensor completion via nuclear norm minimization. *Foundations of Computational Mathematics*, 16(4):1031–1068, 2016.

Anru Zhang. Cross: Efficient low-rank tensor completion. *The Annals of Statistics*, 47(2): 936–964, 2019.

Anru Zhang and Rungang Han. Optimal sparse singular value decomposition for high-dimensional high-order data. *Journal of the American Statistical Association*, 114(528):1708–1725, 2019.

Anru Zhang and Dong Xia. Tensor svd: Statistical and computational limits. *IEEE Transactions on Information Theory*, 64(11):7311–7338, 2018.

Anru R Zhang, Yuetian Luo, Garvesh Raskutti, and Ming Yuan. Islet: Fast and optimal low-rank tensor regression via importance sketching. *SIAM Journal on Mathematics of Data Science*, 2(2):444–479, 2020.

Cun Hui Zhang. Nearly unbiased variable selection under minimax concave penalty. *The Annals of Statistics*, 38(2):894–942, 2010.

Wenyang Zhang, Sik-Yum Lee, and Xinyuan Song. Local polynomial fitting in semivarying coefficient model. *Journal of Multivariate Analysis*, 82(1):166–188, 2002.

Xin Zhang and Lexin Li. Tensor envelope partial least-squares regression. *Technometrics*, 59 (4):426–436, 2017.

Yanqing Zhang, Xuan Bi, Niansheng Tang, and Annie Qu. Dynamic tensor recommender systems. *Journal of Machine Learning Research*, 22(65):1–35, 2021.

Hua Zhou, Lexin Li, and Hongtu Zhu. Tensor regression with applications in neuroimaging data analysis. *Journal of the American Statistical Association*, 108(502):540–552, 2013.

Ya Zhou, Raymond KW Wong, and Kejun He. Broadcasted nonparametric tensor regression. *Journal of the Royal Statistical Society Series B: Statistical Methodology*, 86(5):1197–1220, 2024.

Xuening Zhu, Zhanrui Cai, and Yanyuan Ma. Network functional varying coefficient model. *Journal of the American Statistical Association*, 117(540):2074–2085, 2022.

