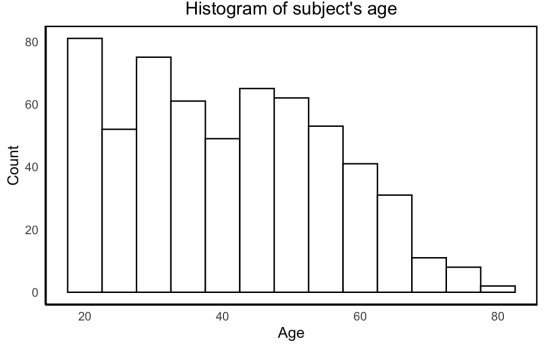

(a) Histogram of patient visits

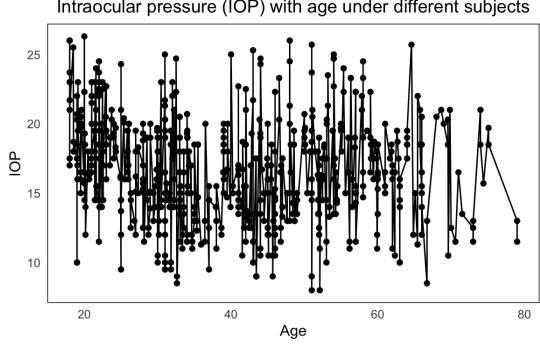

(b) Longitudinal IOP elevation

Figure 2: Fundus image analysis: statistical summary for original GRAPE dataset. (a) shows the histogram of subject's ages in the whole dataset. (b) shows the intraocular pressure (IOP) value with different ages.

Table 6: Fundus image analysis: MAE (left) and RMSE (right) for tensor partition on colored fundus photograph (CFP) and region of interest (ROI) with different $S$ and $R$. All results are generated from 10-fold cross validation.

| $R$ | | | | $S_1 \times S_2 \times S_3$ | $R$ | | | |
|---|---|---|---|---|---|---|---|---|
| 1 | 2 | 3 | 4 | | 1 | 2 | 3 | 4 |
| Colored fundus photograph (CFP) | | | | | | | | |
| 0.8799 | 0.7933 | 0.8389 | 0.9050 | $2 \times 2 \times 1$ | 1.1457 | 0.9793 | 1.0929 | 1.2842 |
| 0.7956 | **0.7728** | 0.8075 | 0.8355 | $3 \times 3 \times 1$ | 0.994 | **0.9581** | 1.0305 | 1.0818 |
| 0.8392 | 0.8886 | 0.9666 | 1.0915 | $4 \times 4 \times 1$ | 1.079 | 1.2014 | 1.3201 | 1.4951 |
| 0.8935 | 1.1866 | / | / | $6 \times 6 \times 1$ | 1.148 | 1.6424 | / | / |
| Region of interest (ROI) | | | | | | | | |
| 0.7658 | 0.7602 | 0.7517 | 0.7594 | $2 \times 2 \times 1$ | 0.9619 | 0.9519 | 0.9521 | 0.9528 |
| 0.7640 | **0.7449** | 0.7726 | 0.7838 | $3 \times 3 \times 1$ | 0.9552 | **0.9447** | 0.9912 | 1.0233 |
| 0.7478 | 0.8023 | 0.8470 | 0.9094 | $4 \times 4 \times 1$ | 0.9503 | 1.0330 | 1.1179 | 1.1985 |
| 0.7958 | 1.0077 | / | / | $6 \times 6 \times 1$ | 1.0168 | 1.3167 | / | / |

Figure 3 shows the estimated penalized varying-coefficient functions $a_{rs}(t)$ for CFP and ROI. The corresponding mean and CI curves are generated from bootstrap method.

Table 7 shows the estimated constant coefficient $\boldsymbol{\beta}$ for gender and 59 different VF locations with tensor covariate CFP and ROI. Gender and significant VFs are selected to present in the main text.

At last, we compared our VCTR model with other two constant coefficient tensor models consistent with those in section **??**, and the experiment results are shown in Table 8. We can find model $M_3$ proposed in Zhou et al. (2013) is overfitting. While this model has the best performance in in-sample errors, its prediction ability is the worst in out-of-sample error. That is, even if we set CP-rank $R = 1$ for the kruskal regression in model $M_3$, it still has $R(p_1 + p_2 + p_3) + p_0 = 392$ unknown parameters. Our