# OpenReview forum: "Varying Coefficient Tensor Regression"
_SLADS/Section_B — Accepted by SLADS_Section_B_

### Review · Reviewer_AAim · 2026-03-23

**Summary Of Contributions:**

This paper proposes a varying coefficient tensor regression framework for scalar-on-tensor regression with an additional scalar index variable. The main idea is to first partition a high-dimensional tensor covariate into sub-tensors, then apply a CP decomposition to each partition, and finally use the resulting low-dimensional subject-specific factors as predictors in a varying coefficient model. The paper further develops a penalized estimation procedure, based on spline approximation and group SCAD-type penalties, to perform simultaneous variable selection and identification of whether a coefficient is zero, constant, or genuinely varying. A final refinement step applies additional smoothing after model structure identification. The manuscript also provides asymptotic results for the local-linear estimator and for the penalized estimator under a set of regularity conditions, and illustrates the method through simulations and a glaucoma imaging application based on the GRAPE dataset.

In my reading, the paper's main methodological contribution is the integration of three ingredients into one framework:

*tensor partition + low-rank tensor feature extraction + varying-coefficient regression*,

followed by a structured penalization scheme to distinguish among zero, constant, and varying effects. This is a meaningful extension of existing tensor regression methods, which are mostly parametric and typically do not allow the tensor effect to vary with an index variable.

**Audience:**

Yes

**Broader Impact Concerns:**

I do not see major ethical concerns specific to the methodology itself.

**Claims And Evidence:**

No

**Requested Changes:**

**Critical**:
- The manuscript should explicitly discuss the fact that the GRAPE records are longitudinal/repeated observations and that treating each record as independent may affect both estimation uncertainty and interpretation. At a minimum, this limitation should be clearly acknowledged. Preferably, the authors should either:
  - reanalyze the data with subject- or eye-level resampling / data splitting,
  - use a clustered or repeated-measures validation strategy,
  - or explain why the dependence is weak enough not to materially affect conclusions.
- The authors should clearly distinguish between the setting covered by the asymptotic theory, and the practical settings used in the application. In particular, they should discuss the independence assumption $t_i\perp (\mathbb{X}_i^*, z_i)$, explain its role in the theoretical development, and comment on whether this assumption is plausible in the fundus-image application. As written, the theory appears to rely on conditions that may not hold in the applied setting, and this issue should be addressed more directly.

**Recommended**:
- It would strengthen the paper to explain what aspects of the tensor effect remain identifiable after partition-wise CP decomposition and how the varying coefficients relate back to image regions.

**Strengths And Weaknesses:**

**Strengths**:
- Extending tensor regression beyond constant linear effects is worthwhile, especially for biomedical imaging applications where interactions with age or time are scientifically plausible. The proposed model is well motivated from this perspective.
- The manuscript includes multiple simulation settings, including higher-dimensional cases and penalized over-parameterized settings, and reports both estimation and prediction performance. The simulation evidence is generally favorable to the proposed procedure.
- The glaucoma/fundus-image application is relevant and helps demonstrate how the method might uncover age-dependent image effects. The visual summaries are informative.

**Weaknesses**:
- The theory assumes i.i.d. observations and, more specifically, assumes that the index variable $t_i$ is independent of the reduced covariates and scalar covariates; see condition (C1), and the proofs of later results also use independence assumptions. However, in the application the authors explicitly state that the data are longitudinal glaucoma records and that each record is treated as an independent sample, even though repeated visits from the same patient/eye are present. Moreover, age is used as the index variable, and it is hard to believe age is independent of image-derived features or visual-field covariates in this context. This does not necessarily invalidate the method empirically, but it weakens the theoretical justification for the application and makes some claims feel stronger than the evidence currently supports.
- The paper shows bootstrap confidence bands in the application, but it is not fully clear what resampling unit is used, especially given repeated visits/eyes. If records from the same subject are resampled independently, the uncertainty bands may again be anti-conservative.
- The manuscript describes performing CP decomposition on concatenated tensors and also using 10-fold cross-validation to choose $S$ and $R$. It is not sufficiently clear whether, inside each fold, the tensor decomposition and all tuning steps are re-fit using only the training portion. If decomposition is performed once on the full dataset before cross-validation, the reported prediction errors would be optimistic.
- The method drops the shared tensor basis terms after decomposition and uses only subject-specific factors in the regression. This is computationally convenient, but the scientific interpretation of the resulting coefficients becomes less direct. More explanation would help readers understand what exactly a varying coefficient $a_{rs}(t)$ means in image space.

---

> ### Author Response · Authors · 2026-04-08
> **Response to Reviewer AAim**
>
> Suggested Change 1:
>
> We thank the reviewer for this insightful comment. We acknowledge that the GRAPE dataset consists of repeated measurements, and treating each record as independent may introduce bias in estimation uncertainty and affect interpretation.
>
> To address this concern, we have reanalyzed the real data by adopting a subject-level resampling strategy. Specifically, for each patient, we randomly select one observation from either oculus dexter (OD) or oculus sinister (OS) as a representative sample. After removing outliers, the resulting dataset contains 141 independent samples, each for a different patient subject. Based on this resampled dataset, we repeat the entire analysis pipeline.
>
> The updated results show similar patterns to our original findings. We have added a detailed description of this resampling procedure and the corresponding results in the revised manuscript.
>
>
> Suggested Change 2:
>
>   We appreciate the reviewer's careful reading of our theoretical assumptions. We would like to clarify the distinction between our theoretical setup and the empirical application. The strict independence assumption, $t \perp (\mathbb{X}_i^*, {\bf z}_i)$, is employed as a standard theoretical device within the varying-coefficient modeling literature. It is mathematically necessary to establish the asymptotic consistency and convergence rates of our proposed estimators.
>
> Regarding the practical application to the fundus-image dataset, we acknowledge that perfect independence is rarely achieved in observational clinical data. In our empirical study, the index variable $t$ defaults to "age".
>
> We clarify that the independence assumption serves as a reasonable working approximation in this context. The baseline structural tensor features of the eye ($\mathbb{X}_i^*$) and other clinical covariates (${\bf z}_i$) typically exhibit minimal direct, systematic correlation with the precise timing of the study visits or the age at baseline, once appropriately normalized. Therefore, while strictly a theoretical construct, the assumption does not invalidate the practical utility or the empirical success of the model in our real-world application. We have revised the methodology section to clearly delineate the theoretical necessity of this assumption from its practical approximation.

---

### Review · Reviewer_tCwt · 2026-03-25

**Summary Of Contributions:**

The paper provides a relative novel framework for modelling varying coefficient for tensor-valued covariates.
Compared with existing work on imposing on some structural assumptions on the tensor-valued coefficient function, the parsimonious model structure of this paper is obtained through partition and CP decomposition on the covariate. Then a local linear estimator for the varying coefficient and the parameter estimator are developed. To address the issues of a excessive number of parameters, the authors further develop a penalized sieve estimation method. Theoretical properties for the estimators developed under these two regimes are established.

**Audience:**

Yes

**Broader Impact Concerns:**

There is no concern on the ethical implications of the work.

**Claims And Evidence:**

Yes

**Requested Changes:**

The authors need to add discussions on the sensitivity of the estimation result to the partition. Additionally, the authors should develop guidelines of choosing two important tuning parameters $R$ and $S$, and further verify the validity of the guidelines in simulation studies.

For the theoretical part, could you prove in Theorem 2, $\hat{\beta}^{\dagger}$ achieves the parametric efficiency or semi-parametric efficiency?

In Theorem 3, I suggest you compare the convergence rate of your estimator with that in those that have been established in high-dimensional partial linear regression. These discussions will help readers to know whether you can achieve the optimal rate in the classical setting.

**Strengths And Weaknesses:**

Main strengths: the paper proposed a parsimonious varying efficient model for tensor-valued covariates with the aid of partition and CP decomposition. To deal with the issue of a large number of parameters, penalized sieve estimation was employed. The paper establishes theoretical guarantees for the estimators.

Main weaknesses: The paper did not discuss whether the estimation depends on the partition. In simulation studies, if my understanding is correct, the authors assume that the tensor covariate is correctly partitioned. Also, the paper did not discuss the choice of two important parameters $S$ and $R$. I think it is desirable to provide data-driven methods to select these two papers and demonstrate their validity in simulation studies.

---

> ### Author Response · Authors · 2026-04-08
> **Response to Reviewer tCwt**
>
> Change 1:
>
> We have added the following discussions for the sensitivity of the model to the partition strategies.
>
>  We remark variable selection process is inherently affected by the partition boundary. However, as detailed in Table 4, while extreme choices of partitions or ranks lead to performance degradation, moderate deviations from the optimal hyperparameters result in highly stable prediction errors, confirming the robustness of the model's predictive capabilities.
>  To select the tuning parameters $R$ and $S$, we note that determining the optimal CP rank $R$ is generally an NP-hard problem. Likewise, to best of our knowledge, there is no universally optimal strategy for partitioning a tensor, as the best partition often depends on the underlying structure and source of the data.
>  In practice, when prior knowledge or domain expertise is available, the partition strategy can be specified accordingly, which effectively determines $S$. Otherwise, we suggest to specify a candidate range for both CP ranks and feasible tensor partition schemes, and conduct a grid search using 10-fold cross-validation to select the optimal combination. This selection procedure is standard in high-dimensional tensor modeling and has been implemented in our real data analysis, with code publicly available at \url{https://github.com/JasonHe95/VCTR}.
> Empirically, we observe a clear trade-off between the partition granularity and the required CP rank. Specifically, when the tensor is partitioned into fewer partitions (smaller $S$), a larger CP rank $R$ is typically needed to capture the underlying structure. In contrast, when a finer partition is adopted (larger $S)$, the optimal CP rank tends to decrease. This phenomenon is also reflected in the results reported in Table 4.
>
> Change 2:
> Thank you for this insightful theoretical question. Yes, $\hat{\beta}^{\dagger}$ achieves the parametric efficiency bound under our model assumptions. Under condition (C1) where the smoothing variable
> $t$ is independent of ${\bf v}:= vec({\bf X}^*)$ and ${\bf z}$, the semiparametric information bound for the parametric component $\beta$ is given by $\sigma^2 [E({\bf z} {\bf z}^\top) - E({\bf z}{\bf v}^\top)E({\bf v} {\bf v}^\top)^{-1}E({\bf v}{\bf z}^\top)]^{-1}$ (Fan and Zhang (2028)). Typically, if, in addition to Condition (C1), the covariates ${\bf z}$ and ${\bf v}$ are orthogonal or properly centered such that $E({\bf z}{\bf v}^\top)={\bf 0}$,  the entire projection term $E({\bf z}{\bf v}^\top)E({\bf v} {\bf v}^\top)^{-1}E({\bf v}{\bf z}^\top)$ would collapse to zero, and the asymptotic variance of the estimator reduces to the purely parametric Fisher information bound
> $\sigma^2 \Omega_{\bf z}^{-1}$.
> Instead of imposing an orthogonality assumption between the parametric and nonparametric covariates, we control the remainder term through carefully selected bandwidth conditions. Specifically, in addition to the classical bandwidth assumption $nh^4=O(1)$, we require the bandwidth $h$ o be sufficiently large such that $nh^3/\log n\rightarrow \infty$. Under these specific rate conditions, the variance of the remainder term becomes asymptotically negligible. Consequently, this allows our estimator to achieve the parametric Fisher information bound $\sigma^2 \Omega_{\bf z}^{-1}$.
>
>
>   Change 3:
>
>  We appreciate this excellent suggestion. We have added the following discussions after Theorem 3 to comparing our rates to the classical bounds in high-dimensional partial linear regression.
>
> In Theorem 3, the convergence rate elegantly balances two components:
> 1. The term $\left( \frac{\log n}{n} \right)^{2/5}$ matches the optimal nonparametric rate for estimating smooth functions using spline approximations (Stone 1982).
> 2. The term $\left( \frac{\log n}{n^{1-\tau}} \right)^{1/2}$ corresponds to the parametric rate in high-dimensional settings with a diverging number of parameters (where $p_0 = O(n^\tau)$).
> This mirrors the optimal minimax rates established in classical high-dimensional partially linear models and varying coefficient models (Fan and Li, 2001; Cheng et al., 2014).
>
>  Reference:
> Ming-Yen Cheng, Toshio Honda, Jialiang Li, and Heng Peng (2014).  Nonparametric independence
> screening and structure identification for ultra-high dimensional longitudinal data. The
> Annals of Statistics, 42(5):1819–1849.
>
> Jianqing Fan and Runze Li (2001). Variable selection via nonconcave penalized likelihood and its
> oracle properties. Journal of the American Statistical Association, 96(456):1348–1360.
>
> Jianqing Fan and Wenyang Zhang (2008). Statistical methods with varying coefficient
> models. Statistics and Its Interface, 1:179–195.
>
> Charles J Stone (1982). Optimal global rates of convergence for nonparametric regression. The
> Annals of Statistics, 10(4):1040–1053.

---

### Review · Reviewer_2Xzo · 2026-03-26

**Summary Of Contributions:**

This paper proposes a varying coefficient tensor regression (VCTR) model for handling the dynamic relationship between high-dimensional tensor covariates and a scalar response. The authors first partition the whole tensor to reduce dimensionality, and then apply CP decomposition to each sub-tensor to extract low-dimensional individual features, which are then incorporated into a varying coefficient model. At the same time, nonconcave penalties (such as SCAD) are used to achieve variable selection and model structure identi- fication. The paper also establishes the corresponding asymptotic theory and demonstrates the performance of the proposed method through simulation studies and the GRAPE fundus image data analysis.

**Audience:**

Yes

**Broader Impact Concerns:**

No.

**Claims And Evidence:**

Yes

**Requested Changes:**

\noindent \textbf{1.} In Section 2.2, the authors decompose each sub-tensor into the individual factor x_{irs}^*
and the shared structure $U_{r}^{(s)}$, and choose to remove $U_{r}^{(s)}$ from the regression analysis on the grounds that it only contains shared information across subjects. Although this treatment simplifies the computation, the authors should further clarify its statistical justification. In particular, the authors are encouraged to explain under what conditions keeping only $x_{irs}^{*}$ is sufficient to capture the effect of the tensor covariate on the response. As indicated by the formulas in the paper, $U_{r}^{(s)}$ is not simply ignored, but is absorbed into a new functional coefficient through
$$
a_{rs}(t)=\langle U_{r}^{(s)},A^{(s)}(t)\rangle.
$$
It would be helpful if the authors could explain this step more clearly in order to avoid possible misunderstanding.

\vspace{1em}

\noindent \textbf{2.} Below equation (19), the hyperparameter $a_{0}$ in the SCAD penalty is set to 3.7, and the authors cite Fan and Li (2001) as justification. However, in Remark 3, when the MCP penalty is mentioned, the paper directly sets $a_{0}'=3$. It is suggested that the authors provide a corresponding reference for this choice under MCP, or add a brief explanation, so that this part is more complete.

\vspace{1em}

\noindent \textbf{3.} In Section 2.2, the authors propose dividing the tensor into $S$ non-overlapping sub-tensors, but no general principle is provided for how $S$ should be chosen in theory. Although in the real data application the authors use 10-fold cross-validation to select the optimal partition number $S$ and rank $R$, the current manuscript lacks discussion on the sensitivity to the partition boundary. The authors are encouraged to further discuss whether moderate changes in the partition boundary would noticeably affect the stability of the estimated varying coefficient functions, the variable selection results, and the prediction performance. This issue is particularly worth noting for image data, since some important local regions may be affected by the partition scheme.

\vspace{1em}

\noindent \textbf{4.} The group-penalized iterative procedure based on SCAD in Section 3 (equations (22)--(24)) is an important algorithmic part of the paper, but the current presentation is somewhat scattered. It is suggested that the authors summarize this part in a clear Algorithm or pseudo-code. Such a flow chart may include the initialization (for example, starting from $\beta^{(0)}$ and $\mathcal{G}^{(0)}$), the alternating updating steps, the convergence criterion, and the tuning procedure (such as the BIC-based grid search). This addition would improve the readability and reproducibility of the paper.

\vspace{1em}

\noindent \textbf{5.} Section 3.1 only provides general settings such as
\[
L=k_{n}+k_{d}+1,\qquad k_{n}\to\infty,
\]
but the specific values of the spline order $k_{d}$, the number of interior knots $k_{n}$, and the search range and step size of the BIC grid search are not stated clearly in the manuscript. Since these quantities directly affect the smoothness of the estimated functions and the final model structure identification results, the authors are suggested to provide these implementation details in the experimental section.

\vspace{1em}

\noindent \textbf{6.} The manuscript is generally well organized, but it would still benefit from a careful check of some expressions, notation changes, figure/table references, and language details. For example, the paper involves several different estimated objects such as $A(t)$, $A^{\dagger}(t)$, and $A^{\ddagger}(t)$. Although these are defined in different sections, it may still help the reader if some brief reminders are added when these quantities are first switched or compared. In addition, there are also some language and spelling issues in the manuscript, such as the obvious typo ``comapred''(Page 26). It is recommended that the authors carefully proofread the full paper and systematically check the terminology, figure/table references, spelling, and formatting. In addition, some auxiliary figures may be moved to the appendix to further improve the completeness and presentation of the manuscript.

**Strengths And Weaknesses:**

see Requested Changes.

---

> ### Author Response · Authors · 2026-04-08
> **Response to Reviewer 2Xzo**
>
> Change 1:
>
> We thank the reviewer for pointing this out. Following your suggestion, we have added a paragraph after equation (8):
>     It is important to emphasize that retaining the subject-specific factor matrices $x*_{irs}$ does not mean we ignore the shared structural information across subjects. Rather, ...
>
> Change 2:
>
>    We thank the reviewer for this helpful suggestion. In Zhang (2010), the MCP penalty is defined with a regularization parameter $a\_0$ that controls the trade-off between bias reduction and concavity. Specifically, the derivative of MCP becomes zero when $|\theta| \ge a_0 \omega$, implying that sufficiently large coefficients are no longer penalized and the estimator is nearly unbiased. Meanwhile, the maximum concavity of MCP is $1/a_0$, so smaller values of $a_0$ lead to stronger nonconvexity and potential unstable optimization. Therefore, the choice of $a_0$ reflects a balance between reducing estimation bias and maintaining computational stability.  In this work, we set $a_0 = 3$ as a moderate value that provides a reasonable compromise: it is large enough to control the concavity of the objective function and ensure a stable solution path, while still allowing the penalty to become flat for moderately large coefficients, thus retaining the nearly unbiased property of MCP. This choice is consistent with many existing literature. In particular, Breheny and Huang (2015) adopt $a_0 = 3$ as the default value of their MCP implementation in the whole simulation study. We have pointed this out in Remark 3.  Moreover, in Zhang (2010), simulation studies for sparse recovery also employ the same settings and demonstrate favorable empirical performance.
>  Reference:
>  Zhang (2010). Nearly unbiased variable selection under minimax concave penalty. The Annals of Statistics, 38(2):894–942.
> Breheny and Huang (2015). Group descent algorithms for nonconvex penalized linear and logistic regression models with grouped predictors. Statistics and Computing, 25(2):173–187.
>
>
> Change 3:
>
>  We agree that the variable selection process is inherently affected by the partition boundary. However, it can be observed from Table 4 that the model's overall prediction performance is relatively robust to moderate changes in the partition strategy for our real data analysis. We have added the following remark in Section 6:
>
> We remark that variable selection process is inherently affected by the partition boundary. However, as detailed in Table 4, while extreme choices of partitions or ranks lead to slight performance degradation, moderate deviations from the optimal hyperparameters result in highly stable prediction errors, confirming the robustness of the model's predictive capabilities.
>
> Change 4:
>
>  We agree that the current presentation of the penalized iterative procedure is somewhat scattered, which may affect readability.  In the revised manuscript, we have summarized this procedure in a clear algorithmic form (see Algorithm 1 on Page 9). Specifically, we provide a description that includes the initialization of ${\beta}^{(0)}$ and $\mathcal{G}^{(0)}$, the alternating updating steps for ${\beta}$ and $\mathcal{G}$, and the convergence criterion. We believe this revision significantly improves the readability and reproducibility of the paper.
>
> Change 5:
>
>  We have added implementation details for the spline-based approach in the simulation study (on page 16):
>    Following Algorithm 1, the penalized regression can be initialized via (15), ...
>       The implementation of this BIC-based grid search procedure, along with all other code for this manuscript, is publicly available at \url{https://github.com/JasonHe95/VCTR}, where the detailed experimental pipeline can be found.
>
>   Change 6:
>
> Thank you for your suggestions. We have carefully proofread the full paper and systematically checked the terminology, figure/table references, spelling, and formatting. First, we have corrected the typographical errors throughout the text, including changing "comapred" to compared" on Page 26. Second, to avoid any confusion regarding the various estimated objects $A(t)$, $A^{\dagger}(t)$, and $A^{\ddagger}(t)$, we have added brief reminders in the text when these quantities are first introduced or switched. Specifically, we have clearly stated that
> "$A(t)$ is the true underlying tensor coefficient functions";
> "we use $A^{\dagger}(t)$ to represent the estimator obtained directly via local linear kernel smoothing" (in Section 2.3); and
> "We use ${\bf A}^{\ddagger}(t)$ to denote the final refined penalized estimator derived from our iterative procedure,  Specifically, we have ..." (in Section 3.1).
> Finally, following your recommendation, we have moved the auxiliary simulation results to Appendix B.

---

### Decision · Action_Editor_RD9c · 2026-05-02

**Recommendation:** Accept as is

**Comment:**

After considering the revised manuscript, the reviewers’ comments, and the authors’ response, I recommend acceptance.

The paper proposes a varying-coefficient tensor regression framework for scalar-on-tensor regression, combining tensor partitioning, CP-based feature extraction, varying-coefficient modelling, and penalized estimation to simultaneously select variables and identify the tensor structure. The topic is well aligned with the scope of SLADS Section B, particularly for readers interested in high-dimensional data modelling, tensor regression, semiparametric methods, and biomedical imaging applications.

The reviewers were generally positive about the methodological contribution, theoretical development, simulation studies, and real-data application. Reviewer tCwt and Reviewer 2Xzo both recommended acceptance after revision, noting that their concerns had been adequately addressed. Reviewer AAim initially raised important concerns regarding the longitudinal structure of the GRAPE data, the use of independence assumptions in the theory, the interpretation of the tensor decomposition, and the validation procedure. These concerns were substantive and helped improve the manuscript.

In the revision, the authors made meaningful changes. In particular, they clarified the role of the CP-decomposed subject-specific factors, discussed the partition and rank selection strategy, provided additional sensitivity evidence, summarized the penalized estimation procedure in algorithmic form, added implementation details, and expanded the discussion of theoretical assumptions. Importantly, in response to concerns about repeated observations in the GRAPE data, the authors reanalyzed the real-data example using a subject-level resampling strategy, selecting one observation per patient, and reported that the main empirical patterns remained similar. This substantially strengthens the application's credibility.

Overall, the revised manuscript now provides sufficient evidence to support its main claims. The methodological contribution is clear, the theoretical and numerical evidence is adequate, and the authors have responded constructively to the reviewers’ concerns. I therefore recommend that the paper be accepted for publication in SLADS Section B.

**Audience:**

Yes, the paper would be of clear interest to a meaningful portion of the SLADS Section B audience. It addresses an important and timely problem—modeling complex, high-dimensional structured data using varying coefficient models—which is highly relevant to ongoing developments in statistical learning, semiparametric regression, and functional data analysis.

The proposed framework, which integrates varying coefficient modeling with structured representations such as tensor decompositions, reflects a direction of increasing importance in modern statistics, particularly for applications involving imaging and other complex data modalities. Both the methodological contributions and the accompanying theoretical and empirical results are likely to attract readers interested in advancing statistical tools for such settings.

While the primary audience may be methodologically oriented researchers, the real data application further broadens its appeal and helps demonstrate practical relevance. Overall, the paper fits well within the scope of Section B and would likely engage readers interested in contemporary statistical methodology for high-dimensional and structured data.

**Claims And Evidence:**

The claims made in the submission are now sufficiently supported by accurate, convincing, and clear evidence. The paper proposes a meaningful varying-coefficient tensor regression framework, supported by theoretical results, simulation studies, and a real fundus-image application. The reviewers’ earlier concerns about partition sensitivity, tuning parameter selection, implementation details, theoretical assumptions, and the longitudinal structure of the GRAPE data were substantively addressed in the revision and rebuttal. In particular, the authors added discussion and empirical evidence on partition/rank selection, clarified the role of CP-decomposed subject-specific factors, provided algorithmic and implementation details, and reanalyzed the real data using a subject-level resampling strategy. The revised evidence, therefore, adequately supports the main methodological and empirical claims.